# Repulsive Sema3E-Plexin-D1 signaling coordinates both axonal extension and steering via activating an autoregulatory factor, Mtss1

**Namsuk Kim[1†], Yan Li[1†], Ri Yu[1], Hyo-Shin Kwon[1], Anji Song[1], Mi-Hee Jun[1], Jin-Young Jeong[1,2], Ji Hyun Lee[1], Hyun-Ho Lim[1], Mi-Jin Kim[3], Jung-Woong Kim[3], Won-Jong Oh[1]***

[1]Neurovascular Unit Research Group, Korea Brain Research Institute, Daegu, Republic of Korea; [2]Department of Brain and Cognitive Sciences, Daegu Gyeongbuk Institute of Science and Technology, Daegu, Republic of Korea; [3]Department of Life Sciences, Chung-Ang University, Seoul, Republic of Korea

**Abstract** Axon guidance molecules are critical for neuronal pathfinding because they regulate directionality and growth pace during nervous system development. However, the molecular mechanisms coordinating proper axonal extension and turning are poorly understood. Here, metastasis suppressor 1 (Mtss1), a membrane protrusion protein, ensured axonal extension while sensitizing axons to the Semaphorin 3E (Sema3E)-Plexin-D1 repulsive cue. Sema3E-Plexin-D1 signaling enhanced Mtss1 expression in projecting striatonigral neurons. Mtss1 localized to the neurite axonal side and regulated neurite outgrowth in cultured neurons. Mtss1 also aided Plexin-D1 trafficking to the growth cone, where it signaled a repulsive cue to Sema3E. Mtss1 ablation reduced neurite extension and growth cone collapse in cultured neurons. *Mtss1*-knockout mice exhibited fewer striatonigral projections and irregular axonal routes, and these defects were recapitulated in *Plxnd1*- or *Sema3e*-knockout mice. These findings demonstrate that repulsive axon guidance activates an exquisite autoregulatory program coordinating both axonal extension and steering during neuronal pathfinding.

**\*For correspondence:**
ohwj@kbri.re.kr

[†]These authors contributed equally to this work

**Competing interest:** The authors declare that no competing interests exist.

## Editor's evaluation

In this manuscript, the authors proposed a novel and attractive model to address a fundamental question of how the locational and function of axon guidance molecules are regulated. They presented convincing data to support their working model. They showed important findings that Sema3E-Plexin-D1 signaling regulates the expression of Mtss1, which regulates the localization of Plexin-D1 and contributes to striatonigral axonal growth and turning.

## Introduction

In the developing nervous system, axons of newly generated neurons extend toward destination targets and make connections to establish a functional circuit following an exquisitely designed program. In this long-range pathfinding process, axons encounter attractive and repulsive signals from guidance molecules, and diverse combinations of ligand–receptor pairs communicate signals to a neuron from the environment (*Kolodkin and Tessier-Lavigne, 2011*; *Tessier-Lavigne and Goodman, 1996*). In addition to the conventional guidance mode, which has been established, recent studies

have demonstrated the complexity of signaling through different mechanistic layers, such as crosstalk between guidance molecules (*Dupin et al., 2015*; *Poliak et al., 2015*), guidance switching between different holoreceptor complexes (*Bellon et al., 2010*), or guidance tuning by intrinsic regulators (*Bai et al., 2011*; *Bonanomi et al., 2019*). In general, the specific cognate guidance receptors that sense extracellular signals are mostly localized in growth cones, a specialized structure at the fore of a growing axon, and these receptors convey intracellular signaling cues within neurons (*Dent et al., 2011*; *Franze, 2020*). Therefore, proper signaling from guidance molecules in the growth cone surface is critical as axons travel to their destination. Because the axonal destination can be as far as a meter or more away from the soma, various transport systems consisting of specific adaptors and motor proteins transport guidance proteins to axon terminals and are thus critical for axonal movement (*Dent et al., 2011*; *Winckler and Mellman, 2010*). However, how individual guidance molecules are correctly delivered to growth cones and can accommodate the axonal growth pace is unclear, and the molecular machinery critical for the specific transportation of guidance molecules is unknown.

Growth cones are highly dynamic and motile cellular structures that facilitate axon growth and steering through activated receptors that alter cytoskeletal actin and microtubule assembly (*Lowery and Van Vactor, 2009*; *Vitriol and Zheng, 2012*). Therefore, guidance receptors undoubtedly need to be localized to these protrusive structures to control actin dynamics. Since actin filament assembly is typically accompanied by membrane remodeling, a group of cytoskeletal scaffold proteins linking actin to the cell membrane must be activated (*Vitriol and Zheng, 2012*). One such protein group consists of Bin/Amphiphysin/Rvs (BAR) domain proteins, which have been implicated in many actin-associated membrane functions, such as cell motility, endocytosis, and organelle trafficking (*Chen et al., 2013*). Among these proteins, metastasis suppressor 1 (Mtss1, also called missing in metastasis), one of a few inverse BAR (I-BAR) domain subfamily proteins, is notable due to its capability of forming cellular protrusions by promoting inverse membrane curvature (*Machesky and Johnston, 2007*). Because of its unique ability to connect the plasma membrane inner leaflet with actin, the role of Mtss1 has been characterized in promoting spine protrusions as well as neuronal dendrite growth (*Kawabata Galbraith et al., 2018*; *Saarikangas et al., 2015*; *Yu et al., 2016*). However, in contrast to these many studies of Mtss1 on the dendritic side during development, few studies have investigated whether Mtss1 is expressed and plays a specific role in axons.

Semaphorin 3E (Sema3E), a class 3 secreted semaphorin family protein, conveys guidance signals by directly binding with the Plexin-D1 receptor in both the nervous and vascular systems (*Gu et al., 2005*; *Oh and Gu, 2013a*). The Sema3E-Plexin-D1 pair mainly transmits a repulsive guidance cue via local cytoskeletal changes, thereby inhibiting axonal overgrowth and/or ectopic synapse formation in the central nervous system (*Chauvet et al., 2007*; *Ding et al., 2012*; *Fukuhara et al., 2013*; *Mata et al., 2018*; *Pecho-Vrieseling et al., 2009*). Previous studies have demonstrated that Sema3E-Plexin-D1 signaling is involved in dendritic synapse formation as well as traditional axon projection in the basal ganglia circuitry, which is essential for diverse behavioral and cognitive functions in the brain (*Ding et al., 2012*; *Ehrman et al., 2013*). Notably, Plexin-D1 is expressed only in direct-pathway medium spiny neurons (MSNs) projecting to the substantia nigra pars reticulata (SNr), one of two distinct types of MSNs in the striatum (*Ding et al., 2012*). Plexin-D1-positive striatonigral axons travel through the corridor between the globus pallidus (Gp) and reticular thalamic nucleus (rTh)/zona incerta (ZI), in which Sema3E molecules reside and emit repulsive signals to direct proper pathway formation toward the SNr (*Chauvet et al., 2007*; *Ehrman et al., 2013*). However, how the striatonigral pathway coordinates axonal growth and steering during pathfinding remains largely unknown.

In this study, we investigated the molecular mechanism of the repulsive Sema3E-Plexin-D1 guidance signaling pair in striatonigral-projecting neurons during mouse basal ganglia circuit development. We found that Sema3E-Plexin-D1 signaling coordinates axonal extension and diversion by enhancing the action of the facilitator protein Mtss1 during active striatonigral projection progression. In the context of the important and intricate networks in the brain, this study provides evidence showing that autoregulatory factor expression regulated by guidance signaling leads to the correct neuronal trajectory to the destination.

## Results

### Sema3E-Plexin-D1 signaling regulates Mtss1 expression in the developing striatum

The majority of striatal neurons are MSNs (up to 90%), and the MSNs are equally divided into direct and indirect pathways (*Gerfen and Surmeier, 2011*). In a previous study, we found that *Plxnd1* is selectively expressed in direct-pathway MSNs (also called striatonigral neurons) that project directly to the substantia nigra, with approximately 45% of striatal neurons identified as Plexin-D1-positive neurons (*Ding et al., 2012*). Because of the relative abundance of Plexin-D1-positive neurons, we expected a high probability of discovering the potential downstream responsive genes modulated by Sema3E-Plexin-D1 signaling in the striatum. Therefore, we performed bulk RNA sequencing (RNA-seq) with striatal tissues at P5, when *Plxnd1* expression is high in the striatum, and compared the results obtained with control (*Plxnd1^f/f^*) and conditional neuronal *Plxnd1*-knockout (*Nes-Cre; Plxnd1^f/f^*) mice (*Figure 1A*). *Plxnd1* mRNA ablation in striatal tissues was validated in pan-neuronal *Plxnd1*-knockout (Nestin-Cre) mice (*Figure 1—figure supplement 1A and B*).

Next, we performed gene expression profile analysis. The principal component analysis (PCA) plot showed that *Plxnd1*-knockout accounted for the largest variance, and the results obtained in biological replicates showed high reproducibility (*Figure 1—figure supplement 1C*). Application of a conservative DEseq approach to RNA-seq data analysis confirmed 2360 differentially expressed transcripts (*Figure 1—figure supplement 1D*). Gene Ontology (GO) analysis was then performed, and biological connections between upregulated (1240 transcripts) and downregulated (1120 transcripts) differentially expressed genes (DEGs) in *Plxnd1*-knockout mice compared to wild-type (control) mice were identified (*Figure 1—figure supplement 1E*). Clustering of the downregulated DEGs in *Plxnd1*-knockout mice enabled their classification into several categories that were associated with axon guidance, regulation of dendritic spine morphology, and neuronal projection. The volcano plots present the statistical significance of differential transcript expression with the respective fold change values (p<0.05, absolute log2 [fold change, FC] > 1) compared to the expression observed in the control group (*Figure 1B*).

Among the downregulated genes, Mtss1 was particularly notable due to its high relevance to actin cytoskeletal rearrangement (*Kawabata Galbraith et al., 2018*; *Lin et al., 2005*; *Saarikangas et al., 2015*). *Mtss1* gene expression was verified by quantitative RT-PCR (qRT-PCR) performed with *Plxnd1*-knockout striatal tissues (*Figure 1C*). The Mtss1 protein levels were also markedly decreased in the knockout mice at P5 (*Figure 1D and E*). We also analyzed Mtss1 expression in *Sema3e*-knockout striatal samples obtained at P5 and found that its expression was decreased, but less dramatically than it was in *Plxnd1*-knockout striatal samples (*Figure 1F and G*). These results suggest that Sema3E-Plexin-D1 signaling activation can increase Mtss1 expression in striatal neurons during development.

### Mtss1 is selectively expressed in striatonigral-projecting neurons during the active pathfinding period

To determine whether this Mtss1 expression is specific to Plexin-D1-positive neurons, we first performed fluorescence in situ hybridization. Mtss1 expression significantly overlapped with *Plxnd1*-positive neurons in both the cortex and striatum, and its expression was reduced in *Plxnd1*-knockout mice (*Figure 1H and I*). Moreover, when we performed immunostaining with Drd1a-tdT mice, in which direct-pathway MSNs fluoresced red (*Ade et al., 2011*), Mtss1 expression significantly overlapped with Drd1a-tdT striatal neurons (*Figure 1J*; 98.2 ± 1.72%), suggesting that Plexin-D1 signaling mediates Mtss1 expression selectively in striatonigral projecting MSNs. Although Mtss1 was expressed at a low level regardless of Plexin-D1 presence at E16.5, it seemed to be under the control of Plexin-D1 signaling in the developing striatum from the last gestation period to the early postnatal period (*Figure 1K and L*). Furthermore, the Sema3E ligand activating the Plexin-D1 receptor was predominantly expressed in the thalamus and released into the striatum, probably during thalamostriatal projection at E16.5, as observed in the early postnatal stage in a previous study (*Figure 1—figure supplement 2*; *Ding et al., 2012*). Next, we analyzed the expression profiles of Plexin-D1 and Mtss1 from the developmental stage to the adult stage. Both Plexin-D1 and Mtss1 were expressed in the embryonic striatum, and their expression was elevated in the perinatal stage. Interestingly, Mtss1 expression was maintained at a relatively high level from E18.5 to P5 and then declined sharply and

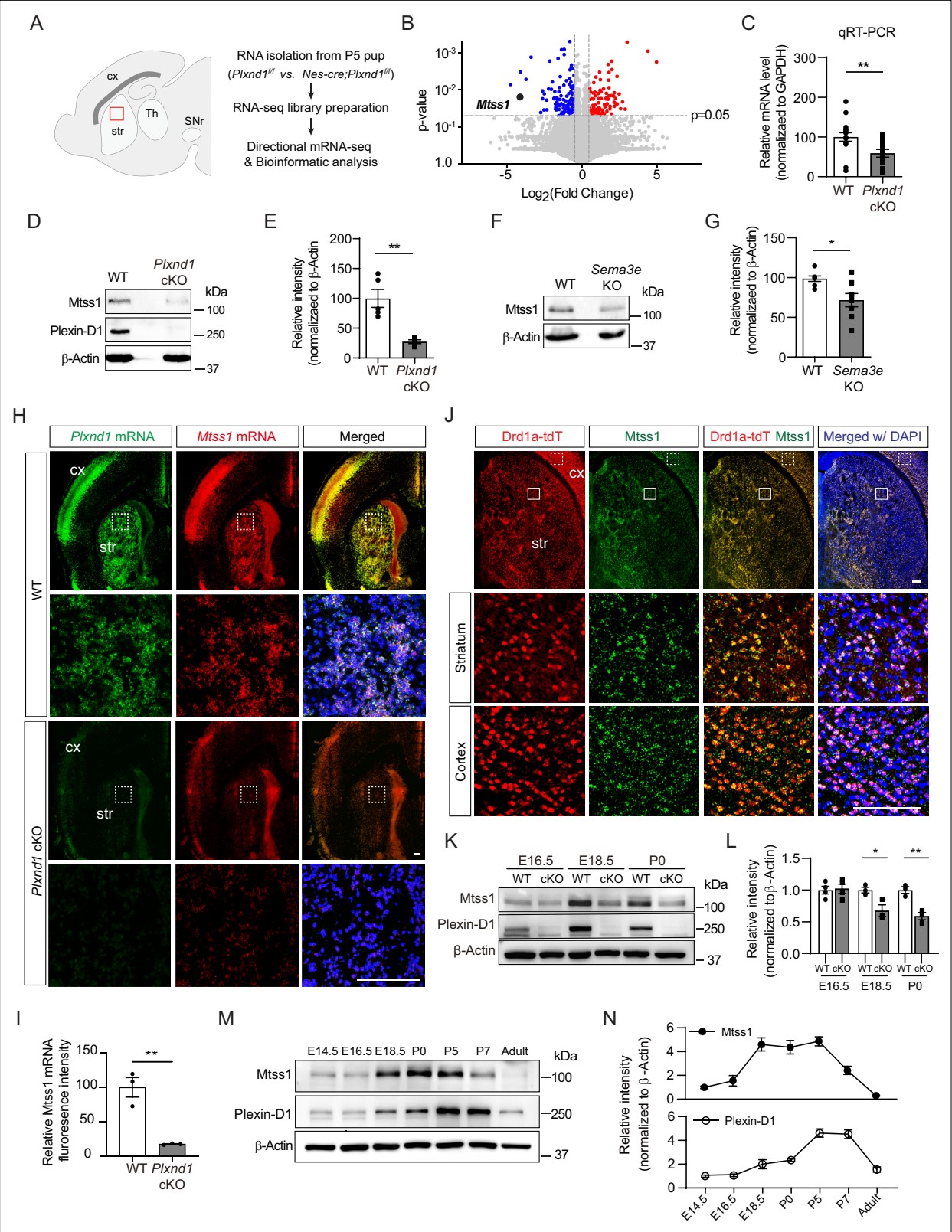

**Figure 1.** Sema3E-Plexin-D1 signaling induces Mtss1 expression selectively in developing striatonigral projecting neurons. (**A**) RNA sequencing (RNA-seq) analysis of wild-type (WT) (*Plxnd1^{f/f}*) and conditional neuronal knockout (cKO) (*Nes-cre; Plxnd1^{f/f}*) pups at P5. The box in red indicates the dorsal striatum region from which RNA was isolated. (**B**) Volcano plot of significant differentially expressed genes (DEGs) between WT and *Plxnd1* cKO. Blue and red circles indicate significantly down- and upregulated genes, respectively, as indicated by a fold change greater than 2. (**C**) Relative levels

*Figure 1 continued on next page*

*Figure 1 continued*

of *Mtss1* expression in the striatum of WT or *Plxnd* cKO mice at P5 were compared by quantitative RT-PCR (RT-qPCR). n = 16 for WT mice, n = 14 for *Plxnd1* cKO mice in four independent experiments. (**D, E**) Western blot images showing Mtss1 expression in the striatum of WT or *Plxnd1* cKO mice and quantification. The values are averaged from n = 5 for WT mice and n = 4 for *Plxnd1* cKO mice. (**F, G**) Western blot images and quantification of Mtss1 expression in the striatum of WT or *Sema3e* KO mice at P5. WT mice, n = 7, and *Sema3e* KO mice, n = 8. (**H**) Fluorescence in situ hybridization (FISH) for *Plxnd1* mRNA (green) and *Mtss1* mRNA (red) in the striatum of WT or *Plxnd1* cKO mice at P5. White dotted boxes are shown in the inset image on the bottom. Scale bar, 200 µm. (**I**) Quantification of fluorescence intensity to measure the expression levels of Mtss1 in (**H**). WT mice, n = 3, and *Plxnd1* cKO mice, n = 3. (**J**) Immunohistochemistry showing tdTomato-expressing Drd1a+MSNs (red) and Mtss1 (green) in the striatum of *Drd1a-tdT* mice at P5. The small boxes in the striatum and cortex are shown at better resolution in the inset images. Scale bar, 100 µm. (**K**) Western blot images showing the expression of Mtss1 and Plexin-D1 in the striatum of WT or *Plxnd1* cKO mice at different developmental stages ranging from embryonic day 16.5 (E16.5) to postnatal day 0 (P0). (**L**) Quantification of band intensity in (**K**). WT, n = 4, and KO, n = 4 at E16.5, WT, n = 3, and KO, n = 3 at E18.5, WT, n = 3, and KO, n = 3 at P0. (**M, N**) Western blot images showing the temporal expression of Plexin-D1 and Mtss1 in the striatum from E14.5 to adulthood (8 weeks old) and quantification. Error bars, mean ± SEM; *p<0.05, **p<0.01 by Student's *t*-test for all quantifications. The values represent the average band intensity, n = 3 at each age. str, striatum; cx, cortex; Th, thalamus; SNr, substantia nigra.

The online version of this article includes the following source data and figure supplement(s) for figure 1:

**Source data 1.** Western blots shown in *Figure 1D, F, K, and M*.

**Figure supplement 1.** Identification of Mtss1 in the striatum on P5 through RNA sequencing (RNA-seq) analysis.

**Figure supplement 1—source data 1.** RT-PCR shown in *Figure 1—figure supplement 1B*.

**Figure supplement 2.** Sema3E expression through thalamostriatal projections at E16.5.

**Figure supplement 2—source data 1.** Western blots shown in *Figure 1—figure supplement 2C*.

---

disappeared in the adult striatum, and although Plexin-D1 showed a similar expression pattern, its expression was maintained at a low level in the adult striatum, presumably to regulate other functions such as thalamostriatal synapse formation (*Figure 1M and N*; *Ding et al., 2012*).

## Sema3E-Plexin-D1 signaling regulates Mtss1 expression in cultured medium spiny neurons

To determine whether Plexin-D1-driven Mtss1 expression can be recapitulated in vitro, we compared the Mtss1 levels in cultured striatal neurons isolated from wild-type and *Plxnd1*-null mice. In wild-type neurons, both Plexin-D1 and Mtss1 expression levels were low at day 3 in vitro (DIV3) and then increased by DIV6. In contrast, *Plxnd1*-knockout neurons failed to elevate Mtss1 expression by DIV6, suggesting that Mtss1 expression is induced at the cellular level rather than by indirect systemic changes at the circuit level in vivo (*Figure 2A and B*). In addition, the expression of Mtss1 was decreased in cultured *Sema3e*-knockout neurons (*Figure 2C and D*). Next, to further confirm that the Sema3E-Plexin-D1 guidance pair is required for the activation of Mtss1 expression in striatonigral neurons, we supplemented the *Sema3e*- or *Plxnd1*-knockout neurons with exogenous Sema3E ligand according to the scheme in *Figure 2E*. Given that some Sema3E ligands are naturally present in our striatal cultures from globus pallidus neurons, conducting tests on the exogenous Sema3E effect in the *Sema3e*-knockout cultures is ideal to minimize experimental variation. In fact, Mtss1 expression was increased by Sema3E replenishment in *Sema3e*-knockout neurons, whereas it was not altered in *Plxnd1*-knockout neurons (*Figure 2F–I*). Moreover, since Akt is already known to mediate the Sema3E-Plexin-D1 signaling cascade in neurons (*Burk et al., 2017*), we next examined whether disturbing Akt activity alters Mtss1 expression in cultured neurons. Treatment with an Akt inhibitor, MK2206, down-regulated Mtss1 expression but caused no changes in Plexin-D1 levels (*Figure 2J–M*). Furthermore, the elevated expression of Mtss1 due to the addition of exogenous Sema3E supplement in *Sema3e*-knockout neurons was also diminished by MK2206, suggesting that the Sema3E-Plexin-D1 pathway is involved in Mtss1 expression through Akt signaling (*Figure 2N and O*). These results suggest that Mtss1 is a downstream expression target of Sema3E-Plexin-D1 signaling in direct-pathway MSNs.

## Mtss1 is important for neurite extension in direct-pathway MSNs

Since Mtss1 has a well-characterized role in the regulation of filopodia and spine precursors (*Saari-kangas et al., 2015*; *Yu et al., 2016*), we first tested the morphological changes induced by Mtss1 in COS7 cells. Mtss1 was weakly expressed in COS7 cells, but its levels were not altered after over-expressing Plexin-D1 with or without Sema3E (*Figure 3—figure supplement 1A and B*). Mtss1

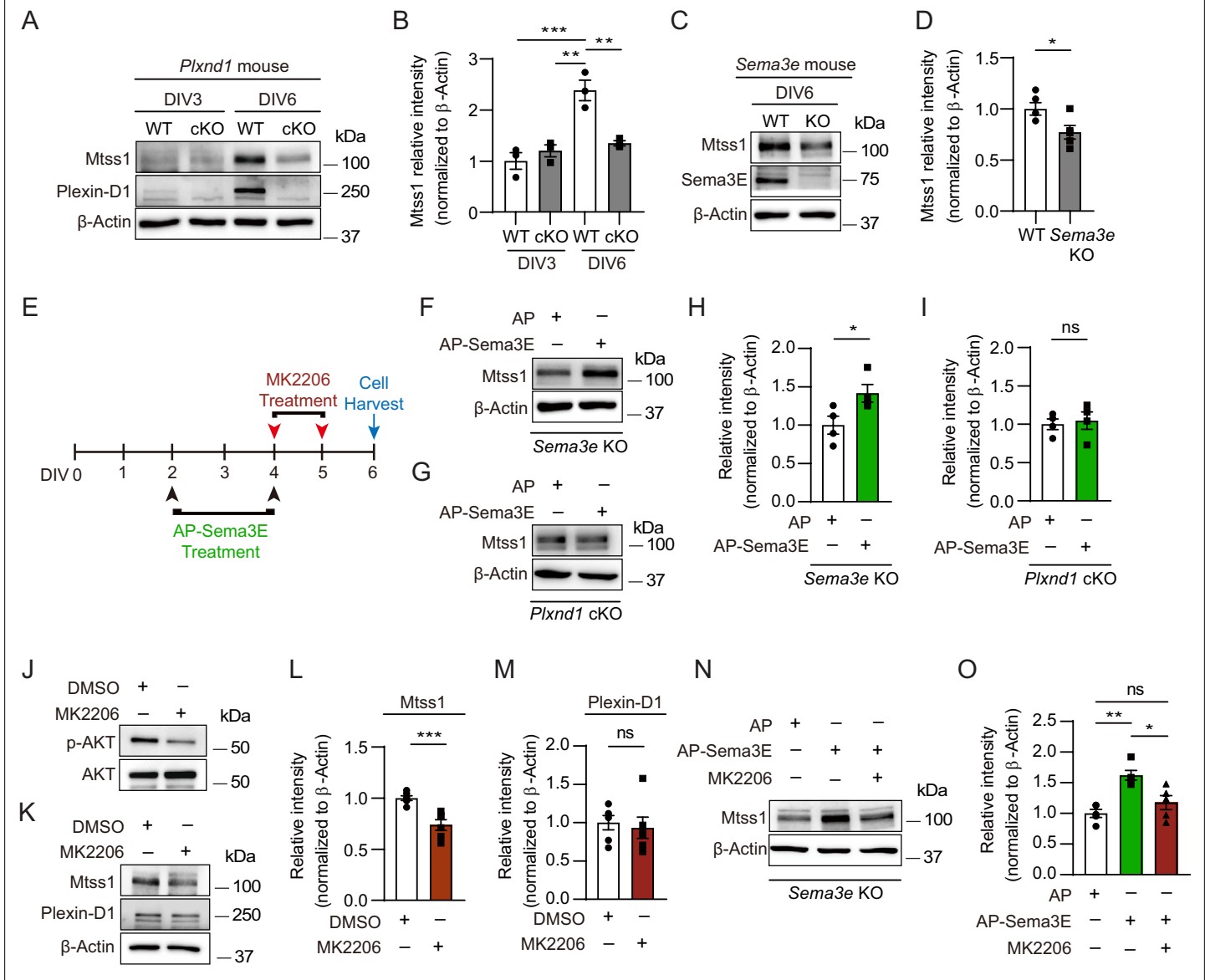

**Figure 2.** In cultured medium spiny neurons (MSNs), Mtss1 expression is directly regulated by Sema3E-Plexin-D1 signaling through the AKT pathway. (**A**) Western blot images showing Mtss1 expression in MSNs derived from the striatum of wild-type (WT) or *Plxnd1* conditional knockout (cKO) mice at P0 and measured at DIV3 and DIV6 in culture. (**B**) Quantification of band intensity in (**A**). Two-way ANOVA with Tukey's post hoc correction for multiple comparisons; n = 3. (**C**) Mtss1 expression in MSNs obtained from the striatum of WT or Sema3e KO mice at P0 and measured at DIV6 in culture. (**D**) Quantification of the blots shown in (**C**). Student's *t*-test; n = 5 for WT, n = 5 for KO in five independent experiments. (**E**) Schematic illustration of the experimental strategy for Sema3E-ligand or MK2206, an AKT inhibitor treatment in MSN culture. (**F, G**) Western blot images showing Mtss1 expression after AP-Sema3E (2 nM) treatment in cultured MSNs derived from Sema3e KO mice or *Plxnd1* cKO mice. (**H, I**) Quantification of (**F, G**). Student's *t*-test; AP, n = 4, AP-sema3E, n = 4 for *sema3e* KO mice, AP n = 4, AP-sema3E n = 4 for *Plxnd1* cKO mice in three independent experiments. (**J, K**) Western blot to analyze the expression of Mtss1 and Plexin-D1 after MK2206 (100 nM) treatment in cultured MSNs and subsequent quantification for band intensity (**L, M**). Student's *t*-test; n = 6 for sham, n = 6 for MK2206 in six independent experiments. (**N O**) Western blot image and analysis showing Mtss1 expression in *Sema3e* knockout MSNs treated with MK2206 after incubation with AP-Sema3E. Two-way ANOVA with Tukey's post hoc correction for multiple comparisons; n = 5 in five independent experiments. Error bars, mean ± SEM; *p<0.05, **p<0.01, ***p<0.001 by indicated statistical tests. .

The online version of this article includes the following source data for figure 2:

**Source data 1.** Western blots shown in *Figure 2A, C, F, G, J, K, and N*.

overexpression in COS7 cells led to a diverse degree of morphological changes, such as excessively spiky or thin and long processes, and highly localized in F-actin-enriched protrusions. However, overexpression of Mtss1 lacking the I-BAR domain failed to generate these protrusive shapes (*Figure 3—figure supplement 1C and D*). Mtss1 lacking WH2 domain showed much weaker effect because WH2 is an important region for Mtss1 interaction with F-actin (*Mattila et al., 2003*). These results suggest that Mtss1 is involved in F-actin dynamics and thus may be an important regulator of neurite outgrowth in cultured MSNs, similar to the situations in other types of neurons (*Kawabata Galbraith et al., 2018*; *Saarikangas et al., 2015*; *Yu et al., 2016*).

Interestingly, Mtss1 was significantly localized to the Tau-positive-axonal side of cultured MSNs at DIV3 (*Figure 3A*). We measured the neurite length of the direct-pathway MSNs that had been genetically labeled with red fluorescence in Drd1a-tdT crossbred reporter mice. We observed significant growth retardation in MSNs lacking *Mtss1* compared to wild-type neurons at DIV3 and DIV6 (*Figure 3B, C and E, F*). Furthermore, we also observed that *Mtss1*-deficient neurons failed to extend neurites more than twice the size of the cell body, suggesting that a lack of *Mtss1* presumably caused a severe neurite growth defect (*Figure 3D*). Next, we analyzed neurite length in *Plxnd1*-knockout neurons. Interestingly, there was no difference in neurite length between Drd1a-positive wild-type MSNs and *Plxnd1*-knockout MSNs at DIV3 (*Figure 3G and H*), probably due to the low Mtss1 induction shown in *Figure 2A*, whereas the length was significantly reduced in the *Plxnd1*-deficient MSNs at DIV6 (*Figure 3I and J*). These results suggest the possibility that Mtss1 expression is independent of Plexin-D1 signaling in the young neurons as observed in early development (*Figure 1K*). However, Mtss1 expression appears to be highly induced by Sema3E-Plexin-D1 activation to regulate axonal extension as neurons mature. To further confirm that low Mtss1 expression is critical for shortening neurite outgrowth, we overexpressed Mtss1 in *Plxnd1*-deficient MSNs. In comparison to the neurite length observed in GFP-overexpressing MSNs, ectopic Mtss1 overexpression rescued the growth reduction phenotype in the *Plxnd1*-knockout neurons (*Figure 3K and L*). These results suggest that the Sema3E-Plexin-D1 repulsive guidance cue is capable of regulating axonal growth through positive facilitator proteins such as Mtss1.

## The Mtss1 I-BAR domain binds to Plexin-D1, and this interaction is Sema3E-independent

Since both Plexin-D1 and Mtss1 regulate actin cytoskeletal rearrangement as a guidance molecule and membrane transformer, respectively, near the cell surface, we speculated that Plexin-D1 and Mtss1 might interact with each other via the BAR domain to induce actin-related cellular events. To test whether Plexin-D1 and Mtss1 can physically interact, we generated multiple deletion constructs of human Plexin-D1 and Mtss1 (*Figure 4A*). When we overexpressed full-length Plexin-D1 and Mtss1 together in HEK293T cells, both proteins were successfully pulled down together (*Figure 4B*). However, Plexin-D1 with the intracellular domain (ICD) deleted failed to bind Mtss1, indicating an intracellular Plexin-D1 and Mtss1 interaction (*Figure 4C*). When we overexpressed full-length Plexin-D1 and each Mtss1 deletion construct, every Mtss1 construct containing the I-BAR domain coprecipitated with Plexin-D1, but I-BAR-deficient Mtss1 failed to interact with Plexin-D1 (*Figure 4D*). In a previous study, it was determined that Plexin-D1 interacts with SH3-domain binding protein 1 (SH3BP1), another protein of the BAR domain family (*Tata et al., 2014*), thus we investigated whether Plexin-D1 also binds to other BAR domain-containing proteins in general. As previously reported, we found that overexpression of Plexin-D1 in HEK293T cells resulted in the formation of a complex with overexpressed SH3BP1 (*Figure 4—figure supplement 1A*). However, it did not form a complex with srGAP2, a protein containing an F-BAR domain, or IRSP53, another protein containing an I-BAR domain like Mtss1 (*Figure 4—figure supplement 1B and C*). Conversely, we also examined whether Mtss1 can interact with other members of the Plexin family. As shown in *Figure 4—figure supplement 1D and E*, overexpressed Mtss1 was unable to form a complex with Plexin-B2 or -B3 proteins. These findings suggest that the formation of the Plexin-D1-Mtss1 complex is relatively specific.

Since Sema3E binding to Plexin-D1 caused SH3BP1 release from the complex (*Tata et al., 2014*), we examined whether Sema3E binding influences Mtss1 dissociation from Plexin-D1. In contrast to the effect on SH3BP1, Sema3E treatment did not interfere with Plexin-D1-Mtss1 complex formation, indicating that the complex is formed in a Sema3E-independent manner (*Figure 4E and F*). Moreover, to determine whether Plexin-D1 and Mtss1 can bind directly to each other, we performed an in vitro

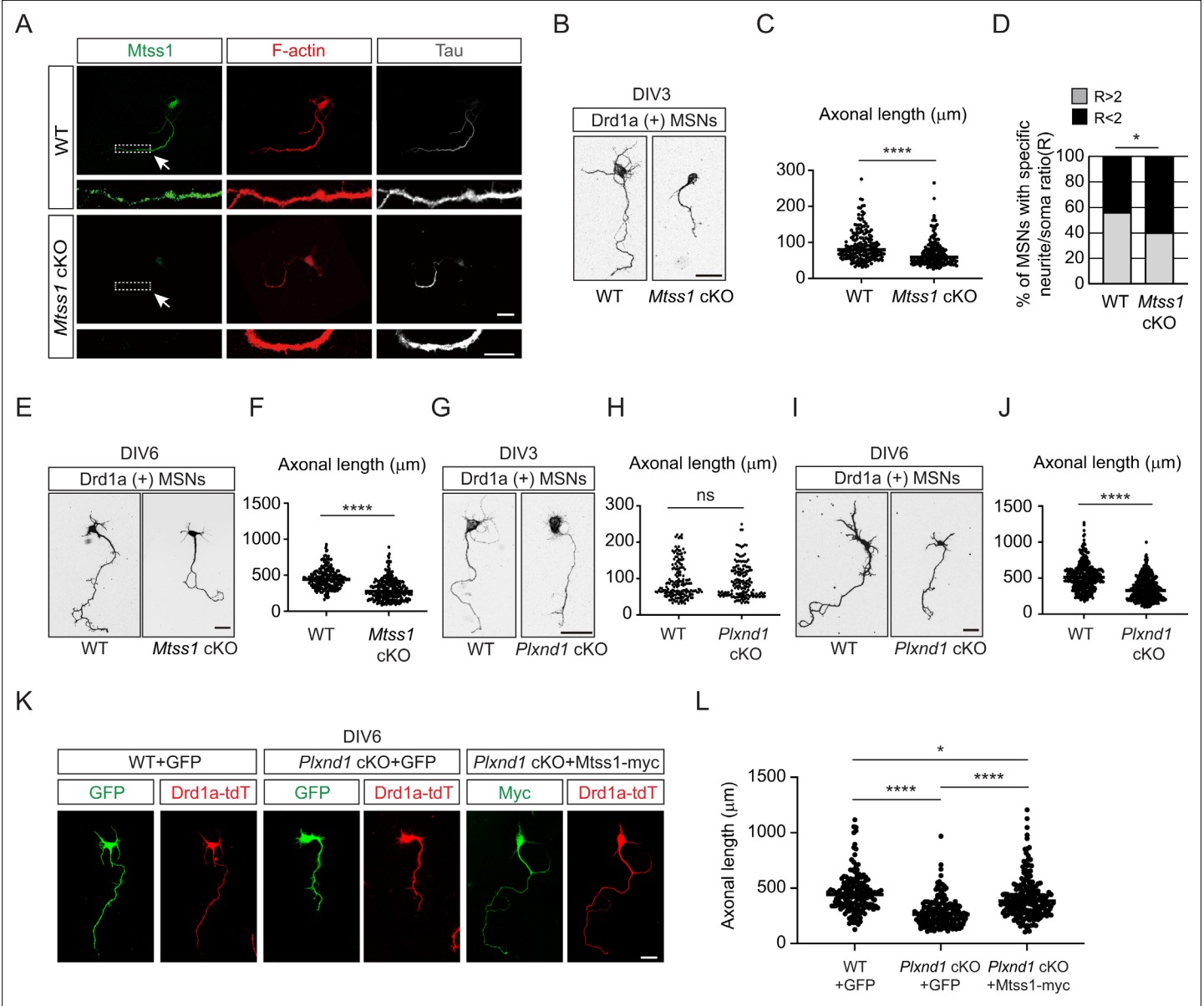

**Figure 3.** Mtss1 contributes to neurite extension of Drd1a-positive medium spiny neurons (MSNs) under the regulation of Sema3E-Plexin-D1 signaling. (**A**) Immunocytochemistry for Mtss1 (green), Tau (gray), and F-actin (red) in cultured MSNs at DIV3 obtained from wild-type (WT) or *Mtss1* conditional knockout (cKO) mice. White dotted boxes are shown in the inset image on the bottom. Scale bar, 10 µm. (**B**) Representative images of Drd1a+MSNs at DIV3 derived from WT (*Drd1a-tdT; Mtss1$^{f/f}$*) or *Mtss1* cKO (*Drd1a-tdT; Nes-cre; Mtss1$^{f/f}$*) mice. Scale bar, 20 µm. (**C**) Quantification of neurite length in (**B**) was performed as previously reported (*Chauvet et al., 2016*). The values represent the average ratio of the fold change in length compared to the control samples. Student's *t*-test; n = 179 for WT, n = 184 for *Mtss1* cKO, in three independent experiments. (**D**) Percentage of Drd1a-positive MSNs with neurites shorter than twice the cell body diameter in (**B, C**). $\chi^2$ test; n = 48 for WT, n = 48 for *Mtss1* cKO in four independent experiments. (**E**) Representative images of Drd1a+MSNs at DIV6 derived from WT (*Drd1a-tdT; Mtss1$^{f/f}$*) or *Mtss1* cKO (*Drd1a-tdT; Nes-cre; Mtss1$^{f/f}$*) mice. Scale bar, 50 µm. (**F**) Quantification of neurite length in (**E**). Student's *t*-test; n = 210 for WT, n = 260 for *Mtss1* cKO in three independent experiments. (**G**) Representative images of Drd1a+MSNs at DIV3 derived from WT (*Drd1a-tdT; Plxnd1$^{f/f}$*) or *Plxnd1* cKO (*Drd1a-tdT; Nes-cre; Plxnd1$^{f/f}$*) mice. Scale bar, 20 µm. (**H**) Quantification of neurite length in (**G**). Student's *t*-test; n = 167 for WT, n = 159 for *Plxnd1* cKO in three independent experiments. (**I**) Representative images of Drd1a+MSNs at DIV6 derived from WT or *Plxnd1* cKO mice. Scale bar, 50 µm. (**J**) Quantification of neurite length in (**I**). Student's *t*-test; n = 339 for WT, n = 403 for *Plxnd1* cKO in three independent experiments. (**K**) Representative images of GFP- or Mtss1-myc-transfected-Drd1a+MSNs at DIV6 derived from WT or *Plxnd1* cKO mice. Scale bar, 50 µm. (**L**) Quantification of neurite length in (**K**). Error bars, mean ± SEM; *p<0.05, ****p<0.0001 by one-way ANOVA with Tukey's post hoc correction for multiple comparisons; n = 163 for WT + GFP, n = 187 for *Plxnd1* cKO + GFP, n = 195 for *Plxnd1* cKO + Mtss1-myc in three independent experiments. Error bars, mean ± SEM; ns p>0.05, *p<0.05, ****p<0.0001 by indicated statistical tests.

*Figure 3 continued on next page*

Figure 3 continued

The online version of this article includes the following source data and figure supplement(s) for figure 3:

**Figure supplement 1.** Expression of Mtss1 induces I-BAR domain-dependent morphological changes in COS7 cells, generating protrusions.

**Figure supplement 1—source data 1.** Western blots shown in *Figure 3—figure supplement 1A*.

protein–protein binding assay using the purified ICD of Plexin-D1 and the I-BAR domain of Mtss1 proteins. As shown in *Figure 4G*, both proteins precipitated together, suggesting that Plexin-D1 and Mtss1 play a role as direct binding partners.

## Mtss1 transports Plexin-D1 to the growth cone in cultured direct-pathway MSNs

Because of the high probability of localization of the Plexin-D1-Mtss1 complex at the cell membrane, Mtss1 may regulate Plexin-D1 function at the cell surface. To investigate the role of Mtss1 in Plexin-D1 activity, we first examined whether Mtss1 affects the Plexin-D1 level on the plasma membrane by performing a surface molecule biotinylation analysis. We observed that overexpressed Plexin-D1 proteins in COS7 cells were efficiently biotinylated on the cell surface, but the Plexin-D1 protein level on the surface was not changed during Mtss1 coexpression (*Figure 5—figure supplement 1A and B*). Another potential mechanism by which Mtss1 may affect Plexin-D1 activity might be endocytic regulation because Plexin-D1 is rapidly endocytosed after Sema3E treatment (*Burk et al., 2017*). However, Mtss1 coexpression did not affect Sema3E-induced Plexin-D1 endocytosis (*Figure 5—figure supplement 1C and D*). Additionally, Mtss1 overexpression did not change the binding affinity of Sema3E for Plexin-D1 (*Figure 5—figure supplement 1E*). These data, including those regarding Sema3E-independent complex formation (*Figure 4E and F*), suggest that Mtss1 does not directly affect the functional role of Sema3E-Plexin-D1 at the cell-surface level.

We next hypothesized that Mtss1 forms a complex with Plexin-D1 that targets Plexin-D1 to filopodium-like structures since we observed that Mtss1 is mainly involved in protrusion formation and neurite outgrowth in cultured cells. To test this possibility, we coexpressed Plexin-D1 and Mtss1 and analyzed their localization in COS7 cells. As shown in *Figure 5—figure supplement 1F and G*, when Plexin-D1 and Mtss1 were coexpressed, both proteins were mostly present in F-actin-enriched protrusions. However, overexpression of Plexin-D1 lacking the ICD overlapped with Mtss1 to a lesser extent than overexpression of wild-type Plexin-D1, whereas Mtss1 was abundant in the protrusions. In addition, Mtss1 lacking the I-BAR domain, which possesses membrane-bending activity, did not generate filopodia-like protrusions in COS7 cells but was localized with F-actin, including in marginal areas, probably via the Mtss1 WH2 domain (*Mattila et al., 2003*). Interestingly, although Plexin-D1 was evenly distributed throughout a cell, most Plexin-D1 was not present with Mtss1 missing the I-BAR domain. In addition, the localization of Plexin-D1 and Mtss1 in the protrusions was also not disturbed in the presence of Sema3E (*Figure 5—figure supplement 1H*). These results suggest that Mtss1 not only induces cell protrusion formation but also contributes to Plexin-D1 localization to specific sites.

Next, to confirm that Mtss1 leads Plexin-D1 to protrusive structures, such as growth cones in cultured neurons, Mtss1 and Plexin-D1 localization was analyzed by transfecting *Mtss1*-deficient striatal neurons with expression constructs carrying both proteins. Both overexpressed Plexin-D1 and Mtss1 proteins seemed to be colocalized along growing neurites, but the Mtss1 mutant lacking the I-BAR domain showed reduced Plexin-D1 level as well as a low colocalization rate (*Figure 5A–C*). As shown in COS7 cells (*Figure 5—figure supplement 1F*), wild-type Mtss1 proteins were present with Plexin-D1 in F-actin-enriched regions of growth cones; in contrast, mutant Mtss1 was expressed at a lower level than wild-type Mtss1 and failed to colocalize with Plexin-D1, resulting in reduced Plexin-D1 localization in the growth cone (*Figure 5D–F*). However, the intensity of Plexin-D1 upon co-expression of wild-type Mtss1 or mutant Mtss1 remained unchanged in the cell body (*Figure 5—figure supplement 2A and B*). It is plausible that overexpressed proteins accumulate in the soma, and a limited proportion of them are destined to the axon terminal on demand (*Droz et al., 1973*). To explore the role of Mtss1 in transporting Plexin-D1, we directly monitored the movement of Plexin-D1 along neurites in real time by imaging Drd1a-positive MSNs expressing Plexin-D1-GFP fusion proteins in wild-type and *Mtss1*-knockout neurons at DIV6. Our live-cell imaging showed that the majority of Plexin-D1-containing vesicles were dynamically transported in both proximal and distal

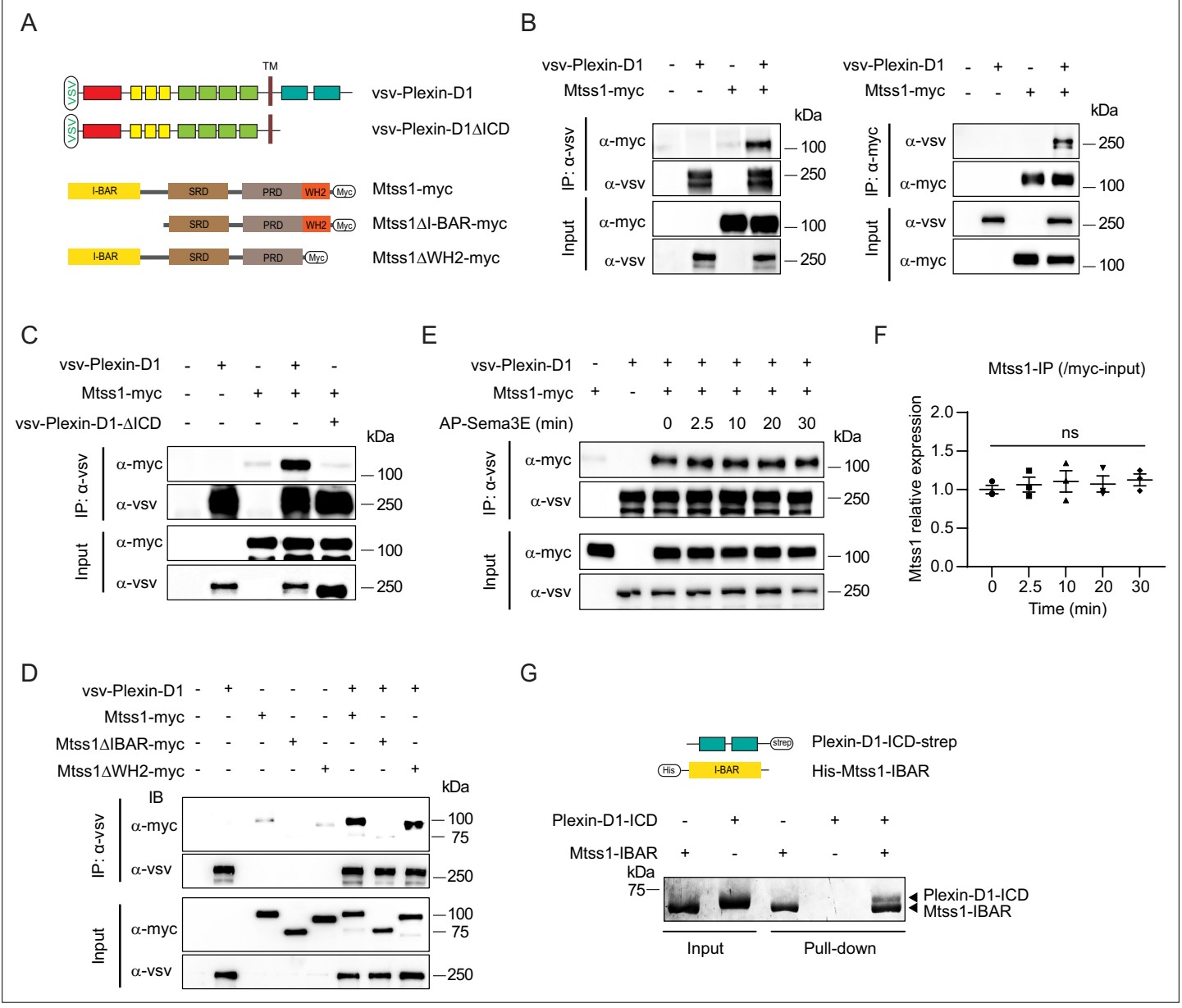

**Figure 4.** The Mtss1 I-BAR domain directly binds to Plexin-D1, independent of Sema3E. (**A**) Schematics depicting full-length constructs of Mtss1 and its truncation mutants. (**B**) Coimmunoprecipitation and immunoblot analysis of HEK293T cells transfected with Mtss1-myc with vsv-Plexin-D1. The interaction between Mtss1 and Plexin-D1 was investigated by immunoprecipitation with anti-vsv (left) or anti-myc (right) antibodies and subsequent western blotting with reciprocal antibodies. (**C**) Immunoprecipitation and western blot analysis after Plexin-D1 and Mtss1 overexpression. The vsv-Plexin-D1ΔICD did not bind to Mtss1-myc. (**D**) Immunoprecipitation and western blot assays to identify the binding domain in Mtss1 that interacts with Plexin-D1. (**E**) The interaction between Mtss1 and Plexin-D1 was assessed over time following treatment with Sema3E (2 nM) and was not affected by AP-Sema3E treatment. (**F**) Graph quantifying the band intensity in (**E**). Error bars, mean ± SEM; ns p>0.05 by two-way ANOVA with Bonferroni's post hoc correction for multiple comparisons; n = 3 in three independent experiments. (**G**) Pull-down assay and visualization of the protein bands using Coomassie staining. His-Mtss1-IBAR binds directly to Plexin-D1-ICD-strep.

The online version of this article includes the following source data and figure supplement(s) for figure 4:

**Source data 1.** Western blots and gel shown in *Figure 4B, C, D, E, and G*.

**Figure supplement 1.** The interaction between BAR domain-containing proteins and Plexin-D1, or between Mtss1 and Plexin family proteins.

**Figure supplement 1—source data 1.** Western blots shown in *Figure 4—figure supplement 1A–E*.

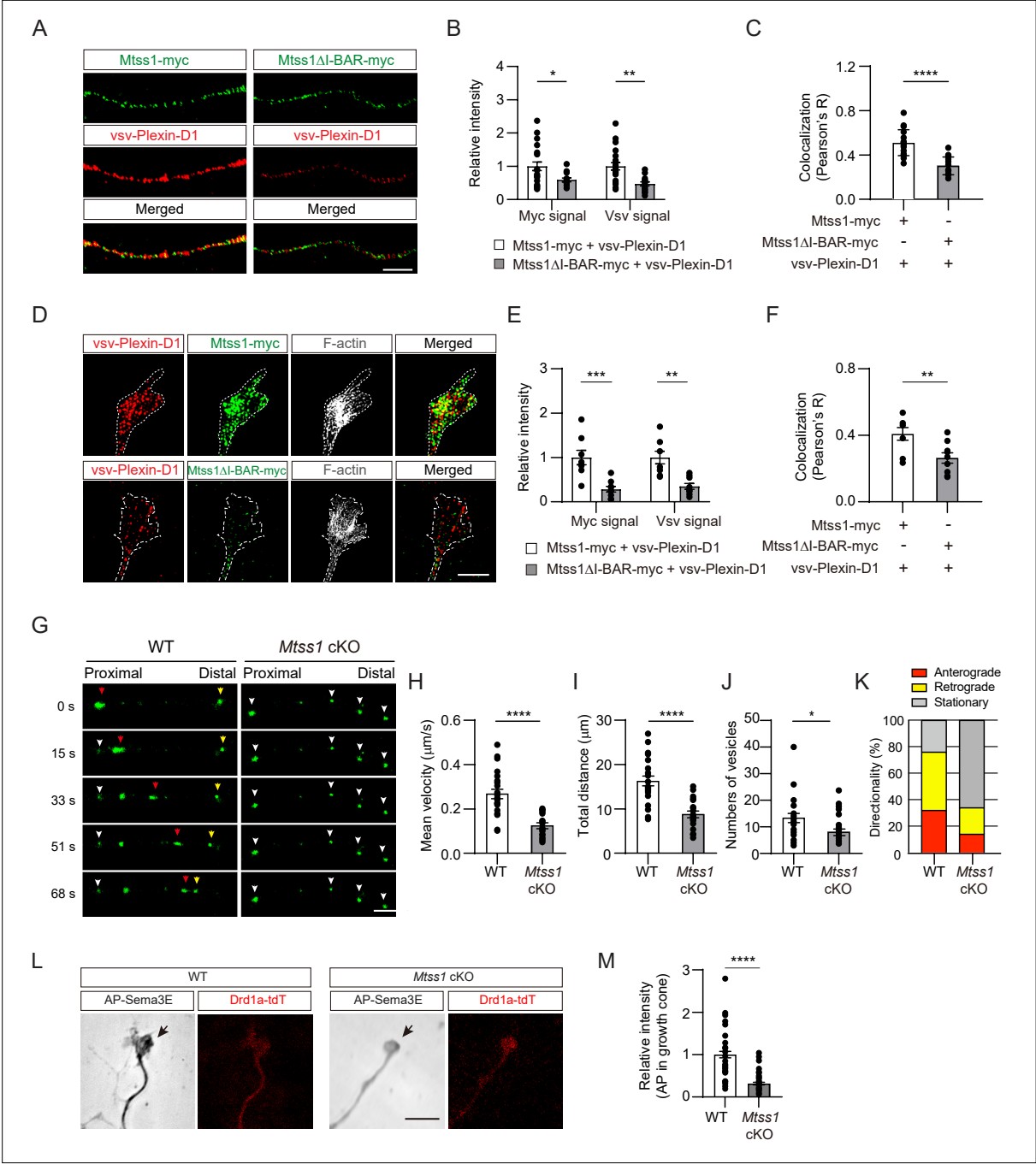

**Figure 5.** Mtss1 facilitates Plexin-D1 transport to the growth cone in cultured Drd1a-positive medium spiny neurons (MSNs). (**A**) Immunocytochemistry for Mtss1-myc or Mtss1ΔI-BAR -myc (green), vsv-Plexin-D1 (red) in the axons of MSNs transfected with vsv-Plexin-D1 and Mtss1-myc or Mtss1ΔI-BAR-myc, using *Mtss1*-null mice as a background. The images were acquired using structured illumination microscopy (N-SIM). Scale bar, 5 µm. (**B**) Quantification of the fluorescence intensity in the axons of (**A**). Two-way ANOVA with Tukey's post hoc correction for multiple comparisons; vsv-Plexin-D1+Mtss1-myc, n = 21, and vsv-Plexin-D1+Mtss1ΔI-BAR-myc, n = 14. (**C**) Quantification of colocalization by Pearson's correlation coefficient calculated using Costes' randomized pixel scrambled image method. Student's *t*-test; vsv-Plexin-D1+Mtss1-myc, n = 21, and vsv-Plexin-D1+Mtss1ΔI-BAR-myc, n = 14. (**D**) Immunocytochemistry for vsv-Plexin-D1 (red), Mtss1-myc (green), and F-actin (gray) in the growth cones of MSNs transfected with vsv-Plexin-D1 and Mtss1-myc or Mtss1ΔI-BAR-myc originating from *Mtss1*-null mice. Scale bar, 5 µm. (**E**) Quantification of the intensities in the growth cones. The values represent the average fold change in expression compared to the control samples (vsv-Plexin-D1+Mtss1-myc). Two-way ANOVA with Tukey's post hoc correction for multiple comparisons; vsv-Plexin-D1+Mtss1-myc, n = 8, and vsv-Plexin-D1+Mtss1ΔI-BAR-myc, n = 9. (**F**) Quantification of colocalization by Pearson's correlation coefficient calculated using Costes' randomized pixel scrambled image method. Student's *t*-test; vsv-Plexin-D1+Mtss1-myc, n = 8, and vsv-Plexin-D1+Mtss1ΔI-BAR-myc, n = 9. (**G**) Representative time-lapse images of Plexin-D1-GFP-positive vesicles (green) from

*Figure 5 continued on next page*

*Figure 5 continued*

wild-type or *Mtss1* conditional knockout (cKO) MSNs at DIV6. Plexin-D1-positive vesicles transported toward distal or proximal directions are indicated by red and yellow arrowheads, respectively. White arrowheads indicate stationary vesicles. A mean velocity of less than 0.1 μm/s was considered to be a stationary condition. See *Videos 1–4*. Scale bar, 5 μm. (**H–J**) Quantification of mean velocity (**H**), total travel distance (**I**), and number of vesicles (**J**) along neurites of Plexin-D1-positive vesicles. Student's *t*-test; n = 23 for WT, n = 25 for *Mtss1* cKO in four independent experiments. (**K**) Distribution analysis of vesicle directionality. (**L**) The AP-Sema3E binding assay was performed to visualize Plexin-D1 protein (black arrows) in the growth cones of WT or *Plxnd1*-deficient MSNs. Scale bar, 10 μm. (**L**) The AP-Sema3E binding assay performed to visualize Plexin-D1 protein (black arrows) in the growth cones of WT or *Mtss1*-deficient MSNs. Localization of Plexin-D1 (black arrows) in the growth cone investigated in cultured MSNs from WT (*Drd1a-tdT; Mtss1^{f/f}*) or *Mtss1* cKO (*Drd1a-tdT; Nes-cre; Mtss1^{f/f}*) mice. (**M**) Quantification of Plexin-D1 intensity shown in (**L**). Mann–Whitney test; WT n = 51, KO n = 50. Error bars in all graphs, mean ± SEM; *p<0.05, **p<0.01, ***p<0.001. ****p<0.0001 by indicated statistical tests.

The online version of this article includes the following source data and figure supplement(s) for figure 5:

**Figure supplement 1.** Mtss1 expression alters Plexin-D1 localization to the protrusion structure in COS7 cells without affecting its endocytosis or Sema3E binding.

**Figure supplement 1—source data 1.** Western blots shown in *Figure 5—figure supplement 1A and C*.

**Figure supplement 2.** No significant alteration in the expression of vsv-Plexin-D1 or Mtss1-myc or Mtss1ΔI-BAR-myc in the medium spiny neuron (MSN) soma.

directions in wild-type MSNs. In contrast, the Plexin-D1-positive vesicles in *Mtss1*-deficient MSNs remained stationary, and their overall numbers in neurites were also slightly reduced (*Figure 5G–K* and *Videos 1–4*). These observations demonstrate that Mtss1 facilitates the dynamic transportation of Plexin-D1 along the growing neurites of direct-pathway MSNs, leading to an increased rate of Plexin-D1 localization in the growth cones.

Moreover, to examine whether endogenous Mtss1 expression affects Plexin-D1 localization in the growth cone, leading to an active guidance role, we performed an AP-Sema3E binding analysis in Drd1a-positive MSNs at DIV6 (*Gu et al., 2005*). The AP-Sema3E binding assay has been used in many previous studies as an alternative method to detect endogenous Plexin-D1 protein due to the lack of reliable anti-Plexin-D1 antibodies (*Bellon et al., 2010*). The *Mtss1*-knockout neurons exhibited a low level of AP-Sema3E binding in the growth cones, indicating that the endogenous trafficking of Plexin-D1 to the tip of the growing axon is disrupted in the absence of Mtss1 (*Figure 5L and M*). These results suggest that Mtss1 serves to deliver Plexin-D1 to the growth cone.

## Plexin-D1 trafficking to the growth cone by Mtss1 potentiates the repulsive response to Sema3E

Because Mtss1 facilitates Plexin-D1 transport to the growth cone, proper Plexin-D1 localization at the membrane may contribute to the triggering of repulsive signaling by Plexin-D1 in response to Sema3E. To test this hypothesis, we performed a growth cone collapse assay at DIV3 in direct-pathway MSNs and found that wild-type Drd1a-tdT-positive striatal neurons underwent a high collapse rate after exogenous Sema3E treatment, whereas the growth cones of neurons lacking *Mtss1* did not collapse at a significantly different rate (*Figure 6A and B*). However, reintroduction of wild-type Mtss1 into *Mtss1*-knockout MSNs resulted in growth cone collapse, but overexpressed Mtss1 lacking the I-BAR domain

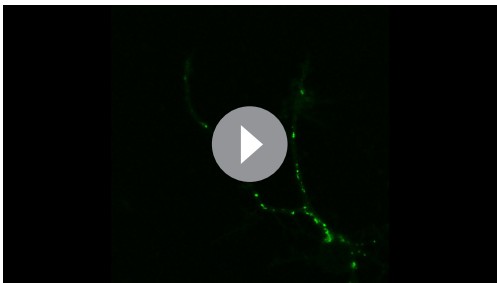

**Video 1.** Time-lapse live imaging of Plexin-D1-GFP-positive vesicles (green) from wild-type medium spiny neurons (MSNs) at DIV6_1.

https://elifesciences.org/articles/96891/figures#video1

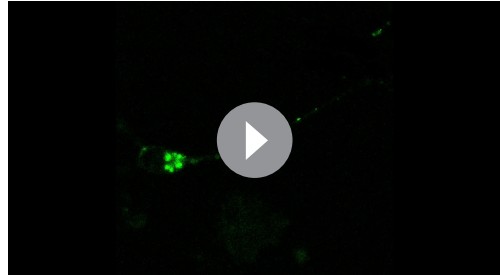

**Video 2.** Time-lapse live imaging of Plexin-D1-GFP-positive vesicles (green) from wild-type medium spiny neurons (MSNs) at DIV6_2.

https://elifesciences.org/articles/96891/figures#video2

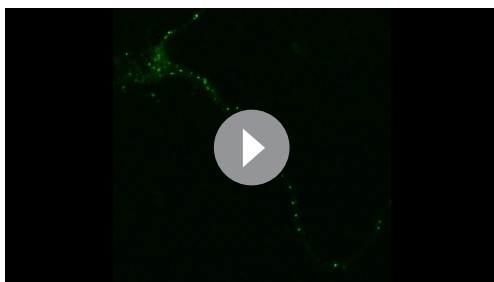

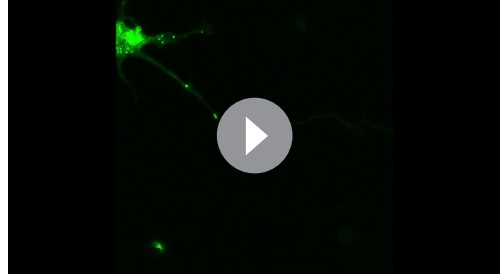

**Video 3.** Time-lapse live imaging of Plexin-D1-GFP-positive vesicles (green) from Mtss1 conditional knockout (cKO) medium spiny neurons (MSNs) at DIV6_1.
https://elifesciences.org/articles/96891/figures#video3

**Video 4.** Time-lapse live imaging of Plexin-D1-GFP-positive vesicles (green) from Mtss1 conditional knockout (cKO) medium spiny neurons (MSNs) at DIV6_2.
https://elifesciences.org/articles/96891/figures#video4

showed less response to Sema3E (*Figure 6C and D*). In addition, *Plxnd1*-null MSNs also showed a very low collapse rate regardless of Sema3E addition at DIV6 (*Figure 6E*), consistent with a previous report that Sema3E-Plexin-D1 signaling acts as a repulsive guidance cue (*Chauvet et al., 2007*). The reduced collapse phenotypes were rescued by overexpression of full-length Plexin-D1 but not by ICD-deleted Plexin-D1 (*Figure 6F*). In summary, Mtss1 targeting of Plexin-D1 to the growth cone is critical for robust Sema3E-induced repulsive signaling.

## Absence of Mtss1 reduces projection density and Plexin-D1 localization in the striatonigral pathway

First, to determine whether Mtss1 is indeed expressed in the projecting striatonigral axons and their destination, the SNr, we performed immunostaining using wild-type or *Mtss1*-knockout mouse brains. We observed Mtss1 localization in the striatonigral tract and SNr, but the immunostaining signal disappeared in *Mtss1*-deficient mice (*Figure 7—figure supplement 1*). Although we also detected Mtss1 in the SNr region, we could not rule out the possibility that substantia nigra neurons express Mtss1 at this stage. Next, to investigate the role of Mtss1 in striatonigral pathway development in vivo, we performed an AP-Sema3E binding assay to examine the Plexin-D1-positive tract in brain tissue (*Chauvet et al., 2007*). In mice expressing wild-type *Mtss1*, a significant amount of Plexin-D1 was observed in the neuronal tract reaching the substantia nigra. In contrast, *Mtss1*-knockout mice exhibited a relatively small area of Plexin-D1-positive striatonigral tracts, including both poor neuronal projection and reduced Plexin-D1 localization at E17.5 (*Figure 7A–D*). In addition, despite these reduced neuronal projections in the *Mtss1*-knockout mice, the density of the AP-Sema3E-positive tracts was reduced even more (*Figure 7E*), indicating that the absence of Mtss1 prevents both normal axonal projections and Plexin-D1 trafficking. In the coronal view, the bundle density of Plexin-D1-positive projections passing between the rTh and Gp was reduced in the *Mtss1*-knockout mice (*Figure 7F*). Because most mice with conditional Nestin-Cre-driven *Mtss1* deletion were born alive, we analyzed the Plexin-D1-positive striatonigral pathway at P5. Consistent with the results obtained with E17.5 embryos, *Mtss1*-deficient neonates showed fewer Plexin-D1-positive striatonigral projections in the coverage area and a reduced path width (*Figure 7G–J*).

To further identify Plexin-D1-positive striatonigral pathway defects, we crossed Drd1a-tdT transgenic reporter mice with conditional *Mtss1*-knockout mice and visualized the pathway in the offspring. Consistent with the results shown in AP-Sema3E binding experiments, the total boundary area with Drd1a-tdT-positive projections was smaller and less compact in the *Mtss1*-knockout mice than in the wild-type mice at P5 (*Figure 7K–M*). At P30, the projection density defects were more obvious, but the boundary area in the wild-type and mutant mice was not significantly different, indicating that *Mtss1* deficiency led to the formation of fewer striatonigral axonal bundles (*Figure 7—figure supplement 2A–C*). We assumed that Mtss1 regulates the initial striatonigral axonal projection during development of the neonate and that the pathway establishment ends by P7 (*Morello et al., 2015*) therefore, the scarcity of the projections may be clearer when the brain increases to the adult size. Moreover, we examined whether Mtss1 expression affects Plexin-D1 levels on the Drd1a-tdT-positive tracks at P5 using the same methodology as employed at E17.5 (*Figure 7A*). Our findings revealed a decreased

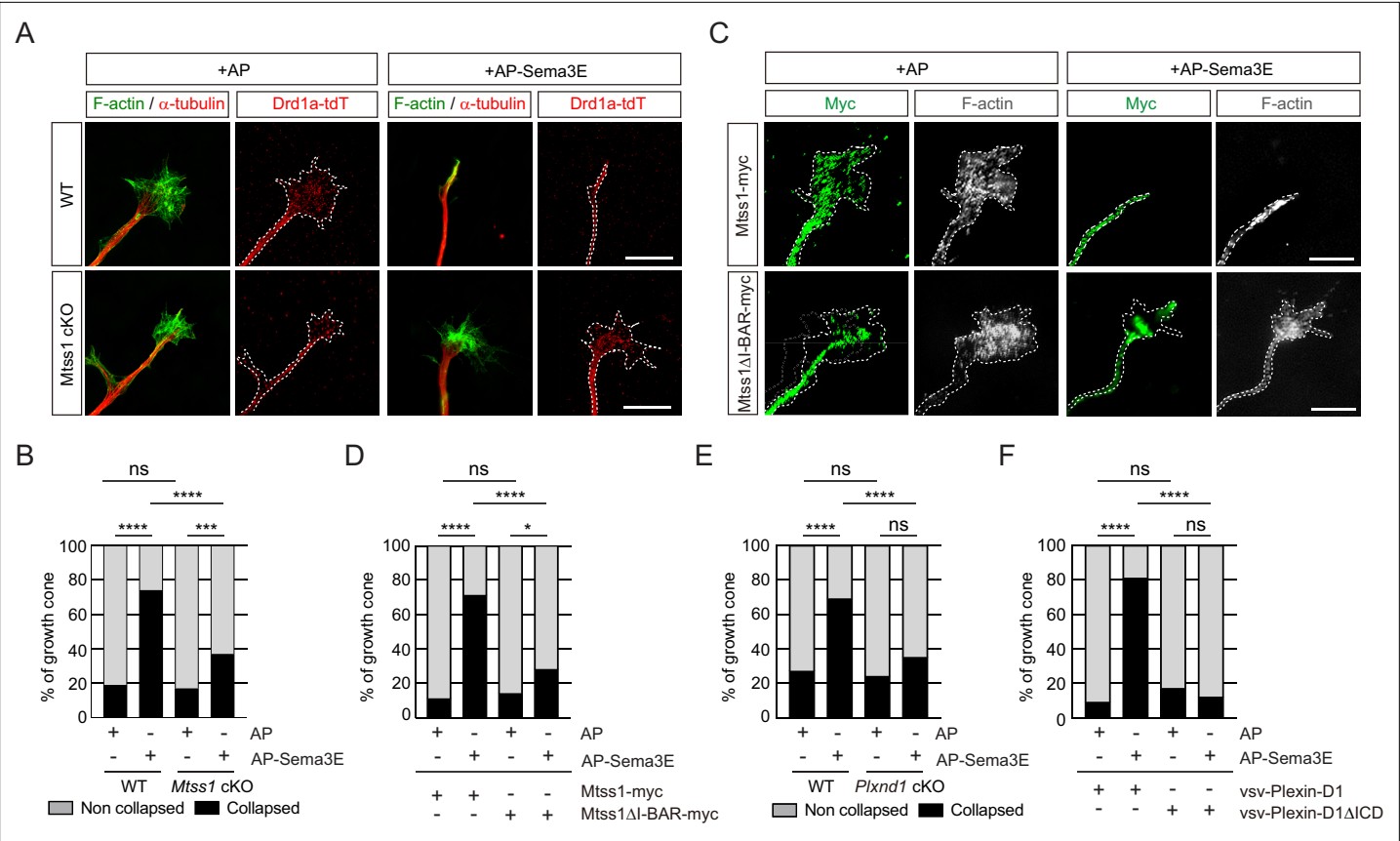

**Figure 6.** The repulsive response through Sema3E-Plexin-D1 signaling is attenuated in the absence of Mtss1. (**A**) A growth cone collapse assay in the presence or absence of Sema3E (2 nM) was performed with medium spiny neuron (MSN) cultures derived from wild-type (WT) (*Drd1a-tdT; Mtss1^{f/f}*) or *Mtss1*-KO (*Drd1a-tdT; Nes-cre; Mtss1^{f/f}*) mice at DIV3. Scale bar, 10 μm. The images were obtained from structured illumination microscopy (N-SIM). (**B**) Quantification of collapsed growth cones in (**A**). Error bars, mean ± SEM; ***p<0.001, ****p<0.0001 by $\chi^2$ test; WT + AP, n = 155, WT + AP-Sema3E, n = 163, KO + AP, n = 149, KO + AP-Sema3E, n = 149. (**C**) A growth cone collapse assay in the presence or absence of Sema3E (2 nM) was performed with MSN cultures following ectopic expression of Mtss1-myc or Mtss1Δ I – B A R-myc in the *Mtss1*-null background at DIV3. Scale bar, 10 μm. (**D**) Quantification of collapsed growth cones in (**C**). Error bars, mean ± SEM; ***p<0.001, ****p<0.0001 by $\chi^2$ test; Mtss1-myc+AP, n = 28, Mtss1-myc+AP-Sema3E, n = 28, Mtss1ΔI-BAR-myc+AP, n = 28, Mtss1Δ I – B A R-myc+AP-Sema3E, n = 43. (**E**) Quantification of the collapse assay in the presence or absence of Sema3E (2 nM) was performed with MSN cultures at DIV6 from WT (*Drd1a-tdT; Plxnd1^{f/f}*) or *Plxnd1*-KO (*Drd1a-tdT; Nes-cre; Plxnd1^{f/f}*) mice. ****p<0.0001 by $\chi^2$ test; WT + AP, n = 44, WT + AP-Sema3E, n = 45, KO + AP, n = 45, KO + AP-Sema3E, n = 46. in three independent experiments. (**F**) Quantification of a growth cone collapse assay in the presence or absence of Sema3E (2 nM) was performed with MSN cultures following ectopic expression of vsv-Plexin-D1 or vsv-Plexin-D1ΔICD in the *Plxnd1*-KO background at DIV6. ****p<0.0001 by $\chi^2$ test; vsv-Plexin-D1+AP, n = 35, vsv-Plexin-D1+AP-Sema3E, n = 32, vsv-Plexin-D1ΔICD + AP, n = 30, vsv-Plexin-D1ΔICD + AP-Sema3E, n = 26 in three independent experiments. tdT, tdTomato.

ratio of AP-Sema3E binding per tdT staining, suggesting that Mtss1 is implicated not only in axonal projections but also in Plexin-D1 trafficking in navigating striatonigral projections (*Figure 7N–Q*).

We then investigated whether Mtss1 specifically contributes to the development of descending striatonigral projections rather than dendritic arborization. Since global *Mtss1*-mutant mice have enlarged brain ventricles and decreased cortical thickness (*Minkeviciene et al., 2019*), we first examined whether the axonal projection in *Mtss1*-knockout mice is due to cellular death of MSNs in the striatum. Using cleaved caspase 3 staining to detect apoptotic cells, we found few dying cells in the wild-type and *Mtss1*-mutant neonates (*Figure 7—figure supplement 3A and B*), indicating that no significant cell pathology was induced by *Mtss1* expression deficiency. Because Mtss1 is selectively expressed in direct-pathway MSNs, which comprise approximately 45% of striatal neurons (*Figure 1*), we expected that Golgi staining would be sufficient to detect any dendritic defects, such as aberrant number and/or length of branches. However, we found no detectable difference between the wild-type and *Mtss1* mutants at P5 (*Figure 7—figure supplement 3C–E*). In addition, no significant

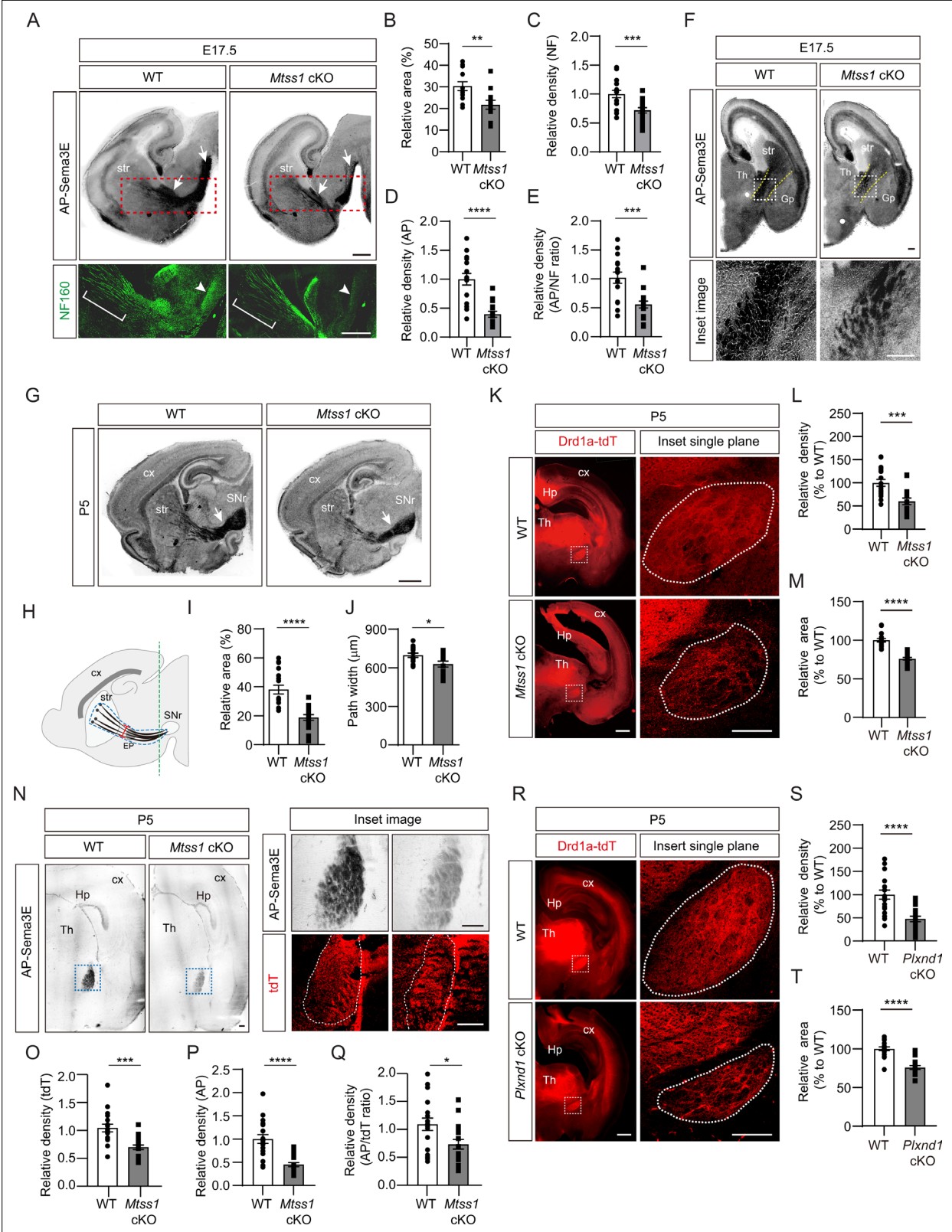

**Figure 7.** A reduced number of Plexin-D1 molecules localize to the developing striatonigral projections in *Mtss1*-deficient mice. (**A**) AP-Sema3E binding assay (top) to detect Plexin-D1 expression (white arrows) in striatonigral projections and immunohistochemistry (bottom) for neurofilaments (NFs) indicated by red dotted square, performed in adjacent parasagittal sections at E17.5 of wild-type (WT) or *Mtss1* conditional knockout (cKO) mice. The diminished projections are marked by white brackets (middle of the striatum) and arrowheads (near substantia nigra regions). Scale bar,

*Figure 7 continued on next page*

*Figure 7 continued*

500 µm. (**B**) Quantification of the Plexin-D1-positive area in the total striatonigral projection at E17.5 of WT or *Mtss1* cKO mice. Student's *t*-test; WT, n = 12, KO, n = 12 (three sections/mouse). (**C**) Quantification of the fluorescence density (intensity/area) of NF in striatonigral projections at E17.5 in WT or *Mtss1* cKO mice. Student's *t*-test; WT, n = 16, KO, n = 17 (three or four sections/mouse). (**D**) Quantification of the AP density (intensity/area) in striatonigral projections at E17.5 in WT or *Mtss1* cKO mice. Student's *t*-test; WT, n = 16, KO, n = 17 (three or four sections/mouse). (**E**) The ratio of AP to NF density (intensity/area). Student's *t*-test; WT, n = 16, KO, n = 17 (three or four sections/mouse). (**F**) Coronal view of Plexin-D1 localization in striatonigral projections at E17.5 in WT or *Mtss1* cKO mice. Yellow dotted lines indicate the corridor between the thalamus and globus pallidus. Insets show the images in dotted boxes at higher resolution. Scale bar, 200 µm. (**G**) Representative images of Plexin-D1 molecules in striatonigral projections visualized by AP-Sema3E binding assay in WT or *Mtss1* cKO mice at P5. White arrows indicate striatonigral projections. Scale bar, 1 mm. (**H**) Schematic representing the quantified region. The dotted blue lines indicate the striatonigral projection-covering areas. The width (red segment) of the striatonigral tract was measured as previously described (*Burk et al., 2017*). (**I, J**) Quantification of the Plexin-D1-positive area (%) in dotted blue area and projection width at P5 according to the scheme shown in (**H**). Mann–Whitney test (**I**) and Student's *t*-test (**J**). n = 18 (three sections/mouse). (**K**) Immunohistochemistry of coronal sections of striatal projections labeled with td-Tomato endogenously expressed through the *Drd1a* promotor in WT (*Drd1a-tdT; Mtss1^{f/f}*) or *Mtss1* cKO mice (*Drd1a-tdT; Nes-cre; Mtss1^{f/f}*) at P5. The white dotted boxes on the left images are shown in the inset images on the right, which were captured in a single plane using a high-resolution confocal microscope. Scale bar, 500 µm. (**L, M**) Quantification of the density (intensity/area) (**L**) and area size (**M**) of the striatonigral projection in the dotted region in the inset images. Mann–Whitney test (**L**) and Student's *t*-test (**M**); n = 18 per group (six sections/mouse). (**N**) AP-Sema3E binding assay and tdT immunostaining in adjacent sections of WT or *Mtss1* cKO mice located near the SNr. The green line in (**H**) indicates the location for cross-sectioning. The inset images of Plexin-D1-positive striatonigral projections were taken from the blue dotted boxes on the left panels. The dotted white lines indicate the tdT-positive striatonigral projections. Scale bar, 200 µm. (**O, P**) Quantification of the density (intensity/area) of tdT (**O**) and AP (**P**) of striatonigral projections WT or *Mtss1* cKO mice. (**Q**) The ratio of AP to tdT density (intensity/area). Student's *t*-test; WT, n = 18, KO, n = 18 (six sections/mouse). (**R**) Immunohistochemistry of coronal views of striatonigral projections in WT (*Drd1a-tdT; Plxnd1^{f/f}*) or *Plxnd1* cKO mice (*Drd1a-tdT; Nes-cre; Plxnd1^{f/f}*) at P5. The white dotted boxes on the left images are shown in the inset images on the right, which were obtained using a high-resolution confocal microscope in a single plane. Scale bar, 500 µm. (**S, T**) Quantification of the density (intensity/area) of tdT (**S**) and area size (**T**). Student's *t*-test; n = 18 per group (six sections/mouse). Error bars, mean ± SEM; *p<0.05, **p<0.01, ***p<0.001. ****p<0.0001 by indicated statistical tests. str, striatum; cx, cortex; Th, thalamus; SNr, substantia nigra; EP, entopeduncular nucleus; Gp, globus pallidus; Hp, hippocampus.

The online version of this article includes the following source data and figure supplement(s) for figure 7:

**Figure supplement 1.** Expression of Mtss1 in the striatonigral tract and SNr at P5.

**Figure supplement 2.** Mtss1 or Plexin-D1 deficiency reduced striatonigral axonal bundles without altering striatonigral projection patterns at P30.

**Figure supplement 3.** The absence of Mtss1 does not affect medium spiny neuron (MSN) survival, dendritic arborization, and Plexin-D1 expression during striatonigral pathway development.

**Figure supplement 3—source data 1.** Raw uncropped western blot & gel images.

**Figure supplement 4.** Striatonigral projection defects are observed in Sema3e-null mice at P5.

change in Plexin-D1 levels was observed in the striatum or cultured neurons of *Mtss1*-deficient mice compared to those in littermate controls (*Figure 7—figure supplement 3F–I*). These results indicate that the weakening of the Plexin-D1-positive striatonigral pathway in *Mtss1*-knockout mice is caused by both impairments of neuronal projections and inappropriate Plexin-D1 distribution due to Mtss1 downregulation.

## The absence of Plexin-D1 or Sema3E reduces the axonal projection of direct-pathway MSNs

Since Mtss1 expression is under the control of Sema3E-Plexin-D1 signaling, we investigated whether *Plxnd1* deletion leads to phenocopying of the striatonigral projection defects observed in *Mtss1*-knockout mice. During the neonatal period, the boundary area and compactness of descending striatonigral projections were small and loose in the *Plxnd1*-knockout mice, but only a reduced projection density was observed in adult mice, which was similar to that of *Mtss1*-knockout mice (*Figure 7R–T*, *Figure 7—figure supplement 2D–F*). Moreover, *Sema3e*-knockout at P5 also showed these projection defects, albeit somewhat milder compared to those observed in *Plxnd1*- or *Mtss1*-deficient mice (*Figure 7—figure supplement 4A–E*). These phenocopy results suggest that activation of Sema3E-Plexin-D1 signaling, which leads to induction of Mtss1, is required for the striatonigral trajectory during the axonal pathfinding period.

## Absence of Mtss1 or Plexin-D1 results in irregular projection patterns in direct-pathway MSNs

Subsequently, we examined whether striatonigral projection patterns are altered by Mtss1 deficiency. Interestingly, we observed that the descending striatonigral projections in the wild-type mice were relatively straight and untangled near the Gp region through the sagittal view (*Figure 8A*). When visualizing the striatonigral projection using 3D imaging by sparsely labeling with DiI injection at the dorsal striatum, the axonal bundles appeared relatively fasciculated and straight in the wild-type at P5 (*Figure 8B and C*). However, *Mtss1* mutants showed irregular projection patterns with random directionality and mild defasciculation (*Figure 8A–C*). To assess axonal deviation, we measured the number of projections intersecting each other in the same region, as shown in *Figure 8A and B*. The deficiency of Mtss1 led to more deviant ectopic projections (*Figure 8E*). However, the misguidance defects were not apparent at P30, likely because abnormal projections were discarded as mice matured into adults (*Figure 7—figure supplement 2G*). These results suggest that inefficient Plexin-D1 trafficking to the extending axons in the *Mtss1*-knockout mice may have weakened the proper guidance response.

Previous studies have reported that Sema3E-Plexin-D1 signaling defects lead to ectopic projection during development or misguidance in the adult striatonigral pathway (*Chauvet et al., 2007*; *Ehrman et al., 2013*), but these phenotypes were not detected in our study. Instead, we also observed irregular projection patterns near the Gp in *Plxnd1*-knockout mice at P5 but not at P30 (*Figure 8D and F*, *Figure 7—figure supplement 2H*), similar to those seen in *Mtss1*-knockout mice. Moreover, *Sema3e*-knockout at P5 showed a few axonal bundles deviated from their typical trajectories (*Figure 7—figure supplement 4F and G*). These results demonstrate that Sema3E-Plexin-D1 signaling, probably in concert with Mtss1 molecules, is specifically involved in the proper guidance of descending striatonigral projections. Collectively, our results confirm that Sema3E-Plexin-D1 signaling activates Mtss1 action, through which striatonigral neurons are extended and steered through the proper route to the target destination (*Figure 8J*).

Finally, since Sema3E-Plexin-D1 is well known to regulate both neural and vascular development (*Oh and Gu, 2013a*), we wondered whether Mtss1 also plays a role in the vasculature as a downstream player of the common guidance cue. Although it was expressed in neuronal cells, Mtss1 was not detected in vascular endothelial cells expressing Plexin-D1 during development (*Figure 8—figure supplement 1A*). In addition, *Mtss1*-knockout mice did not result in an obvious intersomatic vasculature defect, which is a typical phenotype observed in *Sema3e* or *Plxnd1* mutants (*Burk et al., 2017*; *Gu et al., 2005*; *Figure 8—figure supplement 1B*). Interestingly, Mtss1 was highly expressed in the two different types of cultured endothelial cells but was not controlled by Sema3E-Plexin-D1 signaling (*Figure 8—figure supplement 1C*). These results provide evidence that the activation of Mtss1 by the Sema3E-Plexin-D1 signaling pathway and its function in neurons could be selective and distinct.

## Discussion

There are two main aspects of the traditional axon guidance concept: attractive cue-guided axon growth and repulsive cue-guided axon growth. In these processes, the axon terminals are constantly facing both types of signals en route to their destination. In the present study, we show that repulsive guidance cues, namely, Sema3E-Plexin-D1 pairs, induce a dual-functioning facilitator, Mtss1, through which navigating axons ensure incessant extension to their target tissues while exhibiting sensitivity and subsequent steering in response to repulsive signals. In our model (*Figure 8—figure supplement 2*), Plexin-D1 on the cell body of direct-pathway MSNs in the striatum receives its specific ligand, Sema3E, through the thalamostriatal projection at the late embryonic stage and induces a unique pool of regulatory factors, including Mtss1. Mtss1 then generates axonal projections to targets in the substantia nigra while enabling Plexin-D1 to transport along axons. In the growth cones, Mtss1 promotes membrane curvature to form a protruding filopodium, and Plexin-D1 is positioned on the cell surface to sense the external repulsive guidance signals from Sema3E. Currently, it is not clear whether Plexin-D1 and Mtss1 are present as a complex at the membrane surface of the growth cone, but Mtss1 does not seem to directly affect the repulsive signaling activation of Sema3E-Plexin-D1. In this way, Mtss1 provides an efficient transport system for its own activator during striatonigral axon growth and potentiates the repulsive guidance cue.

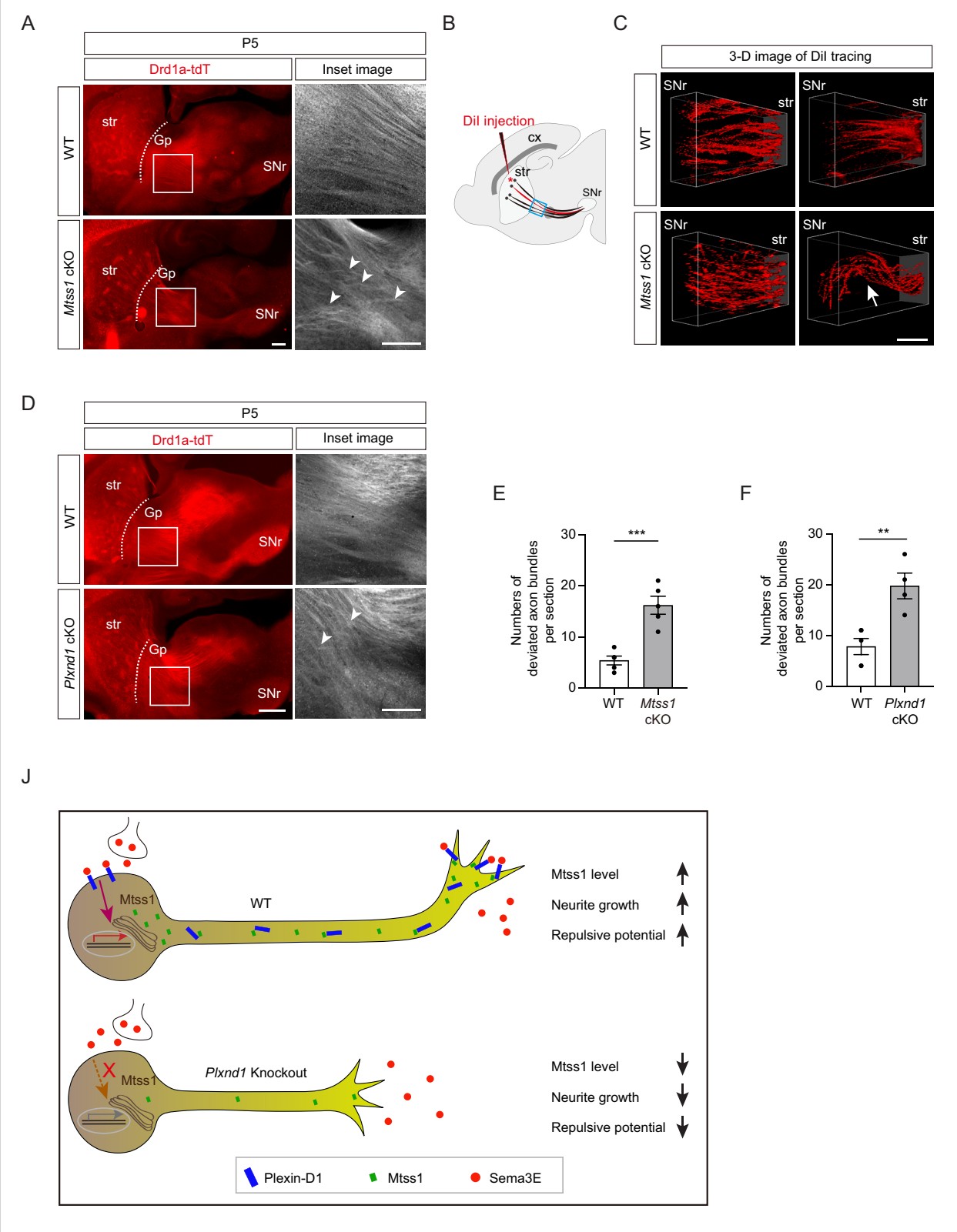

**Figure 8.** Direct-pathway medium spiny neurons (MSNs) exhibit irregular projection patterns in the absence of either Mtss1 or Plexin-D1.
(**A**) Immunohistochemistry of parasagittal sections of striatonigral projections labeled with Drd1a-tdT in wild-type (WT) or *Mtss1* conditional knockout (cKO) mice at P5. The white boxes on the left images are shown in the inset image on the right. Misguided striatonigral projections are indicated by white arrowheads. The phenotype was observed in five out of five *Mtss1* cKO mice at P5. Scale bar, 500 μm. (**B**) Schematics depicting DiI injection (red

*Figure 8 continued on next page*

*Figure 8 continued*

asterisk) into the dorsal striatum for sparse labeling of striatonigral projections. The regions indicated in the blue square were captured for three-dimensional (3D) visualization. (**C**) Representative 3D images of DiI-labeled axonal tracks in WT or *Mtss1* cKO mice at P5. Compared to the fasciculated straight projections in WT (top panels), the striatonigral projections of *Mtss1* cKO mice presented relatively defasciculated (bottom panels) and occasionally severely misrouted patterns (white arrow in the bottom-right panel). The phenotype was observed in three out of three *Mtss1* cKO mice. Scale bar, 200 µm. (**D**) Representative images showing parasagittal sections of brains from WT or *Plxnd1* cKO mice at P5. The misrouted projections are also indicated by white arrowheads in the magnified inset images. The phenotype was observed in four out of four *Plxnd1* cKO mice at P5. Scale bar, 500 µm. (**E, F**) Quantification of the number of intersecting axonal bundles within the corresponding area from Mtss1 cKO (**A**) or Plxnd1 cKO (**D**). (**E**) Student's *t*-test; n = 5 mice for WT, n = 5 mice for Mtss1 cKO. (**F**) Student's *t*-test; n = 5 mice for WT, n = 4 mice for Plxnd1 cKO. Error bars, mean ± SEM; **p<0.01, ***p<0.001 by indicated statistical tests. (**J**) Model showing that Mtss1, upregulated by the Sema3E-Plexin-D1 signaling pathway, promotes axonal growth and directs Plexin-D1 to the growth cone to receive a repulsive guidance signal.

The online version of this article includes the following source data and figure supplement(s) for figure 8:

**Figure supplement 1.** No Mtss1 was found in endothelial cells at E14.5, and no vascular defects were observed in *Mtss1*-conditional knockout (KO) mice.

**Figure supplement 1—source data 1.** Western blots shown in *Figure 8—figure supplement 1C*.

**Figure supplement 2.** Schematic summary showing striatonigral projection development via a serial reciprocal interaction of the Sema3E-Plexin-D1-Mtss1 complex.

## Gene expression by axon guidance signaling

Although many studies have reported the identification of proteins locally synthesized in the axon terminal (*Jung et al., 2012*), most proteins required for growth cone behavior are generated in and delivered from the soma. Because of the diverse roles of guidance molecules, such as driving neuronal cell migration, cell death, and axonal regeneration, as well as in traditional axonal navigation in the nervous system (*Kolodkin and Tessier-Lavigne, 2011*), guidance signaling is generally thought to be involved in the activation of a gene expression program in the nucleus. However, few studies have examined gene expression changes induced by guidance signaling at the level of transcriptional regulation, and the most definitive results have been obtained in the *Drosophila* model (*Russell and Bashaw, 2018*). A well-described example is the Frazzled (Fra) receptor in the regulation of midline axon crossing in *Drosophila*, where the truncated ICD of Fra is generated by gamma-secretase and enters the nucleus to activate transcription of the *commissureless (comm)* gene (*Neuhaus-Follini and Bashaw, 2015*). In a rodent model, microarray analysis of the Robo mutant revealed hundreds of DEGs that may be related to the dynamics of neuronal progenitors in the developing cortex (*Yeh et al., 2014*). Another example is Eph-Ephrin signaling in neural progenitor cells, where Ephrin-B1 reverse signaling downregulates miR-124 expression to inhibit neurogenesis as a posttranscriptional repressor (*Arvanitis et al., 2010*).

Through a bulk RNA-seq analysis, we showed that Sema3E-Plexin-D1 signaling changed the expression of specific genes, including Mtss1, required for the precise axon guidance of striatonigral neurons. Since ablation of Plexin-D1 expression results in the ectopic formation of thalamostriatal synapses on the MSN cell body as well as on dendrites (*Ding et al., 2012*), it is likely that Plexin-D1 is expressed in the soma of MSNs and is capable of transmitting gene expression signals. Although Sema3E-Plexin-D1 signaling regulated the expression of a set of genes in the present study, the precise signaling cascade that extends into the nucleus is still not completely understood. Nevertheless, we cannot dismiss the possibility that Mtss1 expression activation through Sema3E-Plexin-D1 signaling may be subject to post-transcriptional or translational regulatory mechanisms. Thus, to uncover the molecular mechanism through which these guidance molecules mediate specific gene expression or ensure stability, comprehensive studies on gene regulation in a suitable homogeneous cellular model are necessary. A growing body of evidence demonstrates that diverse extracellular stimuli induce mechanotransduction through dynamic changes in the actin and microtubule cytoskeletal networks, which depend on Rho family proteins to cascade signals in the cytosol and nucleus to activate gene expression (*Dupont and Wickström, 2022*; *Giehl et al., 2015*; *Miralles et al., 2003*; *Percipalle and Visa, 2006*; *Samarakoon et al., 2010*). Previously, Sema3E-Plexin-D1 signaling was shown to modulate such cytoskeletal rearrangement through the PI3K/Akt pathway in regulating endothelial cell mobility, axonal growth, and growth cone collapse (*Aghajanian et al., 2014*; *Bellon et al., 2010*; *Burk et al., 2017*). Likewise, we found that disturbance of Akt signaling reduced Mtss1 expression in young

cultured MSNs, suggesting that activation of Sema3E-Plexin-D1 signaling induces specific molecule expression through similar Akt-mediated actin dynamics.

Despite the fact that Mtss1 is a downstream molecule under Sema3E-Plexin-D1 signaling, it is still puzzling how Mtss1 expression is specifically and selectively regulated during striatonigral pathfinding. Since Mtss1 expression did not appear to be affected by *Plxnd1*-knockout at early embryonic stages and some level of Mtss1 persisted in cultured neurons lacking *Sema3e* or *Plxnd1*, we cannot exclude the possibility that other pathways activate Mtss1 expression. Nevertheless, it is certain that Mtss1 expression is under Sema3E-Plexin-D1 signaling activation during striatonigral projection and that, when axons reach their destination at P7, Mtss1 expression is downregulated. Such concurrence of Mtss1 downregulation and axonal projection leads us to speculate that target-derived factor-driven retrograde signaling may be involved in this gene regulation (*Harrington and Ginty, 2013*), and elucidating the relevant mechanism may be an interesting objective for future study. Furthermore, it is noteworthy that levels of both Plexin-D1 and Mtss1 were exclusively upregulated during the late gestation to early postnatal period, followed by an abrupt downregulation. Therefore, it is plausible that the activation of Mtss1 relies on the reinforcement of Plexin-D1 signaling. During angiogenic sprouting, the expression of Plexin-D1 in endothelial cells is controlled by VEGF signaling (*Kim et al., 2022*; *Yu et al., 2022*). However, the upstream pathway responsible for Plexin-D1 expression in neurons and the precise regulation of signaling strength in this context is still uncertain.

## Trafficking of Plexin-D1-Mtss1 in striatonigral neurons

Another intriguing finding in our study is that Plexin-D1-containing vesicles undergo both anterograde and retrograde movement along neurites in cultured direct-pathway MSNs. Until now, neurotrophin or semaphorin 3A-mediated retrograde trafficking has been extensively studied due to its crucial role in controlling the growth of axons and dendrites, promoting neuronal survival, and facilitating synaptogenesis within the peripheral nervous system. Additionally, finely regulated anterograde transport is essential for replenishing receptors on the growth cone surface to ensure the responses of appropriate target-derived guidance signals (*Scott-Solomon and Kuruvilla, 2018*; *Yamashita, 2019*). We observed that Mtss1 contributes to the Plexin-D1 localization in the growth cone of cultured MSNs and in the terminal region of descending striatonigral projections. Previously, it has been reported that signaling endosomes containing Sema3A/Plexin-A at the axonal growth cones are retrogradely transported to soma. This process facilitates the localization of AMPA receptor GluA2 to the distal dendrites, thereby regulating dendritic development in the cultured hippocampal neurons (*Yamashita et al., 2014*). Since it is known that Plexin-D1 plays a role in the somatodendritic synaptogenesis of direct-pathway MSNs (*Ding et al., 2012*), the connection of the retrograde trafficking Plexin-D1-positive vesicles in this process or other developmental mechanisms like retrograde Sema3A signaling in the soma remains to be elucidated.

## Axon-specific role of Sema3E-Plexin-D1-Mtss1 in striatonigral neuron

Sema3E-Plexin-D1 signaling is involved in dendritic synapse formation during the postnatal stage as well as traditional axon projection from the embryonic to the postnatal stage in the basal ganglia circuit (*Ding et al., 2012*). In addition, since axon projection and synapse formation are sequential events that occur in different cellular compartments during circuit establishment (*Kuo and Liu, 2019*), Sema3E-Plexin-D1 signaling may play discrete roles within different parts of a neuron, such as the dendrite versus the axon, through a unique subset of molecules. Indeed, our observations that the duration of Mtss1 expression coincided with active striatonigral axon pathfinding, Mtss1 was significantly expressed on the axonal side, and axon projection defects were observed after Sema3E-Plexin-D1 signaling disruption suggest that the Sema3E-Plexin-D1-Mtss1 complex appears to mainly regulate axon projection and guidance, at least during striatonigral pathway development. However, the cellular distribution of Mtss1 is somewhat controversial, as its expression is mainly observed on the dendritic side of Purkinje cells and hippocampal neurons, and its knockout shows defective dendritic arborization and spine formation (*Kawabata Galbraith et al., 2018*; *Saarikangas et al., 2015*; *Yu et al., 2016*), while a study has also shown that Mtss1 is more localized in the axoplasmic compartment of Purkinje cells (*Hayn-Leichsenring et al., 2011*). In this study, *Mtss1*- knockout did not show clear dendritic changes in MSNs, at least at P5 when an active axon trajectory occurs; however, given the enlarged ventricle and decreased cortical volume phenotype in knockout adults (*Minkeviciene*

et al., 2019), later Mtss1 expression (even at low levels) may affect overall dendritic development. Consistent with this assumption, a previous study reported a slight decrease in dendritic spine density but a slight increase in the number of dendritic crossings without a change in total dendritic length in the direct pathway MSNs of 3- to 4-week-old *Plxnd1*- or *Sema3e*-knockout mice (*Ding et al., 2012*). Thus, we cannot exclude the possibility that the Sema3E-Plexin-D1-Mtss1 complex modulates MSN dendritic development later than the axonal pathfinding period.

## Counterintuitive mechanism of attractive and repulsive guidance by Mtss1

From a traditional guidance point of view, our finding was somewhat unexpected because it was counterintuitive: conventional repulsive guidance cues that mediate growth inhibition induce upregulation of a positive regulator of neurite extension. We revealed that Mtss1 plays a dual role in striatonigral neurons, axon extension by membrane protrusion ability, and axon guidance by efficient Plexin-D1 trafficking to the growth cone. Therefore, our findings represent two important discoveries regarding the axon guidance mechanism. First, axon guidance signaling can switch on the specific regulatory program necessary for facilitating its own function, thereby generating the appropriate machinery to accomplish an intrinsic guidance role during neuronal pathfinding. Second, molecules such as Mtss1 coordinate positive and negative growth potentials in the axonal pathfinding route. Axon guidance cues require various auxiliary proteins to perform their programmed functions, in particular, transporting guidance receptors to the growth cone, endocytic sorting, and activating signaling cascades (*O'Donnell et al., 2009*), but none of the cofactors discovered to date have induced direct expression regulation of the guidance signaling with which it is involved. Previous studies have shown that Sema3E-Plexin-D1 signaling can switch its role from repulsive to attractive by interacting with the coreceptors neuropilin-1 (Npn1) and vascular endothelial growth factor receptor type 2 (VEGFR2) in subiculum neurons (*Bellon et al., 2010*; *Chauvet et al., 2007*). This mechanism is not applicable to developing striatonigral MSNs because the Npn1 receptor is not expressed in early postnatal striatal neurons (*Ding et al., 2012*). Instead, striatonigral neurons seem to adopt a new strategy, such as activating their own attractive driver to promote axonal growth. From a phenotypic perspective, it is also plausible that the positive or negative guidance roles of Mtss1 may be interdependent. Striatonigral extension defects could potentially lead to inadequate responses to Sema3E signals during pathfinding, consequently resulting in misguided projections. Similarly, guidance defects may disrupt normal signaling processes within the growth cone, thereby transmitting incorrect information for the replenishment of Plexin-D1 to the soma. This erroneous signaling could weaken Mtss1 expression and normal trafficking, ultimately leading to axonal growth defects.

## Diverse roles of BAR domain proteins in the axon guidance signaling pathway

Mtss1 promotes membrane curvature through the I-BAR domain and induces the redistribution of lipids in the membrane, thereby increasing the local phosphatidylinositol 4,5-bisphosphate (PIP$_2$) level at the negatively curved membrane. The elevation of local PIP$_2$ levels leads to membrane binding of the I-BAR domain via electrostatic interactions (*Lin et al., 2018*). Interestingly, Sema3E binding to Plexin-D1 elevates PIP$_2$ locally to activate Arf6, resulting in rapid focal adhesion disassembly (*Sakurai et al., 2011*). Since Mtss1 I-BAR can interact with Plexin-D1 at the curved membrane, the local increase in PIP$_2$ in the curved membrane region may trigger signaling cascades. Although BAR-domain proteins play pivotal roles in membrane dynamics, a direct association between BAR-domain proteins and axon guidance receptors has not been extensively studied, and functional relevance in vivo is unclear. One example is the srGAP2 protein, which has been studied and shown to bind directly with the SH3 domain of the Robo1 guidance protein in cooperation with the F-BAR and RhoGAP domains (*Guez-Haddad et al., 2015*). Interestingly, in endothelial cells, Plexin-D1 forms a complex with SH3BP1, another small GTPase protein containing the N-BAR domain (*Tata et al., 2014*). Similar to the effect of Mtss1 and Plexin-D1 complex formation via the I-BAR domain, SH3BP1 colocalized with Plexin-D1 at lamellipodia in a complex formed via the N-BAR domain and mediates Sema3E-induced cell collapse through Rac1 activity regulation. However, in contrast to the effect of the Mtss1 mutant lacking the I-BAR domain, which failed to change the cell morphology, SH3BP1 lacking N-BAR led to cell collapse. Moreover, Sema3E binding to Plexin-D1 caused SH3BP1 to be released from the

complex, whereas Sema3E did not interfere with the Plexin-D1-Mtss1 complex. In addition, the presence of Mtss1 had no effect on the endocytosis of Plexin-D1 by Sema3E or on the ability of Sema3E to bind to Plexin-D1. Hence, it is plausible that Mtss1 plays a role as a facilitator of Plexin-D1 trafficking rather than as a direct downstream signaling transducer such as SH3BP1. Because of the diverse roles played by Sema3E-Plexin-D1 across cell types, the effect of signaling induced by this guidance cue may be determined by distinct downstream molecules that share structural similarities in a relevant biological context. Another intriguing finding is the specific complex formation between Plexin-D1 and Mtss1 in the present study. Plexin-D1 did not form a complex with other BAR-domain-containing proteins, srGAP2 and IRSP53, known to have a role in neurons. While we did not conduct an extensive examination, it is noteworthy that at least two other Plexin proteins were unable to form a complex with Mtss1. Therefore, it would be interesting to explore the structural characteristics underlying these interactions.

## Variable axon projection defects in direct pathway MSNs mediated by Sema3E-Plexin-D1 signaling

To modulate movement information conveyed through basal ganglia circuitry, two distinct types of striatal MSNs send axonal projections to different targets: a direct-pathway MSN expresses the dopamine D1 receptor to promote movement, and an indirect-pathway MSN expresses the dopamine D2 receptor to inhibit movement (*Kreitzer and Malenka, 2008*; *Surmeier et al., 2007*). Because of these unique functional and anatomical features, decoding the distinct molecular properties of the two types of MSNs and the regulatory mechanisms involved in circuitry formation is important. A few previous transcriptome analyses have been performed with juvenile and adult mouse brains (*Heiman et al., 2008*; *Kronman et al., 2019*; *Lobo et al., 2006*), but an understanding of the molecular repertoire of each MSN during development is very limited. In the present study, Mtss1 was identified as a selective molecule expressed in striatonigral projection neurons mediated by Sema3E-Plexin-D1 signaling, but its expression is limited to only the early striatonigral projection period; therefore, the previous transcriptome database in adults may have failed to identify Mtss1 as a specific marker molecule in direct-pathway MSNs.

In this study, Mtss1 expression was found to be relatively high during the perinatal period and then was dramatically downregulated by P7, by which time striatonigral projection has been completed (*Morello et al., 2015*). Consistent with a previous study showing that Plexin-D1-positive cells in the striatum were first detected on E14.5 (*van der Zwaag et al., 2002*), we found that its expression increased in the early postnatal stage. During a similar developing window in the striatonigral pathway, Sema3E was predominantly expressed in the GP and rTh/ZI, which is located in the route to the substantia nigra; therefore, the absence of repulsive Sema3E-Plexin-D1 signaling resulted in defects in striatonigral projection (*Burk et al., 2017*; *Chauvet et al., 2007*; *Ehrman et al., 2013*). However, we did not find ectopic projections, misguidance defects, or enlarged paths in *Mtss1*- or *Plxnd1*-knockout mice. Nevertheless, we observed fewer projections with aberrantly tangled patterns. These discrepancies may be explained by the following observations. First, because we used a genetic model to selectively label the striatonigral projections, we could detect abnormal phenotypes at a better specific resolution. Second, we noticed a certain degree of developmental retardation in the *Mtss1*- or *Plxnd1*-knockout neonates, even among those in the same litter; therefore, we strictly selected samples on the basis of body weight. However, despite the low striatonigral projection formation rate in the *Mtss1*-, *Plxnd1*-, or *Sema3e*-knockout mice, the adult mice showed a normal overall range of projection boundary size and width, suggesting that a decrease in repulsive signals in the mutants may have widened the descending projections. Third, the mouse genetic background may have led to the observed phenotypic discrepancies. We previously observed that certain vascular phenotypes were more evident in *Sema3e*-knockout mice with a 129SVE background than in those with a C57BL/6 background (*Oh and Gu, 2013b*), implying that relatively minor defects may vary depending on the genetic background.

Although Sema3E-Plexin-D1 has a similar mechanism of action as a common guidance cue in the nervous and vascular systems, it also exhibits completely different behavior in each system (*Oh and Gu, 2013a*). For example, the Plexin-D1-Npn1-VEGFR2 complex transmits an attractive signal upon Sema3E binding in specific neurons (*Bellon et al., 2010*), but endothelial cells still respond negatively to Sema3E despite the presence of all three receptors (*Oh and Gu, 2013b*). Interestingly,

Sema3E-Plexin-D1 utilizes Akt-mediated cytoskeletal dynamics for its signaling cascade in both neuronal and endothelial cells (*Burk et al., 2017*; *Moriya et al., 2010*), and various auxiliary molecules, such as SH3BP1, Arf6, and GIPC1, have been revealed in each cell (*Burk et al., 2017*; *Sakurai et al., 2010*; *Tata et al., 2014*). However, only a few of these cofactors have shown a common molecular mechanism and expression by exchanging cells with each other, suggesting that Sema3E-Plexin-D1 signaling requires unique factors to perform different functions. Similarly, we identified Mtss1 as a downstream factor of the Sema3E-Plexin-D1 cue, but its function seems to be limited in neurons, at least in vivo. Therefore, it is necessary to elucidate whether Sema3E-Plexin-D1 signaling activates a specific pool of regulatory factors required for vascular development and to compare this with the results obtained in neurons, which will help us understand the underlying mechanism of specific molecule expression by guidance signaling. Furthermore, given the tremendous complexity of wiring in the central nervous system, it will be intriguing to discover new dual-function molecules similar to those in the Sema3E-Plexin-D1-Mtss1 complex that are involved in the formation of other circuits in the future.

## Materials and methods

### Mice

*Plxnd1^flox/flox^* (*Plxnd1^f/f^*) mice (*Kim et al., 2011*) and *Sema3e^+/-^* mice (*Chauvet et al., 2007*) were maintained on a C57BL/6 (#000664, The Jackson Laboratory) background. Nestin-Cre (#003771), Tie2-cre (#008863), and Drd1a-tdTomato (#016204) mice were obtained from The Jackson Laboratory (Bar Harbor, USA) and maintained on the same background. The frozen sperm of *Mtss1^flox/+^* mice were generously provided by Dr. Mineko Kengaku and rederived at the Laboratory Animal Resource Center in the Korea Research Institute of Bioscience and Biotechnology (Cheongju, Korea). All protocols for animal experiments were approved by the Institutional Animal Care and Use Committee of Korea Brain Research Institute (IACUC-18-00008, 20-00012). All experiments were performed according to the National Institutes of Health Guide for the Care and Use of Laboratory Animals and ARRIVE guidelines.

### Cell lines and primary striatal neuron culture

COS7 (21651, Korean Cell Line Bank), HEK293T (CRL-3216, ATCC), HUVEC (CC-2935, Lonza), and HCMEC/D3 (SCC066, Millipore) cell lines were purchased from the indicated companies, and the cell culture media and culture conditions were as provided by the respective companies. All cell lines were initially authenticated by the company and were tested to be mycoplasma negative. Primary mouse striatal neurons were isolated from neonatal pups as described in a previous report with some modifications (*Penrod et al., 2011*). Whole striatal tissues including the globus pallidus were digested with 20 units/ml papain (LS003124, Worthington, Lakewood, USA) diluted in dissection solution (5 mM MgCl$_2$ and 5 mM HEPES in 1× Hanks' balanced salt solution, pH 7.2) followed by multiple washes in inhibition solution (0.1% BSA and 0.1% Type II-O trypsin inhibitor diluted in dissection solution). The tissues were resuspended in neuronal plating medium (1 mM pyruvic acid, 0.6% glucose, and 10% heat-inactivated horse serum in Minimum Essential Medium with Earle's Salts) and triturated 50 times with a fire-polished Pasteur pipette. The dissociated neurons were centrifuged at 1000 × *g* for 5 min and resuspended in fresh neuronal plating media for cell counting. Then, the cells were plated on coverslips or culture dishes coated with 50 µg/ml poly-D-lysine (P6407, Sigma) and 1 µg/ml laminin (354232, Corning) at a density of 3 × 10$^4$ cells/cm$^2$. After 4 hr of incubation at 37°C, the plating media were replaced with neuronal growth media (0.5 mM L-glutamine, B27 supplements in neurobasal medium [10888022, Gibco]), and a quarter of the media was replaced with fresh growth media every 3 d until harvest.

### Plasmids

A pBK-CMV vector containing VSV-tagged human Plexin-D1 cDNA (*Gu et al., 2005*) was recloned into a pCAG vector (pCAG-vsv-hPlexin-D1), and a Plexin-D1 construct lacking an ICD (amino acids deleted: 1299–1925) was generated by PCR-based mutagenesis (pCAG-vsv-hPlexin-D1ΔICD). To generate the pCAG-hPlexin-D1-GFP fusion construct, a GFP gene fragment was inserted at the C-terminus of the pCAG-vsv-hPlexin-D1 plasmid. pAPtag-5-Sema3E vectors were reported previously (*Chauvet et al., 2007*), and the mouse Plexin-D1 extracellular domain (amino acids: 1–1269) was amplified from

mouse Plexin-D1 cDNA and directly cloned into a pAPtag-5 vector (pAPtag5-mPlexin-D1-ECD). The human full-length Mtss1 expression construct was purchased from Origene (pCMV6-hMtss1, Cat# RC218273, USA), and Myc-tagged Mtss1 deletion constructs (Mtss1ΔI-BAR [amino acids deleted: 1–250], Mtss1ΔWH2 [amino acids deleted: 714–745], and Mtss1-I-BAR [amino acids: 1–250]) were generated by PCR-based mutagenesis. To generate full-length mouse Flag-Plexin-D1, SH3BP1-HA, and HA-srGAp2, each gene was isolated directly from the mouse brain cDNA library and cloned into pCAG vector with the tagging fragment. The mouse vsv-plexin-B2, mouse vsv-plexin-B3, and human pECE-M2-BAIAP2 (IRSP53-Flag) were purchased from Addgene (#68038, #68039, #31656, USA).

## RNA sequencing analysis

RNA sequencing (RNA-seq) library preparation and sequencing were conducted at Ebiogen (Seoul, South Korea). Libraries were constructed using a NEBNext Ultra Directional RNA-seq Kit customized with mouse-specific oligonucleotides for rRNA removal. Directional mRNA-seq was conducted using the paired-end, 6 Gb read option of the Illumina HiSeq X10 system.

## Bioinformatic analysis for RNA-seq

The entire analysis pipeline of RNA-seq was coded using R software (version 3.6), which was controlled by systemPipeR (version 1.18.2). The raw sequence reads were trimmed for adaptor sequences and masked for low-quality sequences using systemPipeR. Transcript quantification of the RNA-seq reads was performed with GenomicAlignments (version 1.20.1) using reads aligned with the *Mus musculus* transcriptome annotation using Rsubread (version 1.24.6). The fragments per kilobase of transcript per million mapped reads (FPKM) values were calculated using the fpkm function of DESeq2 (version 1.24.0) and were processed using the robust median ratio method. Transcript reads were normalized by the voom function of Limma (version 3.40.6). To determine if a transcript was differentially expressed (DE), EdgeR (version 3.26.7) calculated the results based on the normalized counts from entire sequence alignments. Significantly DE transcripts with a fold change greater than the raw FPKM value (>2) and adjusted p-value (<0.01) in all experimental comparisons were selected and used for further analysis. Gene annotations were added with an online database using Ensembl biomaRt (version 2.40.4), and visualization was performed using the R base code and gplots package (version 3.0.1.1). For DEG sets, hierarchical cluster analysis was performed using complete linkage and Euclidean distance to measure similarity. All data analysis and the visualization of DEGs were conducted using R version 3.0.2 (https://www.r-project.org/).

## Quantitative reverse transcription PCR (qRT-PCR)

Total RNA was extracted from dissected tissue using TRIzol (15596026, Thermo). cDNA was synthesized from 200 ng of total RNA with a QuantiTect Reverse Transcription Kit (205313, QIAGEN). Quantitative PCRs were carried out in triplicate using SYBR Green I Master Mix (S-7563, Roche) on a LightCycler 480 system (Roche). Expression was calculated using the $2^{-\Delta\Delta Ct}$ method with *Gapdh* as a reference. The following primers were used (forward primer and reverse primer, respectively): *Plxnd1*: 5'-CTAGAGATCCAGCGCCGTTT, 5'-GGCACTCGACAGTTGGTACA, *Mtss1*: 5'-CCTTTCCC TCATTGCCTGCCT, 5'-TCTGAGATGACGGGAACATGCC, and *Gapdh*: 5'-TGACGTGCCGCCTGGA GAAAC, 5'-CCGGCATCGAAGG TGGAAGAG.

## Transfection

DNA expression constructs were transfected into COS7 or HEK293T cells by Lipofectamine 2000 (11668019, Invitrogen) in OPTI-MEM (31985-070, Gibco) for 4 hr according to the manufacturer's instructions and then replaced with normal culture media until the next procedure. For imaging analysis, 0.5 µg of DNA was transfected into COS7 cells ($1 \times 10^4$ cells/cm²) cultured on coverslips in a 12-well plate. For biochemical analysis, 4 µg of DNA was transfected into HEK293T cells ($3 \times 10^4$ cells/cm²) cultured on a 10 cm dish. To achieve high transfection efficiency into primary neurons, the nucleofection technique using a Lonza Amaxa Nucleofector was performed following the manufacturer's instructions (Basic Nucleofector Kit for Primary Mammalian Neurons, VAPI-1003). To transfect into neurons, 4 µg of expression constructs were added to at least $1 \times 10^6$ isolated neuronal cells for each electroporation, and the transfected cells were plated and cultured as described in the previous section.

## Alkaline phosphatase (AP)-conjugated ligand preparation and binding analysis

AP-conjugated Sema3E and Plexin-D1-ECD ligands were generated in HEK293T cells, and ligand binding experiments were performed as described in previous reports (*Chauvet et al., 2016*; *Chauvet et al., 2007*). Briefly, the AP-conjugated expression construct was transfected into cells by Lipofectamine 2000 and cultured overnight in Dulbecco's modified Eagle's medium containing 10% fetal bovine serum (FBS). Then, the medium was replaced with OPTI-MEM and harvested at 5 d post transfection. The collected conditioned medium was filtered to increase the ligand concentration.

To measure the binding ability of AP-Sema3E to the Plexin-D1 receptor, COS7 cells on a six-well plate were transfected with each expression vector and cultured for 24 hr. The next day, the cells were washed in HBHA buffer (1× HBSS, 0.5 mg/ml BSA, 0.5% sodium azide, and 20 mM HEPES [pH 7.0]) and incubated with 2 nM AP or AP-Sema3E for 1 hr at room temperature (RT). After seven washes in the HBHA buffer, the cells were lysed in 1% Triton X-100 and 10 mM of Tris–HCl (pH 8.0), and the supernatant was obtained by centrifugation at $13,000 \times g$ for 10 min. The lysates were heat-inactivated at 65°C for 10 min, each lysate was used for AP concentration using a BioMate 3S spectro-photometer (Thermo Scientific), and the amount of protein was measured by BCA assay.

For AP-conjugated ligand binding analysis of tissues, 20-μm-thick cryosections were fixed in cold methanol for 8 min and preincubated in 1× phosphate-buffered saline (PBS) containing 4 mM $MgCl_2$ and 10% FBS for 1 hr. Next, a binding solution (1× PBS–$MgCl_2$ and 20 mM HEPES, pH 7.0) containing 2 nM AP-Sema3E was applied, and sections were incubated for 2 hr at RT. After five washes in 1× PBS–$MgCl_2$, the sections were briefly soaked in acetone–formaldehyde fixative (60% acetone, 1.1% formaldehyde, and 20 mM HEPES, pH 7.0) and heat-inactivated in 1× PBS at 65°C for 2 hr. Next, the sections were incubated in AP buffer (NBT/BCIP tables, 11697471001, Roche) until clear purple precipitation was observed at RT. For quantification, three brain sections per animal were analyzed and averaged. For the AP-Sema3E binding analysis of growth cones, MSNs grown for 6 d on glass coverslips were washed in 1× PBS, immediately fixed in cold methanol for 5 min, and blocked in TBS buffer (100 mM TBS, 4 mM $MgCl_2$, 4 mM $CaCl_2$, pH 7.4) with 10% FBS at RT for 1 hr. The MSNs were then incubated in 0.5 nM AP-Sema3E ligand diluted in blocking solution at RT for 1 hr. After five washes in TBS buffer, the MSNs were fixed in acetone/formaldehyde solution, heat-inactivated, and incubated in AP buffer as described above.

## AP treatment of HUVECs and HCMEC/D3 cells

HUVECs were purchased from Lonza, and the HCMEC/D3 cell line was obtained from Millipore. The cell culture media and culture conditions used followed the information provided by the respective companies. Cultured HUVECs and HCMEC/D3 cells were treated with 2 nM AP or AP-Sema3E for 24 hr. Then, the cells were lysed and prepared for western blotting as described below.

## Immunoblotting

Brain tissue was collected in radioimmunoprecipitation assay (RIPA) buffer (50 mM Tris–HCl [pH 8.0], 150 mM NaCl, 1% NP-40, and 1% sodium deoxycholate) with a protease inhibitor cocktail (78444, Thermo Fisher Scientific), and protein amounts were quantified using a BCA protein assay kit (23227, Thermo Fisher Scientific). A total of 40 μg of protein was loaded into each well of a sodium dodecyl sulfate (SDS) polyacrylamide gel, after which it was separated and transferred to a polyvinylidene fluoride membrane (IPVH00010, Merck) at 100 V for 90 min. All membranes were blocked in Everyblot blocking buffer (12010020, Bio-Rad) for 1 hr and probed overnight with primary antibodies in blocking buffer at 4°C. The primary antibodies included the following: anti-Mtss1 (1:1000, Novus, NBP2-24716), anti-Plexin-D1 (1:1000, AF4160, R&D Systems), anti-β-actin (1:5000, 5125S, Cell Signaling), anti-Myc (1:1000, 2276S, Cell Signaling), anti-vsv (1:1000, ab3861, Abcam), anti-Sema3E (1:500, LS-c353198, LSBio), anti-p-AKT (1:1000, 9271, Cell Signaling), and anti-AKT (1:1000, 9272S, Cell Signaling). The membranes were incubated in TBST, and the appropriate horseradish peroxidase (HRP)-conjugated secondary antibodies and bands were developed with enhanced chemiluminescence using Fusion FX7 (Vilber, Germany) and then analyzed using ImageJ software.

## Immunoprecipitation

HEK293T cells were transfected with Lipofectamine 2000, and after 24 hr, they were lysed in a buffer consisting of 100 mM Tris–HCl (pH 7.5), 100 mM EDTA, 150 mM NaCl, and 1% Triton X-100 with freshly added phosphate and protease inhibitors. The cell lysates were centrifuged at 13,000 × $g$ for 10 min at 4°C, and supernatants were incubated with antibodies (1:200) at 4°C overnight. Then, the protein lysates were incubated with magnetic beads for 1 hr at 4°C. Next, the beads were washed five times with lysis buffer, and the bound proteins were eluted with a 2× SDS sample buffer by heating the beads at 95°C for 5 min. The samples were then analyzed by SDS-PAGE and western blotting. The following antibodies were purchased from commercial sources: anti-Myc (1:1000, 2276S, Cell Signaling), anti-vsv (1:1000, ab3861, Abcam), anti-Mtss1 (1:1000, NBP2-24716, Novus), anti-Plexin-D1 (1:1000, AF4160, R&D Systems), and anti-β-actin (1:5000, 5125S, Cell Signaling).

## Protein–protein interaction assay

The I-BAR domain of human Mtss1 was purchased from Sino Biological (Cat# 13085-H10E). The human Plexin-D1 cytosolic domain was prepared. Briefly, the coding sequence of the human PLXND1 (NM_105103.3) cytosolic domain (amino acid residue 1297–1925; hPLXND1_Cyto) was subcloned into the mammalian expression vector CAGs-MCS EEV (SBI, Palo Alto, CA) in-framed fused with a C-terminal twin strep tag (SAWSHPQFEKGGGSGGGSGGGSAWSHPQFEK). HEK293 GnTI⁻ cells were transfected with the PLXND1 expression vector and grown in FreeStyle 293 Expression Medium supplemented with 2% (v/v) FBS in a humidified $CO_2$ incubator (35°C, 5% $CO_2$) with shaking at 150 rpm for 3 d. Cells were harvested, resuspended in resuspension buffer (500 mM NaCl, 10 mM Tris–HCl, pH 8.0, 5% [v/v] glycerol, 3 mM β-mercaptoethanol, protease inhibitor cocktail [Roche], 0.4 mg/ml DNase I [GOLDBIO]), and disrupted by using a Dounce homogenizer followed by sonication. After the removal of cell debris by centrifugation, the cell lysate was loaded onto preequilibrated Strep-Tactin XT 4Flow resin (IBA Lifesciences GmbH, Germany) and washed with wash buffer (500 mM NaCl, 10 mM Tris–HCl, pH 8.0, 5% [v/v] glycerol, 3 mM β-mercaptoethanol). hPLXND1_Cyto with a twin strep tag was eluted with elution buffer (50 mM biotin, 500 mM NaCl, 10 mM Tris–HCl, pH 8.0, 5% [v/v] glycerol, 3 mM β-mercaptoethanol) and further purified by using size-exclusion chromatography on a size-exclusion column (Superdex 200) equilibrated with FPLC buffer (150 mM NaCl, 20 mM Tris, pH 8.0, 10% glycerol [v/v], 2 mM DTT). To obtain the tag-free hPLXND1_Cyto proteins, the C-terminal twin strep tag was removed by incubating with the 3C protease (Takara Bio, Japan; 1.5 unit/50 mg protein) and ran on a size-exclusion column. To analyze direct binding, Mtss1-I-BAR with Plxind-D1-ICD was mixed in binding buffer (50 mM Tris–HCl, 0.1% Triton X-100) and incubated overnight at 4°C. The mixed proteins were then incubated with Ni-NTA beads for 1 hr at 4°C. Next, the beads were washed five times with binding buffer, and the bound protein was eluted with 2× SDS sample buffer by heating the beads at 95°C for 5 min. The samples were then analyzed by SDS-PAGE and stained with Coomassie blue solution. After washing with water five times, the stained gels were visualized using a Bio-Rad instrument.

## Cell surface biotinylation and endocytosis analysis

Transfected COS7 cells on a 100 mm dish were biotinylated by incubation in 1 mg/ml NHS-SS-Biotin (21331, Thermo Scientific), diluted in 1× PBS containing 1 mM $MgCl_2$ and 0.1 mM $CaCl_2$ (PBS–MC) for 15 min, washed in PBS–MC containing 10 mM glycine at least three times, and then rinsed in ice-cold PBS–MC twice at 4°C. For the negative control, the cells were incubated in stripping buffer (50 mM glutathione, 75 mM NaCl, 10 mM EDTA, 75 mM NaOH, and 1% bovine serum albumin [BSA]) and washed twice in PBS–MC. For the neutralization of glutathione, the cells were incubated in PBS–MC containing 50 mM iodoacetamide (I1149, Sigma) three times. All biotinylated or stripped cells were lysed in ice-cold RIPA buffer (50 mM Tris–HCl [pH 8.0], 150 mM NaCl, 1% NP-40, and 1% sodium deoxycholate) with a protease inhibitor cocktail, and 100 µg of protein extracts was incubated in prewashed streptavidin agarose resin (20357, Thermo Scientific) overnight and rotated throughout. Cell extracts were serially washed in bead-washing solution (Solution A: 150 mM NaCl, 50 mM Tris–HCl [pH 7.5], and 5 mM EDTA; Solution B: 500 mM NaCl, 50 mM Tris–HCl [pH 7.5], and 5 mM EDTA; Solution C: 500 mM NaCl, 20 mM Tris–HCl [pH 7.5], and 0.2% BSA) followed by another wash in 10 mM Tris–HCl (pH 7.5). The bound biotinylated proteins were recovered by adding 2× sample buffer and boiling extracts for 5 min, and then the supernatants were subjected to western blotting.

To analyze endocytic protein levels, cells were incubated for 25 min at 37°C in the presence of prewarmed culture media with 2 nM AP or AP-Sema3E ligands after surface biotinylation. Then, the biotinylated proteins remaining on the cell surface were removed by stripping procedures, and the rest of the experiment was continued as described in the above section. Except for those used in the ligand stimulation process, all reagents were prechilled, and experiments were performed in an ice or cold chamber.

## Immunostaining

For immunocytochemistry, cultured cells or neurons on coverslips were fixed in 4% paraformaldehyde (PFA) for 5 min and washed several times in PBS. Then, the cells were permeabilized in PBST (PBS containing 0.1% Triton X-100) for 5 min, blocked with 5% horse serum in PBST for 60 min at RT, and incubated with primary antibodies diluted in blocking solution overnight at 4°C. The next day, the samples were washed with PBST three times and incubated for 1 hr with Alexa Fluor 488-, 594-, or 647-conjugated secondary antibodies (1:1000, Thermo). To enable visualization of the F-actin, Alexa Fluor-conjugated phalloidin (1:50, Thermo) was added during the secondary antibody incubation. After being washed again with PBST, the samples were mounted with Prolong Diamond antifade solution containing DAPI (P36962, Thermo). Image processing was performed using ImageJ or Adobe Photoshop (Adobe Photoshop CC2019) under identical settings. All other immunostaining procedures were the same as those described above. The following primary antibodies for immunocytochemistry were used: anti-vsv (1:1000, ab3861, Abcam), anti-Myc (1:1000, 2276S, Cell Signaling), phalloidin Alexa Fluor 488 (1:50, A12379, Thermo), phalloidin Alexa Fluor 647 (1:100, A22287, Thermo), anti-RFP (1:1000, ab62341, Abcam), anti-RFP (1:1000, MA5-15257, Thermo), anti-Tau (1:500, sc-1995, Santa Cruz), and anti-alpha-tubulin (1:1000, T5168, Sigma). Images were collected using a Nikon Eclipse Ti-U microscope (Nikon, Japan), Leica TCS SP8 Confocal Microscope (Leica, Germany), or Structured Illumination Microscope (Nikon).

For immunohistochemistry with tissue samples, brains were fixed in 4% PFA overnight and equilibrated with 20% sucrose in 1× PBS. Mouse brain sections were cut into 20 μm slices on a cryostat (Leica Microsystems Inc, Buffalo Grove, IL). Mouse brain sections were permeabilized in PBST (PBS containing 0.2% Triton X-100) for 10 min, blocked with 2% BSA and 5% normal donkey serum in PBST for 60 min at RT, and then incubated in primary antibodies diluted with 2% BSA in PBST overnight at 4°C. The following primary antibodies were used: anti-Mtss1 (1:1000, NBP2-24716, Novus), anti-RFP (1:1000, MA5-15257, Thermo), anti-neurofilament (1:500, 2H3, Hybridoma Bank), anti-CD31 (1:500, 553370, BD Biosciences), and anti-cleaved caspase 3 (1:1000, 9661, Cell Signaling). After being washed with PBS/0.2% Tween 20 (PBST) three times, sections were incubated for 1 hr with Alexa Fluor 488-, 594-, or 647-conjugated secondary antibodies (1:1000, Invitrogen). For negative controls, brain sections were stained with secondary antibodies only. Image processing was performed using ImageJ or Adobe Photoshop (Adobe Photoshop CC2019).

## Growth cone collapse and neurite length analysis

For the growth cone collapse assay, striatal neurons at DIV3 were incubated with 5 nM AP or AP-Sema3E 3 for 25 min. For the preservation of the growth cone structure, 8% PFA was directly added to cultured neurons to equalize at 4% PFA for 10 min at 37°C, and subsequently, another 5 min round of 4% PFA fixation was performed on ice before the immunostaining procedure. Growth cone images were collected from tdT-positive neurons using a Structured Illumination Microscope (SIM, Nikon), and collapsed growth cones were determined blindly. Growth cones with broad lamellipodia were considered intact, whereas those with a few filopodia lacking lamellipodia were defined as collapsed according to previous guidelines (*Oh and Gu, 2013b*).

For measurement of neurite length, dissociated striatal neurons were cultured in the presence of 5 nM ligands and immunostained as described in the above section. The neurons were imaged by a fluorescence microscope (Nikon ECLIPSE Ti-U), and the longest neurite length from tdT-positive neurons was determined using ImageJ software. The neurites that formed a network with another neurite and those whose longest protrusions were smaller than twice the cell body diameter were excluded from measurement according to previous guidelines (*Chauvet et al., 2016*). For quantification of the degree of colocalization, Pearson's correlation coefficients were calculated using the manufacturer's software (Nikon, NIS-Elements software).

## In situ hybridization (ISH)

ISH was performed under RNase-free conditions as described in a previous study (*Ding et al., 2012*). After fixation in 4% PFA for 20 min, 20-μm-thick cryosections were preincubated in hybridization buffer (5× Denhardt's solution, 5× saline sodium citrate [SSC], 50% formamide, 0.25 mg/ml Baker yeast tRNA, and 0.2 mg/ml salmon sperm DNA) for 2 hr at RT. Next, the sections were hybridized in the same buffer containing the indicated digoxigenin-conjugated riboprobe at 60°C overnight. After hybridization, the sections were washed in a serial SSC buffer and formamide solution and then preincubated in buffer 1 (100 mM Tris–HCl, pH 7.5, 150 mM NaCl) with a 1% blocking reagent (Roche) for 1 hr at RT. Next, the sections were incubated with sheep anti-digoxigenin-AP antibody (1:3000, Roche) for 90 min at RT, washed in buffer 1, and then incubated in AP buffer (100 mM Tris–HCl, pH 9.5; 100 mM NaCl; and 5 mM $MgCl_2$) containing 4-nitro blue tetrazolium chloride (NBT, Roche), 5-bromo-4-chloro-3-indolyl-phosphate (BCIP, Roche), and levamisole (1359302, Sigma) until purple precipitates were observed. After mounting them with coverslips, the samples were analyzed using confocal laser-scanning microscopy with a Nikon Eclipse Ti-U Microscope or Leica TCS SP8 Confocal Microscope. For double fluorescence ISH, the tyramide signal amplification method with minor modifications was used according to the manufacturer's instructions (NEL753001KT, PerkinElmer). The following anti-sense riboprobes were used: *Plxnd1* (*Ding et al., 2012*), *Sema3e* (*Gu et al., 2005*), and *Mtss1* (Allen Brain Atlas, Probe RP_040604_01_B06).

## Live-cell imaging

Isolated striatal neurons were transfected with an expression plasmid, as described in the above section, and were plated onto a 35 mm confocal dish (211350, SPL Life Sciences) at a density of $1 \times 10^4$ cells/$cm^2$. At DIV6, time-lapse imaging was performed in a stage-top cell incubator (37°C with 5% $CO_2$ supplied). Images were acquired at 1 s intervals for 3 min using a Nikon A1-Rsi confocal microscope. To quantify Plexin-D1-GFP trafficking, time-lapse images were analyzed using Particle Tracking Recipe in AIVIA microscopy image analysis software (Aivia Inc).

## Golgi staining and dendrite analysis

Golgi staining was conducted according to the manufacturer's protocol (FD Rapid GolgiStain Kit [PK401A, FD NeuroTechnologies, Inc]). In brief, P5 mouse brains were immersed in a staining solution for 2 wk before being transferred to a wash solution for 4 d. Then, 100 μm slices were obtained using a vibratome and collected on gelatin-coated slides. During the staining process, the slices were washed twice with distilled water for 4 min, immersed in the staining solution for 10 min, and then washed again. The slices were then dehydrated, cleared in xylene three times for 4 min, mounted with Eukitt Quick-hardening mounting medium (Sigma, 03989), and imaged by light microscopy with Z-stack. Image processing was performed using Neurolucida360 software in 3D analysis.

## Whole-embryo immunostaining and clearance

Mouse embryos were fixed in 4% PFA overnight at 4°C and washed three times in PBS. Then, the embryos were permeabilized in PBST (PBS containing 1% Triton X-100) for 2 hr at RT, incubated in blocking solution (75% PBST, 20% dimethyl sulfoxide, 5% normal goat serum) for 3 hr at RT, and incubated in anti-CD31 (1:500, company name) diluted in the blocking solution at RT. After 3 d, the embryos were washed with PBST for 8 hr and incubated with Alexa Fluor 488-conjugated secondary antibody (1:500, Thermo) for 2 d at RT. After washing again with PBST for 8 hr, the embryos were cleared with ethyl cinnamate (ECi; Cat# 112372, Sigma) according to a previous protocol (*Klingberg et al., 2017*). Briefly, the embryos were subjected to serial dehydration with ethanol (EtOH) (30, 50, and 70% EtOH, pH 9.0 with 2% Tween 20, followed by twice 100% EtOH). The solutions were changed every 12 hr and incubated at 4°C. The samples were transferred to ECi and incubated at RT with gentle shaking until they became transparent, and cleared samples were stored in ECi at RT until imaging. All clearing procedures were performed in the dark. A Dragonfly 502w (Andor Technology) was used for imaging, and Imaris x64 9.6.1 (Bitplane) was used for image reconstruction.

## DiI injection and imaging

For the tracing of the neural projection, small crystals of DiI (1.1-dioctadecyl-3,3,3,3-tetramethyl-indocarbocyanine perchlorate, Sigma) were inserted into the thalamus of an E16.5 mouse brain fixed in 4%

PFA overnight and sealed with 2% agarose melt in 1× PBS. Then, the brain was incubated in 4% PFA at 37°C for 2 wk and divided into 100 μm thick sections by a vibratome (Leica VT200S). Serial brain slices were immediately collected, and DiI-stained sections were imaged using a fluorescence microscope (Nikon Eclipse Ti-U). To sparsely label the striatonigral projection, DiI crystals were injected into the dorsal striatum of P5 mouse brain and incubated for a month or so under the same conditions as described above. Afterward, the tissue was sectioned into 100-μm-thick slices using a vibratome (Leica VT200S). Z-stack images were obtained by using Leica TCS SP8 Confocal Microscope, and the 3-D images were reconstructed utilizing Leica Application Suite X, an image analysis software.

## Striatonigral projection analysis

For the analysis of striatonigral projections, P5 or P30 brains from wild-type (*Drd1a-tdT; Mtss1^{f/f}*), *Mtss1* cKO (*Drd1a-tdT; Nes-cre; Mtss1^{f/f}*), or *Plxnd1* cKO (*Drd1a-tdT; Nes-cre; Plxnd1^{f/f}*) mice were fixed in 4% PFA overnight and embedded in a 4% agarose block melt in PBS after being washed with PBS three times. Then, the areas of interest in the brain were divided into 100-μm-thick sections by a vibratome (Leica VT200S). Serial brain slices were immediately collected and mounted with Prolong Diamond antifade solution containing DAPI (P36962, Thermo). The sections were imaged with a fluorescence microscope (Nikon Eclipse Ti-U). To analyze the guidance phenotype in the striatonigral projections, we quantified the number of axon bundles intersecting with others between the globus pallidus (Gp) and the entopeduncular nucleus (EP). Since some mutants within the same litter occasionally displayed significant developmental delays, we selectively chose samples from mutant littermates with ±5% body weight variance for a reliable phenotypic assessment.

## Quantification and statistical analysis

The estimate of variance was determined by the standard error of the mean (SEM), and statistical significance was set at $p<0.05$. All data were tested with a Gaussian distribution using the Shapiro–Wilk test before statistical analysis. Pairwise comparisons were performed using the two-tailed Student's *t*-test or Mann–Whitney test, and multiple-group analyses were conducted with one-way or two-way ANOVA with Tukey's or Bonferroni's multiple comparisons test or Kruskal–Wallis test with Dunn's multiple comparisons test. For the growth cone collapse assay, the $\chi^2$ test was used as previously reported (*Burk et al., 2017*). For the quantification of colocalization, the images were analyzed using the Jacop plugin in ImageJ (National Institutes of Health, Bethesda, MD). Statistical data on colocalization were obtained using Costes' randomization based on the colocalization module, as previously described (*Bolte and Cordelières, 2006*). Statistical analyses were performed with Prism 9 (GraphPad Software). At least three pairs of mice were used per experiment for all histological analyses. For the quantification of image data, at least three brain sections per animal were collected and analyzed. All data analyses were performed by an investigator blinded to the groups. No statistical methods were used to predetermine sample sizes, but our sample sizes were similar to those generally employed in the field.

## Acknowledgements

We thank Drs. Ayal Ben-Zvi, Soonmoon Yoo, and Chenghua Gu for reading the manuscript and providing critical advice; Dr. Chenghua Gu for providing *Sema3e* and *Plxnd1-flox* mice; Drs. Mineko Kengaku and Masayoshi Mishina for providing *Mtss1-flox* mice; Juhyun Lee for helping with quantification; the Advanced Neural Imaging Center in KBRI for image analysis. This research was supported by the KBRI basic research program of the Korea Brain Research Institute funded by the Ministry of Science and ICT (KBRI 23-BR-01-02), the National Research Foundation (NRF) funded by the Korean government (NRF-2014R1A1A2058234), the Bio & Medical Technology Development Program of the NRF & funded by the Korean government (MSIT) (NRF-2020M3E5D9079766 and NRF-2022M3E5E8017701) to WO, and the Young Researcher Program of the National Research Foundation (NRF) funded by the Korean government (MSIT) (2020R1C1C1010509) to NK.

## Additional information

### Funding

| Funder | Grant reference number | Author |
|---|---|---|
| Korea Brain Research Institute | KBRI 23-BR-01-02 | Won-Jong Oh |
| National Research Foundation of Korea | NRF-2014R1A1A2058234 | Won-Jong Oh |
| National Research Foundation of Korea | NRF-2020M3E5D9079766 | Won-Jong Oh |
| National Research Foundation of Korea | NRF-2022M3E5E8017701 | Won-Jong Oh |
| National Research Foundation | Young Researcher Program 2020R1C1C1010509 | Namsuk Kim |

The funders had no role in study design, data collection and interpretation, or the decision to submit the work for publication.

### Author contributions

Namsuk Kim, Data curation, Software, Formal analysis, Funding acquisition, Validation, Investigation, Visualization, Methodology, Writing - review and editing; Yan Li, Data curation, Software, Formal analysis, Validation, Investigation, Visualization, Methodology; Ri Yu, Data curation, Formal analysis, Validation, Visualization, Methodology; Hyo-Shin Kwon, Data curation, Formal analysis, Visualization, Methodology; Anji Song, Data curation, Formal analysis, Visualization; Mi-Hee Jun, Ji Hyun Lee, Data curation; Jin-Young Jeong, Data curation, Visualization; Hyun-Ho Lim, Data curation, Writing - review and editing; Mi-Jin Kim, Resources; Jung-Woong Kim, Resources, Data curation, Visualization, Writing - review and editing; Won-Jong Oh, Conceptualization, Resources, Data curation, Software, Formal analysis, Supervision, Funding acquisition, Validation, Investigation, Visualization, Methodology, Writing - original draft, Project administration, Writing - review and editing

### Author ORCIDs

Namsuk Kim http://orcid.org/0000-0001-5043-0293
Yan Li http://orcid.org/0000-0001-5276-1053
Hyun-Ho Lim http://orcid.org/0000-0002-5477-5640
Won-Jong Oh http://orcid.org/0000-0001-8867-7814

### Ethics

All protocols for animal experiments were approved by the Institutional Animal Care and Use Committee of Korea Brain Research Institute (IACUC-18-00008, 20-00012). All experiments were performed according to the National Institutes of Health Guide for the Care and Use of Laboratory Animals and ARRIVE guidelines.

### Decision letter and Author response

Decision letter https://doi.org/10.7554/eLife.96891.sa1
Author response https://doi.org/10.7554/eLife.96891.sa2

## Additional files

### Supplementary files
• MDAR checklist

### Data availability

The accession number for the RNA-Seq data reported in the present study is GSE196558.

The following datasets were generated:

| Author(s) | Year | Dataset title | Dataset URL | Database and Identifier |
|-----------|------|---------------|-------------|-------------------------|
| W-J Oh | 2022 | Axon guidance signal ensures neurite growth pace while sensitizing repulsive cues through induction of a dual function facilitator | https://www.ncbi.nlm.nih.gov/geo/query/acc.cgi?acc=GSE196558 | NCBI Gene Expression Omnibus, GSE196558 |
| W-J Oh | 2023 | Axon guidance signal ensures neurite growth pace while sensitizing repulsive cues through induction of a dual function facilitator | https://www.mousemine.org/mousemine/report.do?id=89860228&trail=%7c89860228 | MouseMine, GSE196558 |

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

# Appendix 1

## Appendix 1—key resources table

| Reagent type (species) or resource | Designation | Source or reference | Identifiers | Additional information |
|---|---|---|---|---|
| Antibody | Anti-Mtss1 (rabbit polyclonal) | Novus Biologicals | Cat# NBP2-24716; RRID:AB_2716709 | IHC (1:500) WB (1:500) |
| Antibody | Anti-Plexin-D1 (goat polyclonal) | R&D Systems | Cat# AF4160; RRID:AB_2237261 | IHC (1:500) |
| Antibody | Anti-Tau (goat polyclonal) | Santa Cruz | Cat# sc-1995; RRID:AB_632467 | ICC (1:500) |
| Antibody | Anti-Neurofilament (mouse) | Hybridoma Bank | Cat# 2H3; RRID:AB_ 531793 | IHC (1:500) |
| Antibody | Anti-β-actin/HRP (rabbit monoclonal) | Cell Signaling Technology | Cat# 5125S; RRID:AB_1903890 | WB (1:5000) |
| Antibody | Anti-Myc (mouse monoclonal) | Cell Signaling Technology | Cat# 2276; RRID:AB_331783 | WB (1:1000) |
| Antibody | Anti-Vsv (goat polyclonal) | Abcam | Cat# ab3861; RRID:AB_304118 | WB (1:1000) |
| Antibody | Anti-Sema3E (human polyclonal) | LSBio | Cat# LS-c353198 | WB (1:500) |
| Antibody | Anti-Phospho-Akt (rabbit polyclonal) | Cell Signaling Technology | Cat# 9271; RRID:AB_329825 | WB (1:1000) |
| Antibody | Anti-Akt (rabbit polyclonal) | Cell Signaling Technology | Cat# 9272; RRID:AB_ 329827 | WB (1:1000) |
| Antibody | Anti-RFP (rabbit polyclonal) | Abcam | Cat# ab62341; RRID:AB_945213 | IHC (1:1000) |
| Antibody | Anti-RFP (mouse monoclonal) | Thermo Fisher Scientific | Cat# MA5-15257; RRID:AB_10999796 | IHC (1:1000) |
| Antibody | Anti-alpha-tubulin (mouse monoclonal) | Sigma-Aldrich | Cat# T5168; RRID:AB_477579 | IHC (1:1000) |
| Antibody | Anti-cleaved caspase3 (rabbit polyclonal) | Cell Signaling Technology | Cat# 9661; RRID:AB_2341188 | IHC (1:1000) |
| Antibody | Anti-CD31 (rat monoclonal) | BD Bioscience | Cat# 553370; RRID:AB_394816 | IHC (1:500) |
| Antibody | Anti-digoxigenin-alkaline phosphatase (sheep polyclonal) | Roche | Cat# 11093274910; RRID:AB_2313640 | In situ (1:3000) |
| Antibody | Anti-mouse IgG/HRP (goat polyclonal) | Thermo Fisher Scientific | Cat# 31430; RRID:AB_228307 | WB (1:10,000) |
| Antibody | Donkey anti-rabbit IgG/HRP (rabbit polyclonal) | Jackson Immuno Research | Cat# 711-035-152; RRID:AB_10015282 | WB (1:10,000) |
| Antibody | Donkey anti-goat IgG/HRP (goat polyclonal) | Jackson Immuno Research | Cat# 705-035-147; RRID:AB_2313587 | WB (1:10,000) |
| Antibody | Donkey anti-rabbit IgG, Alexa Fluor 488 | Thermo Fisher Scientific | Cat# A-21206; RRID:AB_2535792 | ICC (1:1000) IHC (1:1000) |
| Antibody | Donkey anti-mouse IgG, Alexa Fluor 488 | Thermo Fisher Scientific | Cat# A-21202; RRID:AB_141607 | ICC (1:1000) IHC (1:1000) |
| Antibody | Donkey anti-goat IgG, Alexa Fluor 568 | Thermo Fisher Scientific | Cat# A-11057; RRID:AB_142581 | ICC (1:1000) IHC (1:1000) |
| Antibody | Donkey anti-rabbit IgG, Alexa Fluor 568 | Thermo Fisher Scientific | Cat# A-10042; RRID:AB_2534017 | ICC (1:1000) IHC (1:1000) |

*Appendix 1 Continued on next page*

*Appendix 1 Continued*

| Reagent type (species) or resource | Designation | Source or reference | Identifiers | Additional information |
|---|---|---|---|---|
| Antibody | Donkey anti-mouse IgG, Alexa Fluor 568 | Thermo Fisher Scientific | Cat# A-10037; RRID:AB_2757558 | ICC (1:1000) IHC (1:1000) |
| Antibody | Donkey anti-mouse IgG, Alexa Fluor 647 | Thermo Fisher Scientific | Cat# A-31571; RRID:AB_162542 | ICC (1:1000) IHC (1:1000) |
| Chemical compound, drug | MK2206 | SelleckChem | Cat# S1078 | |
| Chemical compound, drug | TRIzolTM Reagent | Thermo Fisher Scientific | Cat# 15596026 | |
| Chemical compound, drug | RNasin Ribonuclease Inhibitor | Promega | Cat# N2115 | |
| Chemical compound, drug | Halt Protease and Phosphatase Inhibitor Cocktail | Thermo Fisher Scientific | Cat# 78444 | |
| Chemical compound, drug | SuperSignal West Pico PLUS Chemiluminescent Substrate | Thermo Fisher Scientific | Cat# 34580 | |
| Chemical compound, drug | SuperSignal West Femto Maximum Sensitivity Substrate | Thermo Fisher Scientific | Cat# 34096 | |
| Chemical compound, drug | ProLong Diamond Antifade Mountant with DAPI | Thermo Fisher Scientific | Cat# P36962 | |
| Chemical compound, drug | Eukitt Quick-hardening mounting medium | Sigma-Aldrich | Cat# 03989 | |
| Chemical compound, drug | Alexa Fluor 488 Phalloidin | Thermo Fisher Scientific | Cat# A12379 | |
| Chemical compound, drug | Alexa Fluor 568 Phalloidin | Thermo Fisher Scientific | Cat# A12380 | |
| Chemical compound, drug | Alexa Fluor 647 Phalloidin | Thermo Fisher Scientific | Cat# A22287 | |
| Chemical compound, drug | Gibco DMEM, high glucose, pyruvate | Gibco | Cat# 11995-065 | |
| Chemical compound, drug | Penicillin-Streptomycin | HyClone | Cat# SV30010 | |
| Chemical compound, drug | Paraformaldehyde | Electron Microscopy Sciences | Cat# 19202 | |
| Chemical compound, drug | Poly-D-lysine hydrobromide | Sigma-Aldrich | Cat# P6407 | |
| Chemical compound, drug | Corning Laminin | Corning | Cat# 354232 | |
| Chemical compound, drug | NBT/BCIP Ready-to-Use Tablets | Roche | Cat# 11697471001 | |
| Chemical compound, drug | DiI (1.1-dioctadecyl-3,3,3,3-tetramethylindocarbocyanine perchlorate) | Sigma-Aldrich | Cat# 468495 | |
| Commercial assay or kit | QuantiTect Reverse Transcription kit | QIAGEN | Cat# 205313 | |
| Commercial assay or kit | Pierce BCA Protein Assay Kit | Thermo Fisher Scientific | Cat# 23225 | |
| Commercial assay or kit | Lipofectamine 2000 Transfection Reagent | Thermo Fisher Scientific | Cat# 11668019 | |
| Commercial assay or kit | Basic Nucleofector Kit | LONZA | Cat# VAPI-1003 | |
| Commercial assay or kit | FD Rapid GolgiStain Kit | FD Neurotechnologies Inc | Cat# PK401A | |
| Cell line (*Homo sapiens*) | HEK293T Kidney (embryo) | ATCC | CRL-3216; RRID:CVCL_0063 | |

*Appendix 1 Continued on next page*

*Appendix 1 Continued*

| Reagent type (species) or resource | Designation | Source or reference | Identifiers | Additional information |
|---|---|---|---|---|
| Cell line (*Cercopithecus aethiops*) | COS7 Kidney | Korean Cell Line Bank | Cat# 21651; RRID:CVCL_0224 | |
| Cell line (*H. sapiens*) | HUVEC Umbilical Vein Endothelial Cells | Lonza | CC-2935; RRID:CVCL_2959 | |
| Cell line (*H. sapiens*) | HCMEC/D3 Human temporal lobe microvessels | Millipore | SCC066; RRID:CVCL_U985 | |
| Strains | Mouse: C57BL/6J | The Jackson Laboratory | Stock# 000664; RRID:IMSR_JAX:000664 | |
| Strains | Mouse: Nestin-Cre | The Jackson Laboratory | Stock# 003771; RRID:IMSR_JAX:003771 | |
| Strains | Mouse: Tie2-Cre | The Jackson Laboratory | Stock# 008863; RRID:IMSR_JAX:008863 | |
| Strains | Mouse: Drd1a-tdTomato | The Jackson Laboratory | Stock# 016204; RRID:IMSR_JAX:016204 | |
| Strains | Mouse: *Mtss1flox/+* | Center for Animal Resources and Development Database (CARD) under permission of Dr. Mineko Kengaku | Card ID#2760 | |
| Strains | Mouse: *Plxnd1flox/flox* | Obtained from Dr. Chenghua Gu | *Kim et al., 2011* | |
| Strains | Mouse: *Sema3e+/-* | Obtained from Dr. Chenghua Gu | *Chauvet et al., 2007* | |
| Software, algorithm | ImageJ | NIH | https://imagej.nih.gov/ij/ | |
| Software, algorithm | Prism 9 | GraphPad | https://www.graphpad.com/scientific-Software/prism/ | |
| Software, algorithm | Image Lab (v5.2.1) | Bio-Rad | https://www.bio-rad.com/ | |
| Software, algorithm | Fusion FX | Vilber | https://www.vilber.com/fusion-fx/ | |
| Software, algorithm | LightCycler480 (v1.5.1) | Roche | https://lifescience.roche.com/ | |
| Software, algorithm | Leica Application Suite X | Leica | https://www.leicamicrosystems.com/ | |
| Software, algorithm | NIS-Elements AR (v4.51.00) | Nickon | https://www.microscope.healthcare.nikon.com/ | |
| Software, algorithm | NIS-Elements (v4.50.00) | Nickon | https://www.microscope.healthcare.nikon.com/ | |
| Software, algorithm | AIVIA | Aivia, Inc | https://www.aivia-Software.com/ | |
| Other | Immobilon-P PVDF Membrane | Merck | Cat# IPVH00010 | |
| Other | RNA-seq (P5 mice, striatum) | Data and code availability section in this paper | GEO: GSE196558 | |

