## [Editor Report]

In this manuscript, the authors proposed a novel and attractive model to address a fundamental question of how the locational and function of axon guidance molecules are regulated. They presented convincing data to support their working model. They showed important findings that Sema3E-Plexin-D1 signaling regulates the expression of Mtss1, which regulates the localization of Plexin-D1 and contributes to striatonigral axonal growth and turning.

---

## [Decision Letter]

**Decision letter after peer review:**

[Editors’ note: the authors submitted for reconsideration following the decision after peer review. What follows is the decision letter after the first round of review.]

Thank you for submitting the paper "Repulsive Sema3E-Plexin-D1 signaling coordinates both axonal extension and steering via activating an autoregulatory factor, Mtss1" for consideration by *eLife*. Your article has been reviewed by 3 peer reviewers, and the evaluation has been overseen by a Reviewing Editor and a Senior Editor. The following individuals involved in review of your submission have agreed to reveal their identity: Hisashi Umemori (Reviewer #2); Pirta Hotulainen (Reviewer #3).

Comments to the Authors:

We are sorry to say that, after consultation with the reviewers, we have decided that this work will not be considered further for publication by *eLife*, at least in the present form.

Specifically, all three reviewers agree that this study is interesting with regard to the regulation of Mtss1 mRNA expression by the presence of PlexD1, and the requirement for Mtss1 to mediate Sema3E repulsive signaling in neurons in vitro. However, mechanistic insight is lacking. In particular, two key questions remain unanswered: how does Sema3E-PlexD1 regulate Mtss1 translation? What the actual functional link is between Mtss1 and Sema3E/PlexD1 signaling? If you are able to address these issues, we would be willing to reconsider the story in the form of a new submission.

*Reviewer #1 (Recommendations for the authors):*

In this study Kim et al. investigate mechanisms of axon extension and repulsive guidance mediated by signaling involving the secreted semaphorin 3E (Sema3E), its receptor plexin D1 (PlexD1), and the cytosolic membrane remodeling protein metastasis suppressor 1 (Mtss1). Sema3E/PlexD1 signaling has been shown to mediate classical repulsive guidance in developing CNS neurons and vasculature, and also select synapse formation for direct-pathway medium spina neurons (MSNs) that project to the substantia nigra (SNr). Here, the authors show using bulk transcriptomic profiling in striatal tissue that in PlexD1-/- mutants there is a dramatic down regulation of Mtss1 mRNA expression, and this can be observed in the striatum in situ and also when assessing Mtss1 protein biochemically. In heterologous non-neuronal cells in culture the authors show that the PlexD1 cytoplasmic domain and also the Mtss1 I-BAR domain are required for a direct interaction between these two proteins in a Sema3E-independent manner (similar assessment of these interactions in neurons in vivo or culture is not presented). In COS cells it is observed that co-expression of PlexD1 and Mtss1 leads to protrusive shape changes with co-localization of these two proteins that appears in part dependent upon the Mtss1 BAR domain-similar co-localization is seen in striatal neurons in culture. Importantly, Mtss1 is shown to be required for Sema3E neuronal growth cone collapse, and also for normal neurite outgrowth. Significant but modest defects in neuronal pathways reaching the SNr are presented in Mtss1 mutants, and these are shown in the context of the striatonigral pathway to be similar to what is observed in PlexD1 mutants.

Overall, this study is interesting with regard to regulation of Mtss1 mRNA expression by the presence of PlexD1, and the requirement for Mtss1 to mediate Sema3E repulsive signaling in neurons in vitro. However, we do not learn whether or not Mtss1 functions more generally in PlexD1-expressing cells that respond to Sema3E (i.e. vasculature), what the mechanism of action is regarding how PlexD1 regulates Mtss1 translation, or what the actual functional link is between Mtss1 and Sema3E/PlexD1 signaling-much of these data presented here are correlative.

1. Figure 1- Are similar effects on Mtss1 expression seen in Sema3E mutants? This is key, regardless of the result, in order to inform conclusions relating to actual activation of PlexD1 signaling with regulating of Mtss1 expression. Further, is there evidence directly showing that in neurons stimulated with Sema3E, Mtss1 expression is increased and, as a control, that this is not observed in Plxnd1-KO neurons. In Figure 1H, is Mtss1 present only in neuronal bodies? And what is the Mtss1 expression pattern in theSNr? Also, is there an explanation for why at P7 (Figure 1G) Mtss1 decreases while PlxnD1 is still elevated?

2. Page 6 – The claim that failure to see elevated Mtss1 expression in cultured striatal neurons shows that Mtss1 expression is regulated in a "cell autonomous manner" is not supported by this experiment since non-cell-autonomous interactions among these neurons in culture are not ruled out using this approach.

3. Figure 2 – Are any of these interactions assessed at the level of Western analysis between PlexD1 and Mtss1 observed in neurons, either in culture or in vivo? Further, if PlexD1 loss-of-function (LOF) leads to loss of Mtss1 expression, are the direct interactions between PlexD1 and Mtss1 in any way required to regulate Mtss1 expression (either in heterologous cell culture or in neurons)? Indeed, is Mtss1 mRNA expression regulated by Sema3E/PlexD1 in COS cells? Further, given results from others involving neuropilin-1 (Nrp1) in PlexD1-medated CNS axon guidance (Chauvet et al., 2007), has this co-receptor component been explored in the context of these interactions with Msst1 or Sema3E-dependence for them (or for that matter any other aspects of Mtss1 function in subsequent experiments in this study)?

4. Figure 3 – Does Sema3E affect the co-localization of PlexD1 and Mtss1 observed in COS cells? Further, the statement on page 8 "….These results confirm that the Plexin-D1 and Mtss1proteins form a complex in specialized cell structures such as filopodia…" is not supported by these imaging data-at best this modest co-localization is correlative.

5. Figure 4 – These data show a degree of co-localization of overexpressed PlexD1 and Mtss1 in Mtss1-/- neurons and the strong result that Mtss1 is required for Sema3E collapse. However, the conclusion that somehow PlexD1membrane targeting is the mechanism that mediates this effect is not strongly supported by these data. Does loss of Mtss1 affect membrane targeting of exogenous and also endogenous PlexD1? Can a transmembrane domain-containing PlexD1 protein targeted to the membrane affect neuronal morphology independent of Mtss1? Also, does the Mtss1 Bar-domain deletion construct fail to rescue Sema3E growth cone collapse in these neurons in vitro? Finally, what is the link to PlexD1 signaling with regard to effects of Mtss1 on neurite outgrowth? Clearly loss of Mtss1 does affect neurite outgrowth, but this is not directly linked in this study to PlexD1 and appears to be independent of Plexin signaling.

Also, what are the growth rates and growth cone responses at DIV6? Based on the Western blos from Figure 1L, there is a large difference between expression of Plxnd1/Mtss1 at DIV3 (when Sema3E-treatments were done in Figures 4H and Suppl. Figure 5) and at DIV 6.

6. Figure 5 – I am not convinced that there is reduced PlexD1 localization on axon trajectories-Sema3E-AP binding is reduced, however so is the trajectory be assessed, so in the absence of a ratiometric assessment of PlexD1 (as evidenced by Sema3E-AP binding) this is not convincing. Further, the statement "….Since no significant change in Plexin-D1 levels was observed in the striatum of Mtss1-deficient mice compared to those in littermate controls…" is not supported by bulk Western analysis of PlexD1 protein since no neuronal localization information is provided by these data.

7. Figure 6 – Have the authors looked in cross section at the Drd1a-tdT-labeled striatonigral pathways in order to provide data supporting the schematic in panel L? This could include higher resolution assessment of axon numbers in this pathway, which would support the overall conclusions regarding guidance effects mediated by Mtss1.

8. Have the authors attempted to generalize their analysis of Mtss1 function to the developing vasculature? Given that the strongest results here relate to regulation of Mtss1 mRNA expression by PlexD1 and also the requirement for Mtss1 for neuronal growth cone collapse, generalizing Mtss1 function to other PlexD1-expressing tissues known to respond to Sema3E would strengthen this study.

9. The authors are urged to review work in the guidance field involving the regulation of transcription by guidance cue receptors so that statements in the Discussion can be brought in line with work in this area (for example, netrin-mediated DCC signaling to the nucleus that regulates commissureless expression-see Russell and Bashaw, 2019, for a review).

*Reviewer #2 (Recommendations for the authors):*

Kim, Li et al. identified Mtss1 as a molecule regulated by Sema3E-Plexin-D1 signaling and investigated its roles in striatonigral axon growth and turning. They first showed that Sema3E-Plexin-D1 signaling induces Mtss1 expression in striatonigral projecting neurons. They then showed that Mtss1 physically interacts with Plexin-D1 and that Plexin-D1 localization at the axon growth cone appears to be perturbed in the absence of Mtss1. Finally, the authors showed that striatonigral axonal projections are impaired in Mtss1 KO mice, an outcome that is phenocopied in Plexin-D1 KO mice. With these data, the authors propose that Mtss1 is upregulated in response to Sema3E, possibly from the thalamostriatal neurons, through Plexin-D1, then interacts with Plexin-D1, facilitates the transport of Plexin-D1 to the growth cone, and sensitizes striatal axons to Sema3E for axonal repulsion, while contributing to axonal extension. This is a novel and exciting model proposing how axonal growth and pathfinding are regulated in the thalamo-striato-nigral circuit. However, additional data would help fully support their model. Furthermore, images and quantifications could be improved, and quantification methods could be clearly described. Specific points are listed below.

1. An interesting aspect of the proposed model is that Sema3E from the thalamostriatal neurons upregulates Mtss1 expression in striatonigral neurons. However, currently, the paper only includes the expression pattern of Sema3E. The authors could experimentally show the role of thalamic Sema3E.

2. Another interesting aspect of the proposed model is that Mtss1 binds to and transports Plexin-D1 to the growth cone. While the authors' data suggest that Plexin-D1 in the growth cone appears to be altered in the absence of Mtss1, the authors could demonstrate that indeed Mtss1 and Plexin-D1 are co-trafficked along axons.

3. The quality of images could be improved. Also, quantifications are missing for several figures. Furthermore, where quantifications are done, the methods used for the quantifications are not clearly described in the methods.

4. The discussion can be expanded. The authors may discuss more about their proposed model (Supplementary Figure 7).

Specific suggestions are listed below.

1. An interesting aspect of the proposed model is that Sema3E from the thalamostriatal neurons upregulates Mtss1 expression in striatonigral neurons. Regarding this point, the authors could inactivate Sema3E from thalamostriatal neurons and demonstrate the effect on Mtss1 expression and striatonigral axonal pathfinding.

2. Another interesting aspect of the proposed model is that Mtss1 binds to and transports Plexin-D1 to the growth cone. Regarding this point, the authors could demonstrate that indeed Mtss1 and Plexin-D1 are co-trafficked along axons, ideally with live imaging.

3. The quality of images should be improved. It is difficult to evaluate colocalization (e.g., Figure 3C, 4A), membrane localization (e.g., Figure 1H), axonal growth and pathfinding from the images presented. Also, quantifications are missing for several figures (including staining and Western blotting). Please quantify (e.g., Figure 1G, 1H, 2). Furthermore, where quantifications are done, the methods used for the quantifications are not clearly described in the methods. Please describe the methods, including the methods of normalization and verification of reproducibility (e.g., Figure 3D, 4B, 4C, 5-7).

Related to this point, Pearson's R may not be sufficient to establish colocalization as it could be influenced by the amount of protein expression. The authors may perform additional colocalization analysis, for example, by using a pixel scrambled image and demonstrating that the colocalization is not random. Also, the authors may consider quantifying the percent of Mtss1 that is positive for Plexin-D1 and vice versa.

4. The discussion can be expanded. The authors may discuss more about their proposed model (points #1 to #4 in Supplementary Figure 7), including the mechanistic insights into the roles of Mtss1 in axonal turning vs. growing.

Additional comments:

1. The term "cell-autonomous" may not be used appropriately by the authors in the paper. Since the authors propose that Sema3E from the thalamus activates Plexin-D1 to upregulate Mtss1, this does not appear cell-autonomous.

2. Plexin-D1-Mtss1 binding in a cell line may involve other molecules. Hence the binding could be indirect.

3. In Figure 4, if Plexin-D1 is not localizing to the growth cone, then where is it localized? The authors could include a lower magnification image showing the cell body and axon and quantify Plexin-D1 localization in these cellular compartments.

4. The use of AP-Sema3E is an established way of detecting Plexin-D1 localization. However, since it is expected that Mtss1 KO mice would have defects in both Plexin-D1 expression/localization and axon projections, the Sema3E binding may not be sufficient to conclude both "poor neuronal projection" and "reduced Plexin-D1 localization". Additionally, here, too, the authors need to explain how the quantifications were done to determine the Plexin-D1 positive path (%).

5. In Supplementary Figure 4E, Mtss1deltaWH2-myc seems to have much weaker effects. Since this construct contains the I-BAR domain, the authors may want to add some explanation.

*Reviewer #3 (Recommendations for the authors):*

In the developing nervous system, the axons of newly generated neurons extend toward destination targets following an exquisitely designed program. Axon guidance molecules are critical for neuronal pathfinding because they regulate both directionality and growth pace. This study describes a novel role for a Mtss1 in axon guidance. In general this is a good study but as Mtss1 has not been found earlier to be expressed in axons, expression and localization of endogenous Mtss1 in axons should be shown convincingly.

I think that authors convincingly show that Sema3E-Plexin-D1signaling regulates Mtss1 expression in projecting striatonigral neurons. Also Plexin-D1 – Mtss1 interaction seems clear as well. I like the idea that Plexin-D1 brings Mtss1 to filopodia where Plexin-D1 can bind Sema3E. However, the results presented for this idea were not convincing. First of all, expression and localization of endogenous Mtss1 should be shown. This is especially important because we did not find Mtss1 from axons (Saarikangas et al., Dev Cell 2015, Supplementary figure 2). There can be many reasons for this, age or cell type, for example, but due to this controversy, expression and localization of endogenous Mtss1 in axons must be shown convincingly.

In Figure 4, overexpressed Mtss1 localization is not supporting the idea of bringing Plexin-D1 to filopodia and plasma membrane. To me this localization looks quite strange. I am not sure what to do with this but maybe localization of endogenous Mtss1 will help here. Maybe overexpressed construct is not folding right? Is it dynamic? Actin looks strange in this cell as well (Figure 4A) (F-actin in the middle, where are filopodia, arcs, lamellipodia?).

Furthermore, the text says on line 195 that "Mtss1 targeting of Plexin-D1 to the growth cone is critical for robust Sem3E-induced repulsive signaling." For this, it would be good to show Plexin-D1 localization (endogenous) in Mtss1 cKO cells vs. control cells. Is there a change? Mtss1 overexpression seems to keep Plexin-D in the middle of growth cone rather than bringing Plexin-D to the filopodia and plasma membrane (Figure 4A). Plexin-D localization without Mtss1 overexpression is missing (is there a change in localization?).

Text says on lines 218-221 that "In mice expressing wild-type Mtss1 showed a significant level of Plexin-D1 in the neuronal pathway reaching the substantia nigra, whereas Mtss1-knockout mice exhibited poor neuronal projection and reduced Plexin-D1 localization on E17.5 (Figure 5A).

I have struggled by myself in measuring dendrite length in vivo and therefore I ask, how it was ensured that slices are from same depth and same cutting angle? If they vary, does it affect on results?

These are the critical issues. If Mtss1 is not expressed in axons (just as a possibility), is it possible that it affects axons by other mechanisms?

---

## [Author Response]

[Editors’ note: the authors resubmitted a revised version of the paper for consideration. What follows is the authors’ response to the first round of review.]

Comments to the Authors:We are sorry to say that, after consultation with the reviewers, we have decided that this work will not be considered further for publication by eLife, at least in the present form.Specifically, all three reviewers agree that this study is interesting with regard to the regulation of Mtss1 mRNA expression by the presence of PlexD1, and the requirement for Mtss1 to mediate Sema3E repulsive signaling in neurons in vitro. However, mechanistic insight is lacking. In particular, two key questions remain unanswered: how does Sema3E-PlexD1 regulate Mtss1 translation? What the actual functional link is between Mtss1 and Sema3E/PlexD1 signaling? If you are able to address these issues, we would be willing to reconsider the story in the form of a new submission.

We appreciate the editors and reviewers for their valuable input on the manuscript we generated in this study. We have incorporated new data as suggested by the editors and reviewers and have carefully addressed their comments in the new manuscript. Thank you for your time and effort in helping us improve the quality of our research.

Key Question 1. how does Sema3E-PlexD1 regulate Mtss1 translation?

First, to support the Mtss1 expression through Sema3E-Plexin-D1 signaling, we examine Mtss1 levels by adding exogenous Sema3E ligand directly to the cultured Sema3e-deficient striatal neurons at DIV2, prior to in vitro neuronal connections according to the scheme in Figure 2E. Sema3E administration increases Mtss1 expression in cultured MSNs from Sema3e-null striatum, but not from Plxnd1 knockout mice (Figures 2E-2I).

We also examined the expression of Mtss1 after treatment with an Akt inhibitor, MK2206 because Akt is already known to mediate the Sema3E-Plexin-D1 signaling cascade in neurons (Burk et al., 2017). We found that the inhibition of Akt signaling attenuated the induction of Mtss1 expression by Sema3E-Plexin-D1 signaling in the cultured MSNs (Figures 2J-2M). Additionally, MK2206 treatment attenuated the increased Mtss1 expression caused by the addition of exogenous Sema3E in the Sema3e-knockout neurons (Figures 2N-2O), suggesting that Sema3E-Plexin-D1 pathway is involved in Mtss1 expression through Akt signaling.

We are highly intrigued by the prospect of deciphering the molecular mechanism through which these repulsive axon guidance molecules mediate specific signaling cascades to regulate gene expression profiles in the future investigations. To pursue this research, we will need to identify an appropriate homogeneous cellular model and conduct in-depth gene regulation studies. We discussed other possibilities in the revised manuscript (Lines 384-417).

Key Question 2. What the actual functional link is between Mtss1 and Sema3E/PlexD1 signaling?

To explore the functional link between Mtss1 and Sema3E-Plexin-D1 signaling, we performed a few more experiments. First, we conducted the rescue experiments by overexpressing Mtss1 in the cultured Plxnd1 KO MSNs. As shown in Figures 3K-3L, Mtss1 overexpression rescued the growth reduction phenotype in Plxnd1 KO MSNs, suggesting that Sema3E-Plexin-D1 signaling regulates axonal growth through Mtss1.

Second, we have shown that Mtss1 is not only involved in axonal projections, but also in Plexin-D1 trafficking in navigation of striatonigral projections. To investigate the functional link in Plexin-D1 trafficking, we performed an AP-Sema3E binding assay to label the endogenous Plexin-D1 in the growth cones of cultured Drd1a-positive MSNs at DIV6 derived from WT or Mtss1 cKO mice. The data show that the intensity of Plexin-D1 was significantly decreased in Mtss1-deficient MSNs compared to WT, indicating that the absence of Mtss1 disrupts the normal trafficking of Plexin-D1 to the growth cones in cultured MSNs (Figures 5L-5M). To further investigate the role of Mtss1 in Plexin-D1 trafficking, we performed the time-lapse live imaging in cultured MSNs from WT or Mtss1 cKO mice. As shown in Figures 5G-5K and Videos 1-4, the dynamic transport of Plexin-D1-containing vesicles was defective in Mtss1-deficient MSNs. These data provide strong evidence that Mtss1 regulates Plexin-D1 trafficking, and its localization in combination with the co-localization analysis in Figures 5A-5F.

Third, using ratiometric assessment of Plexin-D1 localization in Mtss1-KO mice, we found in vivo evidence that despite reduced neuronal projections in the Mtss1-deficient MSNs, Plexin-D1 localization in striatonigral projections was more severely reduced in Mtss1-KO mice (Figures 7A-7E and 7N-7Q). These findings suggest that Mtss1 plays a role in facilitating the dynamic transport of Plexin-D1 along the growing neurites of direct-pathway MSNs, leading to an increased rate of Plexin-D1 localization in growing cones (Figures 5, 7A-7E, and 7N-7Q).

Fourth, we performed additional experiments to investigate whether Mtss1-mediated Plexin-D1 localization to the growth cones accelerates the repulsive response to the Sema3E ligand. We performed a growth cone collapse assay in cultured MSNs (Figures 6A-6F). In the initial submission, we found that WT Drd1a-tdT-positive MSNs showed a high collapse rate after exogenous Sema3E treatment, but MSNs lacking Mtss1 exhibited a lower collapse rate. For this revised manuscript, we performed rescue experiments by reintroducing WT Mtss1 into Mtss1-KO MSNs. We found that reintroduction of Mtss1 resulted in significant growth cone collapse, whereas overexpression of Mtss1 without the I-BAR domain, which does not bind to Plexin-D1, showed a reduced response to Sema3E (Figures 6A-6D). We also observed an irregular projection pattern in Mtss1 cKO mice or Plxnd1 cKO or Sema3e KO mice (Figure 8 and Figure 7—figure supplement 10). These results showed that the repulsive signaling by Plexin-D1 in response to Sema3E was significantly impaired by Mtss1 deficiency. However, unlike SH3BP1 or GIPC (Tata et al., 2014; Burk et al., 2017), Mtss1 was not involved in Plexin-D1 endocytosis and Plexin-D1 display on the plasma membrane (Figure 5—figure supplements 5A-D). Therefore, these results suggest that Mtss1 functions as a facilitator of Plexin-D1 trafficking and enhances the repulsive guidance of Sema3E-Plexin-D1 signaling.

Finally, we conducted experiments to explore whether Mtss1, under the Sema3E-Plexin-D1 signaling, also plays a role in the vasculature as a downstream mediator of common guidance cues. However, we found that (1) Mtss1 is not expressed in the developing vasculature and (2) its expression is not induced in the cultured endothelial cells (Figure 8—figure supplement 11). These results suggest that Mtss1 activation by Sema3E-Plexin-D1 signaling pathway and its function in neurons appear to be selective and distinct. We discussed this in the revised manuscript (Lines 538-551)

Taken together, we believe that our new data provide more compelling evidence to elucidate the mechanism of Mtss1 expression and its functional association with Plexin-D1.

Reviewer #1 (Recommendations for the authors):In this study Kim et al. investigate mechanisms of axon extension and repulsive guidance mediated by signaling involving the secreted semaphorin 3E (Sema3E), its receptor plexin D1 (PlexD1), and the cytosolic membrane remodeling protein metastasis suppressor 1 (Mtss1). Sema3E/PlexD1 signaling has been shown to mediate classical repulsive guidance in developing CNS neurons and vasculature, and also select synapse formation for direct-pathway medium spina neurons (MSNs) that project to the substantia nigra (SNr). Here, the authors show using bulk transcriptomic profiling in striatal tissue that in PlexD1-/- mutants there is a dramatic down regulation of Mtss1 mRNA expression, and this can be observed in the striatum in situ and also when assessing Mtss1 protein biochemically. In heterologous non-neuronal cells in culture the authors show that the PlexD1 cytoplasmic domain and also the Mtss1 I-BAR domain are required for a direct interaction between these two proteins in a Sema3E-independent manner (similar assessment of these interactions in neurons in vivo or culture is not presented). In COS cells it is observed that co-expression of PlexD1 and Mtss1 leads to protrusive shape changes with co-localization of these two proteins that appears in part dependent upon the Mtss1 BAR domain-similar co-localization is seen in striatal neurons in culture. Importantly, Mtss1 is shown to be required for Sema3E neuronal growth cone collapse, and also for normal neurite outgrowth. Significant but modest defects in neuronal pathways reaching the SNr are presented in Mtss1 mutants, and these are shown in the context of the striatonigral pathway to be similar to what is observed in PlexD1 mutants.Overall, this study is interesting with regard to regulation of Mtss1 mRNA expression by the presence of PlexD1, and the requirement for Mtss1 to mediate Sema3E repulsive signaling in neurons in vitro. However, we do not learn whether or not Mtss1 functions more generally in PlexD1-expressing cells that respond to Sema3E (i.e. vasculature), what the mechanism of action is regarding how PlexD1 regulates Mtss1 translation, or what the actual functional link is between Mtss1 and Sema3E/PlexD1 signaling-much of these data presented here are correlative.1. Figure 1- Are similar effects on Mtss1 expression seen in Sema3E mutants? This is key, regardless of the result, in order to inform conclusions relating to actual activation of PlexD1 signaling with regulating of Mtss1 expression. Further, is there evidence directly showing that in neurons stimulated with Sema3E, Mtss1 expression is increased and, as a control, that this is not observed in Plxnd1-KO neurons. In Figure 1H, is Mtss1 present only in neuronal bodies? And what is the Mtss1 expression pattern in theSNr? Also, is there an explanation for why at P7 (Figure 1G) Mtss1 decreases while PlxnD1 is still elevated?

We deeply appreciate and value the insightful feedback the reviewer have provided us. We performed the experiments as you suggested and found a reduction in Mtss1 expression in the striatum of Sema3e KO mice (Figures 1F-1G) as well as in the cultured neurons isolated from Sema3e KO (Figures 2C-2D).

As your suggestion, we checked the expression of Mtss1 in neurons after activating Sema3E-Plexin-D1 signaling by treating with Sema3E ligand. Mtss1 expression was increased by Sema3E replenishment in Sema3e-knockout neurons, whereas it was not altered in Plxnd1-knockout neurons (Figures 2E-2I).

Regarding the localization of Mtss1, we observed Mtss1 in the neuronal bodies in the striatum (Figure 1J) and observed in striatonigral axon tract and SNr (Figure 7—figure supplement 7). Intriguingly, we also found Mtss1 in the axonal side of the cultured MSNs at DIV3 (Figure 3A). While we observed the presence of Mtss1 in the SNr region, we cannot exclude the possibility that substantia nigra neurons may express Mtss1 at this stage.

We appreciate the reviewer’s comment. We have also been contemplating deeply about the issue that you have raised. Our study revealed that the regulation of Mtss1 expression is not solely dependent on Sema3E-Plexin-D1 signaling. As shown in Figures 1K, 1L, 2A, and 2B, Mtss1 expression was observed at basal levels both in vivo and in cultured MSNs even in the absence of Sema3E-Plexin-D1 signaling. This indicates the presence of an additional regulatory mechanism in Mtss1 expression. During the developmental stage of P7, which coincides with the arrival of axons at their target destination, there is a noticeable decrease in the expression of Mtss1. This correlation between the reduction of Mtss1 expression and the completion of axonal projection has led us to speculate that retrograde signaling, possibly mediated by target-derived factors, may play a role in regulating the expression of Mtss1. A possible alternative explanation is that certain factors that suppress the expression of Mtss1 may begin to be expressed as Drd1a+ MSNs mature. We discussed in the revised manuscript (Lines 403-417).

2. Page 6 – The claim that failure to see elevated Mtss1 expression in cultured striatal neurons shows that Mtss1 expression is regulated in a "cell autonomous manner" is not supported by this experiment since non-cell-autonomous interactions among these neurons in culture are not ruled out using this approach.

We have removed the mentioned phrase in the revised manuscript in accordance with your suggestion.

3. Figure 2 – Are any of these interactions assessed at the level of Western analysis between PlexD1 and Mtss1 observed in neurons, either in culture or in vivo? Further, if PlexD1 loss-of-function (LOF) leads to loss of Mtss1 expression, are the direct interactions between PlexD1 and Mtss1 in any way required to regulate Mtss1 expression (either in heterologous cell culture or in neurons)? Indeed, is Mtss1 mRNA expression regulated by Sema3E/PlexD1 in COS cells? Further, given results from others involving neuropilin-1 (Nrp1) in PlexD1-medated CNS axon guidance (Chauvet et al., 2007), has this co-receptor component been explored in the context of these interactions with Msst1 or Sema3E-dependence for them (or for that matter any other aspects of Mtss1 function in subsequent experiments in this study)?

We appreciate the reviewer’s suggestions. We quantified the interaction between Mtss1 and Plexin-D1 in Figure 4F. We attempted to perform co-immunoprecipitation with in vivo striatum tissue to see the direct interaction between them. However, due to the nonspecific band issues of similar sizes, we were not able to clearly see their interaction in vivo and cultured neurons. Instead, we confirmed their direct interaction using pull-down assay using purified proteins as shown in Figure 4G. Furthermore, we performed additional immunoprecipitation experiments to test whether Mtss1 binds to other Plexin family proteins or, conversely, whether Plexin-D1 can interact with any BAR domain-containing protein (Figure 4—figure supplement 4). Our data clearly show that the formation of Mtss1-Plexin-D1 complex is relatively specific.

In this study, despite endogenous Mtss1 expression, Sema3E treatment did not induce Mtss1 expression in the Plexin-D1-overexpressing Cos7 cells or endothelial cells (Figure 3—figure supplements 3A-B and Figure 8—figure supplement 11C). Thus, we do not believe that the Plexin-D1-Mtss1 interaction is directly involved in the regulation of Mtss1 expression in general. Instead, we clearly demonstrated that Mtss1 functions as a specific transporter for Plexin-D1 trafficking in the direct-pathway MSNs (Figures 5G-5M).

Neurophilin is not expressed in the striatum (Ding et al., 2013). We mentioned that in the discussion part of revised manuscript (Lines 470-471).

4. Figure 3 – Does Sema3E affect the co-localization of PlexD1 and Mtss1 observed in COS cells? Further, the statement on page 8 "….These results confirm that the Plexin-D1 and Mtss1proteins form a complex in specialized cell structures such as filopodia…" is not supported by these imaging data-at best this modest co-localization is correlative.

We are thankful for your question and comments. We observed no effect of Sema3E treatment on the co-localization of Plexin-D1 and Mtss1 in the immunocytochemistry of COS cells (Figure 5—figure supplement 5H) and co-immunoprecipitation (Figures 4E-4F). The results suggest that ,unlike SH3BP1 that binds to Plexin-D1, Mtss1 is not involved in the Sema3E-Plexin-D1 signaling pathway but participated in trafficking of Plexin-D1 from cell body to axons.

We have deleted the sentence in the revised manuscript as you mentioned.

5. Figure 4 – These data show a degree of co-localization of overexpressed PlexD1 and Mtss1 in Mtss1-/- neurons and the strong result that Mtss1 is required for Sema3E collapse. However, the conclusion that somehow PlexD1membrane targeting is the mechanism that mediates this effect is not strongly supported by these data. Does loss of Mtss1 affect membrane targeting of exogenous and also endogenous PlexD1? Can a transmembrane domain-containing PlexD1 protein targeted to the membrane affect neuronal morphology independent of Mtss1? Also, does the Mtss1 Bar-domain deletion construct fail to rescue Sema3E growth cone collapse in these neurons in vitro? Finally, what is the link to PlexD1 signaling with regard to effects of Mtss1 on neurite outgrowth? Clearly loss of Mtss1 does affect neurite outgrowth, but this is not directly linked in this study to PlexD1 and appears to be independent of Plexin signaling.

We appreciate the reviewer’s comments. As shown in Figure 5—figure supplements 5A-D. Mtss1 did not affect membrane targeting of Plexin-D1 nor endocytosis of Plexin-D1. Mtss1 was crucial for transportation of Plexin-D1 to the growth cone, as shown in the time-lapse imaging and AP-Sema3E binding experiments in cultured MSNs (Figure 5) and aided to receive repulsive signals to Sema3E (Figure 6). Currently, it is not clear whether Plexin-D1 and Mtss1 are present as a complex at the membrane surface of the growth cone, but at least Mtss1 contributes to Plexin-D1 transport between the soma and the growth cone. We discussed this in the revised manuscript (Lines 363-368).

Ectopic expression of vsv-Plexin-D1DICD, which does not interact with Mtss1 (Figure 4C) did not rescue the reduced collapse phenotype shown in Plexin-D1-deficient MSNs, whereas full-length Plexin-D1 did. (Figure 6F).

Mtss1 lacking I-BAR domain construct failed to rescue the reduced collapse phenotype shown in Mtss1-deficient MSNs (Figures 6A-6D).

To explore whether the phenotype in Plexin-D1 null MSNs at DIV6 due to the reduction of Mtss1, we performed the rescue experiment by overexpressing Mtss1 in Plxnd1 KO neurons. As shown in Figures 3K-3L, the neurite outgrowth defect was recovered, suggesting that Plexin-D1 controls the axonal length of MSNs at DIV6 via regulation of Mtss1 expression in cultured MSNs.

Also, what are the growth rates and growth cone responses at DIV6? Based on the Western blos from Figure 1L, there is a large difference between expression of Plxnd1/Mtss1 at DIV3 (when Sema3E-treatments were done in Figures 4H and Suppl. Figure 5) and at DIV 6.

We appreciate the reviewer’s comments. We performed additional experiments at DIV6. As shown in Figures 3I-3J and 6E, the growth rates and growth cone responses of Plxnd1-null MSNs at DIV6 were significantly reduced. Neurite outgrowth in Mtss1-deficient MSNs was impaired at both DIV3 and DIV6. However, axon length abnormalities in Plexin-D1 null MSNs were only observed at DIV6, but not at DIV3 (Figures 3B-3J). We believe that phenotype is due to the fact that Mtss1 was not regulated by Plexin-D1 in the cultured MSNs at DIV3, as shown in Figures 2A and 2B.

6. Figure 5 – I am not convinced that there is reduced PlexD1 localization on axon trajectories-Sema3E-AP binding is reduced, however so is the trajectory be assessed, so in the absence of a ratiometric assessment of PlexD1 (as evidenced by Sema3E-AP binding) this is not convincing. Further, the statement "….Since no significant change in Plexin-D1 levels was observed in the striatum of Mtss1-deficient mice compared to those in littermate controls…" is not supported by bulk Western analysis of PlexD1 protein since no neuronal localization information is provided by these data.

We greatly appreciate to your profound insight and suggestions. As you suggested, we have performed a ratiometric assessment of Plexin-D1 and axons in WT or Mtss1-KO (Figures 7A-7E and 7N-7Q). At E17.5 in Mtss1 cKO mice, both Plexin-D1 and neurofilament decreased in the striatonigral tract, but the decrease in Plexin-D1 was significantly greater than that in neurofilament, as shown in Figures 7A-7E. Furthermore, we performed additional experiments to convince that Mtss1 affects both striatonigral extension and Plexin-D1 trafficking by quantifying the intensity of Plexin-D1 relative to tdT expression in neonatal pups at P5 (Figures 7N-7Q). We believe that these additional data contribute to more confident conclusions in our study.

7. Figure 6 – Have the authors looked in cross section at the Drd1a-tdT-labeled striatonigral pathways in order to provide data supporting the schematic in panel L? This could include higher resolution assessment of axon numbers in this pathway, which would support the overall conclusions regarding guidance effects mediated by Mtss1.

As the reviewer recommended, we improved the images of axons in the Drd1a-tdT-labeled striatonigral pathway by using Leica TCS SP8 confocal microscope (Figures 7K, 7R, Figure 7—figure supplement 8A, Figure 7—figure supplement 8D, and Figure 7—figure supplement 10A).

8. Have the authors attempted to generalize their analysis of Mtss1 function to the developing vasculature? Given that the strongest results here relate to regulation of Mtss1 mRNA expression by PlexD1 and also the requirement for Mtss1 for neuronal growth cone collapse, generalizing Mtss1 function to other PlexD1-expressing tissues known to respond to Sema3E would strengthen this study.

We appreciate the reviewer’s constructive comments on our study. To address the reviewer’s points, we obtained endothelial cell-specific Tie2-cre mice and investigated the expression of Mtss1 and its potential role in the developing vasculature using Mtss1 cKO mice (Tie2-cre; Mtss1^f/f^). As shown in Figure 8—figure supplement 11, our results showed that Mtss1 mRNA was not as highly expressed as Plexin-D1 mRNA in the vasculature at E14.5, and we did not observe any significant phenotype in the Mtss1 cKO mice unlike Plxnd1 cKO mice (Gu et al., 2005). In addition, when we performed the Sema3E ligand treatment experiment in endothelial cell lines (HUVEC (human umbilical vein endothelial cells) or HCMEC/D3 (human cerebral microvascular endothelial cells)), Mtss1 was not induced by Sema3E-Plexin-D1 signaling, unlike in the cultured MSNs (Figure 2 and Figure 8—figure supplement 11C). We discussed this in the revised manuscript (Lines 538-554).

9. The authors are urged to review work in the guidance field involving the regulation of transcription by guidance cue receptors so that statements in the Discussion can be brought in line with work in this area (for example, netrin-mediated DCC signaling to the nucleus that regulates commissureless expression-see Russell and Bashaw, 2019, for a review).

We added it in the discussion of revised manuscript (Lines 375-379).

Reviewer #2 (Recommendations for the authors):Kim, Li et al. identified Mtss1 as a molecule regulated by Sema3E-Plexin-D1 signaling and investigated its roles in striatonigral axon growth and turning. They first showed that Sema3E-Plexin-D1 signaling induces Mtss1 expression in striatonigral projecting neurons. They then showed that Mtss1 physically interacts with Plexin-D1 and that Plexin-D1 localization at the axon growth cone appears to be perturbed in the absence of Mtss1. Finally, the authors showed that striatonigral axonal projections are impaired in Mtss1 KO mice, an outcome that is phenocopied in Plexin-D1 KO mice. With these data, the authors propose that Mtss1 is upregulated in response to Sema3E, possibly from the thalamostriatal neurons, through Plexin-D1, then interacts with Plexin-D1, facilitates the transport of Plexin-D1 to the growth cone, and sensitizes striatal axons to Sema3E for axonal repulsion, while contributing to axonal extension. This is a novel and exciting model proposing how axonal growth and pathfinding are regulated in the thalamo-striato-nigral circuit. However, additional data would help fully support their model. Furthermore, images and quantifications could be improved, and quantification methods could be clearly described. Specific points are listed below.1. An interesting aspect of the proposed model is that Sema3E from the thalamostriatal neurons upregulates Mtss1 expression in striatonigral neurons. However, currently, the paper only includes the expression pattern of Sema3E. The authors could experimentally show the role of thalamic Sema3E.2. Another interesting aspect of the proposed model is that Mtss1 binds to and transports Plexin-D1 to the growth cone. While the authors' data suggest that Plexin-D1 in the growth cone appears to be altered in the absence of Mtss1, the authors could demonstrate that indeed Mtss1 and Plexin-D1 are co-trafficked along axons.3. The quality of images could be improved. Also, quantifications are missing for several figures. Furthermore, where quantifications are done, the methods used for the quantifications are not clearly described in the methods.4. The discussion can be expanded. The authors may discuss more about their proposed model (Supplementary Figure 7).Specific suggestions are listed below.1. An interesting aspect of the proposed model is that Sema3E from the thalamostriatal neurons upregulates Mtss1 expression in striatonigral neurons. Regarding this point, the authors could inactivate Sema3E from thalamostriatal neurons and demonstrate the effect on Mtss1 expression and striatonigral axonal pathfinding.

We highly appreciate the reviewer’s insightful and helpful comments on our manuscript. To address the suggested concern, we observed the phenotype of striatonigral tract in Sema3e KO mice, as observed in Plxnd1 cKO or Mtss1 cKO mice. In Sema3e KO mice, we found a significant reduction in the width of the striatonigral tract, accompanied by irregular projections, as shown in Figure 7—figure supplements 10A-G. It is important to note, however, that this phenotype was comparatively milder than that observed in Plxnd1 cKO or Mtss1 cKO mice. Typically, the receptor phenotype is more severe than the ligand phenotype in axon guidance molecules. This trend is evident in Figures 7K-7M, 7R-7T, 8A-8F, Figure 7—figure supplements 8A-H, and Figure 7—figure supplements 10A-G, where the phenotype in Plxnd1 cKO or Mtss1 cKO mice is more pronounced than in Sema3e KO mice.

2. Another interesting aspect of the proposed model is that Mtss1 binds to and transports Plexin-D1 to the growth cone. Regarding this point, the authors could demonstrate that indeed Mtss1 and Plexin-D1 are co-trafficked along axons, ideally with live imaging.

We appreciate the reviewer’s constructive comments. We checked the localization of Mtss1 and Plexin-D1 along axons by using structured illumination microscopy (N-SIM). As shown in Figures 5A-5C, Plexin-D1 and Mtss1 proteins colocalized along growing axons, but the Mtss1 mutant lacking the I-BAR domain showed reduced Plexin-D1 levels as well as a low colocalization rate (Due to the reduced intensity, we employed Costes’ randomized pixel scrambled image method for quantification).

To further address the trafficking issue, we conducted time-lapse live imaging experiment. We generated a Plexin-D1-GFP and Mtss1-RFP fusion DNA construct and co-transfection into MSNs. Unfortunately, we were unable to detect co-transfected healthy MSNs due to extremely low transfection efficiency. Instead, we analyzed the live trafficking of Plexin-D1 from WT or Mtss1 KO MSNs after single transfection of the Plexin-D1-GFP construct (Figures 5G-5K, Videos 1–4). To analyze Plexin-D1 trafficking, we utilized Particle Tracking Recipe in AIVIA microscopy image analysis software (Aivia lnc.). As shown in Figures 5G-5I, both velocity and total distance of Plexin-D1-GFP were significantly reduced in MSNs derived from Mtss1 KO mice. This observation is consistent with the diminished levels of endogenous Plexin-D1 in the growth cone of MSN derived from Mtss1 KO mice (Figures 5L-5M).

3. The quality of images should be improved. It is difficult to evaluate colocalization (e.g., Figure 3C, 4A), membrane localization (e.g., Figure 1H), axonal growth and pathfinding from the images presented. Also, quantifications are missing for several figures (including staining and Western blotting). Please quantify (e.g., Figure 1G, 1H, 2). Furthermore, where quantifications are done, the methods used for the quantifications are not clearly described in the methods. Please describe the methods, including the methods of normalization and verification of reproducibility (e.g., Figure 3D, 4B, 4C, 5-7).Related to this point, Pearson's R may not be sufficient to establish colocalization as it could be influenced by the amount of protein expression. The authors may perform additional colocalization analysis, for example, by using a pixel scrambled image and demonstrating that the colocalization is not random. Also, the authors may consider quantifying the percent of Mtss1 that is positive for Plexin-D1 and vice versa.

We appreciate the reviewer’s comments. We have improved the quality of the images (Figures 5D, 7K, 7R, Figure 7—figure supplements 8A and 8D). And we softened our claims about the colocalization in Figure 5—figure supplement 5F because the reviewer #1 also pointed out it. We replaced Figure 1H into Figures 1H and 1J in the revised manuscript. In addition, we have added the new supporting data in Figure 7—figure supplement 7 to clearly show the localization of Mtss1 in the striatonigral tract in vivo. As the reviewer recommended, we performed the quantifications (Lines 106-107, Figures 1N, 4F, 5C, 5F, and Figure 5—figure supplement 5G). Moreover, the detailed methods for the quantification were described in the Figure legend section and Quantification and Statistical analysis section in Materials and methods (Lines 846-859).

As commented by the reviewer, we employed Costes’ randomized pixel scrambled image method in Figures 5C, 5F, and Figure 5—figure supplement 5G. Images were analyzed for colocalization using the Just Another Colocalization Program (Jacop) plugin on ImageJ, which statistical data are reported from the Costes’ randomization based colocalization module (Bolte and Cordelieres, 2006) to account for the influence by the amount of protein expression. We described that in the Figure legend section and Quantification and Statistical analysis section in Materials and methods (Lines 846-859).

4. The discussion can be expanded. The authors may discuss more about their proposed model (points #1 to #4 in Supplementary Figure 7), including the mechanistic insights into the roles of Mtss1 in axonal turning vs. growing.

We expanded our Discussion section in the revised manuscript as the reviewer recommended.

Additional comments:1. The term "cell-autonomous" may not be used appropriately by the authors in the paper. Since the authors propose that Sema3E from the thalamus activates Plexin-D1 to upregulate Mtss1, this does not appear cell-autonomous.

We have removed the phrase in the revised manuscript in accordance with your suggestion.

2. Plexin-D1-Mtss1 binding in a cell line may involve other molecules. Hence the binding could be indirect.

We appreciate the reviewer’s comments. To explore whether the binding between Plexin-D1 and Mtss1 is direct or not, we performed an in vitro binding assay with the purified Plexin-D1 (intracellular domain) and the I-BAR domain of Mtss1 protein. As shown in Figure 4G, they directly bind to each other.

3. In Figure 4, if Plexin-D1 is not localizing to the growth cone, then where is it localized? The authors could include a lower magnification image showing the cell body and axon and quantify Plexin-D1 localization in these cellular compartments.

We appreciate the reviewer’s feedback. We quantified the levels of Mtss1 and Plexin-D1 in the cell body (Figure 5—figure supplements 6A-B) and along axons (Figures 5A-5C) using structured illumination microscopy (N-SIM). Inhibiting the interaction with Mtss1 significantly reduced Plexin-D1 levels in axons. However, the expression levels of vsv-Plexin-D1 or Mtss1-myc or Mtss1DI-BAR -myc in the cell body remained unchanged, at least in our overexpression system. We speculated that this was likely due to the limited proportion of proteins transported to the nerve endings within the total protein pool. Although the extent to which the synthesized proteins in the cell body can be transported to nerve terminals is not fully understood, a previous study in chick peripheral nerve revealed that about 5% of slowly transported proteins can reach to the axoplasm (Droz B. et al., 1973). Thus, we believe that the majority of newly synthesized Plexin-D1 proteins are stored in the cell body and actively transported either anterogradely or retrogradely on demand. We mentioned this possibility in the result part of revised manuscript (Lines 231-234). As shown in Figures 5G-5K and Videos 1-4, the movement of Plexin-D1-positive vesicles dramatically reduced. These results indicate that Mtss1 facilitates the dynamic transportation of Plexin-D1 along the growing neurites of MSNs, leading to an increased rate of Plexin-D1 localization in the growth cones, as shown in Figures 5L-5M. We discussed this in the revised manuscript (Lines 418-432).

4. The use of AP-Sema3E is an established way of detecting Plexin-D1 localization. However, since it is expected that Mtss1 KO mice would have defects in both Plexin-D1 expression/localization and axon projections, the Sema3E binding may not be sufficient to conclude both "poor neuronal projection" and "reduced Plexin-D1 localization". Additionally, here, too, the authors need to explain how the quantifications were done to determine the Plexin-D1 positive path (%).

We appreciate the reviewer’s valuable comments. As the reviewer#1 suggested, we conducted a ratiometric evaluation of Plexin-D1 and axons in WT or Mtss1-KO, as shown in Figures 7A-7E and 7N-7Q. At E17.5 in Mtss1 cKO mice, both Plexin-D1 and neurofilament exhibited a decrease in the striatonigral tract. However, reduction in Plexin-D1 was notably more significant than that in neurofilament, as shown in Figures 7A-7E. Additionally, we performed supplemental experiments to substantiate that Mtss1 influences both striatonigral extension and Plexin-D1 trafficking by quantifying the intensity of Plexin-D1 relative to tdT expression in neonatal pups at P5 (Figures 7N-7Q). We believe that these additional data contribute to more confident conclusions in our study. We described the quantification method in Figure7 legend (Lines 1204-1207).

5. In Supplementary Figure 4E, Mtss1deltaWH2-myc seems to have much weaker effects. Since this construct contains the I-BAR domain, the authors may want to add some explanation.

We appreciate the reviewer’s comments. Mtss1 lacking the WH2 domain showed a much weaker effect because WH2 is an important region for Mtss1 interaction with F-actin (Mattila et al., 2003). We explained that in the result part of revised manuscript (Lines 148-149).

Reviewer #3 (Recommendations for the authors):In the developing nervous system, the axons of newly generated neurons extend toward destination targets following an exquisitely designed program. Axon guidance molecules are critical for neuronal pathfinding because they regulate both directionality and growth pace. This study describes a novel role for a Mtss1 in axon guidance. In general this is a good study but as Mtss1 has not been found earlier to be expressed in axons, expression and localization of endogenous Mtss1 in axons should be shown convincingly.I think that authors convincingly show that Sema3E-Plexin-D1signaling regulates Mtss1 expression in projecting striatonigral neurons. Also Plexin-D1 – Mtss1 interaction seems clear as well. I like the idea that Plexin-D1 brings Mtss1 to filopodia where Plexin-D1 can bind Sema3E. However, the results presented for this idea were not convincing. First of all, expression and localization of endogenous Mtss1 should be shown. This is especially important because we did not find Mtss1 from axons (Saarikangas et al., Dev Cell 2015, Supplementary figure 2). There can be many reasons for this, age or cell type, for example, but due to this controversy, expression and localization of endogenous Mtss1 in axons must be shown convincingly.In Figure 4, overexpressed Mtss1 localization is not supporting the idea of bringing Plexin-D1 to filopodia and plasma membrane. To me this localization looks quite strange. I am not sure what to do with this but maybe localization of endogenous Mtss1 will help here. Maybe overexpressed construct is not folding right? Is it dynamic? Actin looks strange in this cell as well (Figure 4A) (F-actin in the middle, where are filopodia, arcs, lamellipodia?).

We appreciate the valuable comments from the reviewer. The indicated image has been replaced with a new one (Figure 5D). We believe that the F-actin enriched structure in the center of the growth cone is an F-actin arc. Furthermore, Mtss1 was not involved in the plasma membrane localization of Plexin-D1 (Figure 5—figure supplements 5A-B). Instead, Mtss1 facilitates Plexin-D1 trafficking from the cell body to the axons (Figures 5, 7A-7E, 7N-7Q, and Videos 1-4). Notably, we observed endogenous Mtss1 expression in axons of cultured MSNs (Figure 3A) at DIV3 in vitro and in striatonigral projecting axons (Figure 7—figure supplement 7) in vivo.

Furthermore, the text says on line 195 that "Mtss1 targeting of Plexin-D1 to the growth cone is critical for robust Sem3E-induced repulsive signaling." For this, it would be good to show Plexin-D1 localization (endogenous) in Mtss1 cKO cells vs. control cells. Is there a change? Mtss1 overexpression seems to keep Plexin-D in the middle of growth cone rather than bringing Plexin-D to the filopodia and plasma membrane (Figure 4A). Plexin-D localization without Mtss1 overexpression is missing (is there a change in localization?).

We appreciate the reviewer’s constructive suggestions. As shown in Figures 5L-5M, we observed the diminished level of endogenous Plexin-D1 in the growth cone of MSN derived from Mtss1 cKO mice compared to control. The result is consistent with the significant reduction of velocity and total distance of Plexin-D1-GFP in Mtss1-deficient MSNs in time-lapse live imaging experiment (Figures 5G-5K and Videos 1-4).

Text says on lines 218-221 that "In mice expressing wild-type Mtss1 showed a significant level of Plexin-D1 in the neuronal pathway reaching the substantia nigra, whereas Mtss1-knockout mice exhibited poor neuronal projection and reduced Plexin-D1 localization on E17.5 (Figure 5A).I have struggled by myself in measuring dendrite length in vivo and therefore I ask, how it was ensured that slices are from same depth and same cutting angle? If they vary, does it affect on results?

We appreciate the reviewer’s comments. Regarding to the depth, we collected all sections from brain samples and matched them individually by comparing other brain areas such as ventricles, hippocampus, and cerebellum. For the section angle, we placed the brain in a plastic cryomolds in the correct orientation.

These are the critical issues. If Mtss1 is not expressed in axons (just as a possibility), is it possible that it affects axons by other mechanisms?

As shown in Figure 3A and Figure 7—figure supplement 7B, we observed endogenous Mtss1 expression in axons of MSNs at DIV 3 and in striatonigral tracts in vivo at this stage. However, we did not exclude the possibility that Mtss1 has functions in dendrites of MSNs. We discussed that in the Discussion section of the revised manuscript (Lines 418-454).